# Multiscale estimation of the field-aligned current density

Costel Bunescu[1,2], Joachim Vogt[2], Octav Marghitu[1], and Adrian Blagau[1]

[1]Institute of Space Science, Bucharest, Romania.
[2]Department of Physics and Earth Science, Jacobs University Bremen, Bremen, Germany.

**Correspondence:** C. Bunescu (costel@spacescience.ro)

**Abstract.** Field-aligned currents (FACs) in the magnetosphere-ionosphere (M-I) system exhibit a range of spatial and temporal scales that are linked to key dynamic coupling processes. To disentangle the scale dependence in magnetic field signatures of auroral FACs and to characterize their geometry and orientation, Bunescu et al. (2015) introduced the multiscale FAC analyzer framework based on minimum variance analysis (MVA) of magnetic time series segments. In the present report this approach is carried further to include in the analysis framework a FAC density scalogram, i.e., a multiscale representation of the FAC density time series. The new technique is validated and illustrated using synthetic data consisting of overlapping sheets of FACs at different scales. The method is applied to Swarm data showing both large-scale and quiet aurora as well as mesoscale FAC structures observed during more disturbed conditions. We show both planar and non-planar FAC structures as well as uniform and non-uniform FAC density structures. For both synthetic and Swarm data, the multiscale analysis is applied by two scale sampling schemes, namely the linear and the logarithmic scanning of the FACs scale domain. The local FAC density is compared with the input FAC density for the synthetic data, whereas for the Swarm data we cross-check the results with well established single- and dual-spacecraft techniques. The entire multiscale information provides a new visualization tool for the complex FAC signatures that complements other FAC analysis tools.

## 1 Introduction

The dynamics of the magnetosphere-ionosphere (M-I) system at auroral latitudes is essentially controlled by solar wind-magnetosphere (S-M) coupling, subject to ionospheric feedback. One result of the dynamic interaction in the global S-M-I system is the accumulation of magnetic flux in different parts of the system, e.g., the magnetotail. The energy in the large-scale components is transported and dissipated to smaller scale components of the system, e.g., in the polar ionosphere. The transfer of energy and momentum in the system is mediated by field-aligned currents (FACs) flowing along the ambient magnetic field lines, and driving the formation of ionospheric (Hall and Pedersen) currents. The entire chain of the energy flow and conversion mechanisms is governed by a multiscale behavior both in time and in space. The multiscale character is observed in all the measurable quantities associated with the system, like magnetic field measurements from above (spacecraft) and below (ground) the ionosphere. While above the ionosphere one measures the magnetic perturbation of the field-aligned current (closed in the ionosphere mainly by the Pedersen current), the magnetic perturbation observed on ground is related mainly to the Hall component of the ionospheric current. The multiscale character is observed also in the measurements of optical emissions, associated in turn with a multiscale particle precipitation pattern.

The spatial and temporal scales of the auroral arcs observed optically on ground are dependent on the characteristics of the optical instruments (e.g. resolution, sampling frequency, coverage, exposure). Earlier statistical measurements of the auroral arc thickness (Maggs and Davis, 1968) were based on narrow field of view (FoV) TV camera observations and found a median of the scale distribution around 230 m in the range of fine and small scale auroral arcs (70 m-1.5 km). Later measurements (Knudsen et al., 2001) based on All-Sky-Imager (ASI) observations found a maximum of the scale distribution around 18 km in the range of mesoscale arcs (10-100 km). The TV and ASI observations also correspond to different temporal scales because of the large sampling frequency difference, maximum at about ∼25 Hz for TV and ∼0.3 Hz for ASI. Note that arcs which are not quasi-stationary at the exposure timescales are likely to be smeared and integrated to larger scale structures in the optical data. More recently, Partamies et al. (2010), showed measurements based on intermediate FoV optics (FoV of 20° and spatial resolution of 100 m) with a median of the arc widths distribution around 0.5-1.5 km. Partamies et al. (2010) observations fit in between the previous fine and mesoscale arc width distributions. While these studies concentrated on the visible arcs, Trondsen and Cogger (1997) addressed the scale distribution of the black aurora, found to peak around 400-500 m with an average of 615 m (range between 200 m and 1 km). A review on the optical aurora (caused by electrons) with spatial and temporal scales below 1 km and 1 s, respectively, is given by Sandahl et al. (2008). Overall, the results of all these studies together indicate a rather continuous scale spectrum (Partamies et al., 2010).

FAC structures in the auroral zone are typically organized in east-west aligned sheets. The first statistical studies (Iijima and Potemra, 1976a, b) of the large scale FACs separated those into the well known poleward (R1) and equatorward (R2) currents with different orientation depending on the magnetic local time (MLT) sector. This large-scale picture was confirmed later by other studies, e.g. Peria et al. (2013); McGranaghan et al. (2017). Peria et al. (2013) examined the statistical properties of stationary sheet-like FACs (thickness within 10-1000 km and densities larger than 0.1 $\mu$A/m$^2$) observed by FAST. McGranaghan et al. (2017) study, based on Swarm observations, addresses the multiscale character of FACs by separating the FACs contributions from small-scales (∼50 km), mesoscale (∼150 km), and large-scale (∼350 km). Modeling efforts, e.g. He et al. (2012) characterized the FACs properties (e.g. thickness and intensity) as a function of the solar wind properties and geomagnetic indices (e.g. AE index). The internal structure of large-scale FACs, associated with e.g. discrete auroral arcs, shows variability in all observed characteristics (e.g. the spatial and temporal scales, orientation, geometry) depending on MLT and substorm phase. The importance of small-scale FACs is confirmed by Peria et al. (2013) who found that the large-scale FACs account for about 20% of the FAC events and for about half of the total charge transport.

Above the ionosphere, spacecraft observations provide information about the scale distribution and main characteristics of the FACs (mapped to the ionosphere) through the measurements of magnetic fields (upward and downward FACs), associated electric fields (monopolar, converging or diverging bipolar), and particle fluxes (upgoing and downgoing). A scale distribution with a maximum between 4-5 km was obtained by Johansson et al. (2007) using Cluster measurements (3-6 $R_E$ altitude) of intense electric fields (>0.15 V/m). Johansson et al. (2007) found that the associated FACs and density gradients also have typical values within 4-5 km range. Johansson et al. (2007) (Figure 9) also compare the scale distribution with former results. We notice the distribution of the diverging electric fields (Karlsson and Marklund, 1996) observed by Freja with the peak around 4 km. A statistical study of inverted V structures (U shaped potential drops) observed by the FAST satellite (Partamies

et al., 2008) showed typical scale widths of 20-40 km (maximum energies of 2-4 keV). Simultaneous measurements of narrow arc structures (down to a few km) both in particle and optical data were shown by Stenbaek-Nielsen et al. (1998) by analyzing conjugate FAST/aircraft observations. In the small-scale range we mention also the high resolution measurements of fine scale FACs observed by Freja (Lühr et al., 1994) showing a minimum FAC scale of ∼1.7 km for a specific event.

The scale distribution of FACs reflects a variety of M-I coupling mechanisms. At large scales we have a quasi-stationary coupling (FACs closing in the ionosphere), whereas at small and fine scales a time dependent coupling, typically provided by Alfvén waves in different regimes (e.g. shear, kinetic, inertial). The interaction of shear Alfvén waves with the auroral acceleration region (Vogt and Haerendel, 1998; Vogt, 2002) presents a maximum absorption (conversion of Poynting flux to electron energy flux) for wavelengths that are consistent with the scale size of mesoscale auroral arcs. The arc generation
through inertial Alfvén waves (Chaston et al., 2003) shows scales corresponding to fine scale auroral arcs (1 km width) near the polar cap boundary.

Multi-spacecraft missions on low-altitude polar orbits (e.g. Swarm, ST5) offer a high coverage of the auroral oval and enable statistical studies that address the dynamics and stationarity of FACs, more precise FAC estimates, as well as comparison with the currents inferred by ground magnetic field measurements or cross-check with optical observations. Forsyth et al. (2017)
computed the stability of FACs by comparing the lower altitude Swarm satellites (SwA, SwC) FAC density using a shape and an amplitude correlation and found that ∼50% and ∼1-5% of the large- and small-scale FACs, respectively, correlate between the two spacecraft. Previous correlation analysis using SwA/SwC (Lühr et al., 2015) addressed the stationarity and the planar geometry assumption and found small and large-scale FACs stationary on 10 s and 60 s, respectively. Comparison of Swarm FAC density with ground data was done by Juusola et al. (2016). Statistical analysis of the magnetic field perturbation ($\Delta B$)
measured by ST-5 spacecraft (Gjerloev et al., 2011) showed $\Delta B$ dependence on time and scale as well as on the geomagnetic conditions and local time. For small and mesoscale structures the statistical lifetime of the structures varies linearly with the structure scale. The same is true for large-scales, however in this case the lifetime increases faster with the structure scale. The ST-5 data constrained the analysis of Gjerloev et al. (2011) to scale sizes above 20 km, which is situated in the mesoscale range (Knudsen et al., 2001).

Due to the known statistical alignment of the large- and mesoscale FACs with MLT, single-spacecraft methods typically do not consider the orientation of the FACs in the plane perpendicular to $\boldsymbol{B}$. The assumption of east-west alignment was verified by Gillies et al. (2014) in a statistical study of optical observations based on the THEMIS ASI array. Gillies et al. (2014) survey addressed the stable presubstorm auroral arcs to infer their multiplicity and orientation with respect to the magnetic east-west direction. Their results show the prevalence of multiple arc systems with respect to single arcs. Essentially, the quiet
arcs show east-west alignment around 23 MLT and inclination within a few degrees toward north and south at later and earlier times, respectively. The dependence of the tilt angle on MLT is linear, with a variation of about 1° per MLT hour. A similar analysis of the arc orientation was performed by Wu et al. (2017) who found tilts of <10°. Correction of the FAC density with orientation was done by Gillies et al. (2015) using the high resolution Swarm measurements. Due to the small deviations of the arc orientation from the east-west direction they obtained just small corrections when including the orientation. During more
disturbed times one expects to have a higher variability in the arc orientation. We are not aware of statistical studies addressing

the orientation in various substorm phases and at small-scales. In order to obtain more accurate estimates of the FAC density, particularly for the small-scales and locally planar embedded FACs, one has to correct the FAC density by using the orientation information.

With a few exceptions, most of the FAC studies based on Swarm use mainly the low resolution (1 s) data, associated to a mapped scale of $\sim$7.6 km, whereas the full resolution measurements (0.02 s) correspond to $\sim$150 m. Small scale FACs play an important role in different stages of the aurora and a proper multiscale analysis of the FAC density is important. High resolution Swarm data conjugate with THEMIS ASI measurements were used by Gillies et al. (2015) for the study of small-scale pulsating aurora patches. While their findings are related to pulsating aurora, e.g. strong downward currents at the edges of the pulsating form and typically weaker upward currents inside the patches, Gillies et al. (2015) pointed out that the single-spacecraft FAC density provides better identification of the boundaries of the auroral patches, compared to the dual-spacecraft estimate. The small tilt assumption, underlying the single-spacecraft FAC density estimate, is questionable in this case, and likewise for small-scale structures, as proved by e.g. Miles et al. (2018).

To study the multiscale nature of auroral FACs in sufficient rigor and detail, the arsenal of space physics analysis tools ought to be amended with proper multiscale versions of classical methods. The multiscale FAC analyzer (Bunescu et al., 2015), denoted MSMVA, extends minimum variance analysis (MVA) (Sonnerup and Cahill, 1967; Sonnerup and Scheible, 1998) by providing continuous and multiscale information on the planarity and orientation of the FACs. MSMVA allows to identify the location and characteristic scale of the planar FACs. MSMVA was used (Bunescu et al., 2017) to correlate conjugate observations of FACs by FAST and Cluster spacecraft.

This paper extends the MSMVA framework (Bunescu et al., 2015) with the addition of a FAC density scalogram, i.e., a multiscale representation of the FAC density that takes into account the orientation derived from MSMVA. The extended MSMVA framework provides a consistent visualization tool, useful for the analysis of complex FAC systems in terms of their scales. Two different scale sampling schemes are considered and tested using synthetic data and Swarm measurements. The local FAC density around the characteristic scale of the FACs, as identified by MSMVA, is compared with single-spacecraft and dual-spacecraft FAC density estimates (Ritter et al., 2013; Ritter and Lühr, 2006).

The article is organized as follows. Section 2 reviews the MSMVA and describes the multiscale current density. In section 3 the method is applied to the magnetic signatures of synthetic currents showing both large and superposed smaller scale structures. Section 4 shows applications to Swarm events with both quiet and more dynamic, smaller scale FAC features. A discussion is presented in section 5 and the paper is concluded in section 6.

## 2 MSMVA estimation of the FAC density

Statistical studies of FACs are typically carried out in global geocentric coordinates systems such as GEO. Individual crossings are often studied in mean-field aligned (MFA) systems which are local, centered at the spacecraft, and with the third ($z$) axis pointing along the background magnetic field $\boldsymbol{B}$. Then the $y$ and $x$ axis point roughly to the east ($\boldsymbol{B} \times \boldsymbol{R}$ where $\boldsymbol{R}$ is the radial vector to the spacecraft) and to the north, respectively.

In this paper we distinguish between general MFA frames (coordinates $x, y, z$) and reference systems of FAC sheets with coordinates $\xi, \eta, \zeta$. Here $\xi$ is along the sheet normal, $\eta$ is tangential to the sheet, and $\zeta$ points along the ambient magnetic field. The magnetic field perturbation $\Delta\boldsymbol{B}$ (oriented along $\eta$ at an idealized infinite planar sheet) caused by the FAC sheet is obtained after subtraction of an average or model magnetic field from the magnetic vector measurements $\boldsymbol{B}$ (section 4.1).

## 2.1 Principles of single-spacecraft FAC estimation

FAC density estimators can be based on single-spacecraft or multi-spacecraft data (Ritter et al., 2013; Vogt et al., 2013). Here we adopt the single-spacecraft approach to construct a FAC density scalogram, i.e., a multiscale representation of FAC density. Single-spacecraft FAC estimators are based on Ampére's law, $\boldsymbol{j} = \mu_0^{-1}\nabla \times \boldsymbol{B}$, with the field-aligned component given by

$$j_\parallel = j_z = \mu_0^{-1}\left(\partial_x B_y - \partial_y B_x\right) \tag{1}$$

For a sufficiently elongated FAC sheet, in the sheet reference system, Equation (1) reduces to

$$j_\parallel = j_\zeta \simeq \mu_0^{-1}\partial_\xi B_\eta \tag{2}$$

The typical method used to describe the orientation of the FACs is the MVA (Sonnerup and Scheible, 1998) applied to the magnetic field measurements. MVA is based on the assumption of planarity and stationarity. MVA analysis for FACs can be performed on all components of $\boldsymbol{B}$ (3D MVA) or, in a simplified case, on the perpendicular perturbation, $\boldsymbol{B}_\perp$ (2D MVA). The full 3D MVA can be applied in any reference frame, e.g. GEO, MFA and yields $\lambda_{\min}$, $\lambda_{\mathrm{int}}$, and $\lambda_{\max}$ associated to the directions along $\boldsymbol{B}$ ($\boldsymbol{e}_{\min}$), perpendicular ($\boldsymbol{e}_{\mathrm{int}}$) and tangential ($\boldsymbol{e}_{\max}$) to the arc, respectively. The 2D MVA provides only $\lambda_\xi \equiv \lambda_{\mathrm{int}}$ and $\lambda_\eta \equiv \lambda_{\max}$ associated to the normal ($\boldsymbol{e}_\xi$) and tangential ($\boldsymbol{e}_\eta$) directions to the arc, and is rather limited to MFA frames. The GEO frame requires the 3D MVA since we do not have a strict alignment of the $z$ axis with $\boldsymbol{B}$ and part of the variance of $\Delta\boldsymbol{B}$ is contained in the parallel component. We note that various other combinations are also possible by imposing constraints on $\boldsymbol{e}_{\min}$, e.g. aligned with $\boldsymbol{B}$. In the following we use the subscripts min, int, and max when referring to the 3D MVA and $\xi, \eta$ for the 2D MVA.

The analysis performed in this paper is done in the MFA coordinates and takes into account only the variance in $\boldsymbol{B}_\perp$. By using this simplified approach we get a lower variance in the data (not including $\boldsymbol{B}_\parallel$) and thus expect better results with respect to the 3D case. The 2D approach is particularly useful for the case of small scale FACs in order to avoid ambiguous cases where $\boldsymbol{e}_{\min}$ is associated to a perpendicular direction rather than to $\boldsymbol{B}_\parallel$ direction. Moreover, we note that at the low altitude Swarm orbit $B_z$ (or $B_\zeta$) can be affected by large scale remote current systems in the ionosphere, e.g. the electrojet current. A statistical study emphasizing the global characteristics of the Hall current derived from Swarm observations was performed by Huang et al. (2017).

## 2.2 FAC density from single- and multi-spacecraft data

In the idealized case of an infinite planar current sheet oriented along the east-west direction (east-west aligned auroral arcs), the FAC density is approximated by discretizing equation (1) and by using the spacecraft velocity, $\boldsymbol{v}^{\mathrm{sc}}$, to compute the spatial

gradient along the normal to the FAC structure:

$$j_\parallel = (\mu_0 v_\perp^{sc})^{-1} \Delta B_y / \Delta t \tag{3}$$

For the quasi-static FAC approximation and in the case of spacecraft crossing along the normal to the arc, equation (3) gives correct results. In reality, due to the orbital configuration and FAC dynamics, the crossings are not normal to the arc and the FACs show deviations from the quasi-static approximation. Equation (3) was used to obtain estimates of the FAC density

for many single-spacecraft missions like Freja (e.g. Luhr et al., 1996) and FAST (Elphic et al., 1998), or more recently for single-spacecraft FAC estimates from Swarm (Ritter and Lühr, 2006).

    For an east-west aligned FAC sheet, the observed sign of the slope in the $B_y$ time series (with the $y$ axis pointing towards east) depends not only on the FAC direction but also on the direction of the spacecraft velocity $\boldsymbol{V}$ and on the hemisphere. The sign of $B_y$ time series slope equals FAC direction w.r.t. $\boldsymbol{B}_0$ (ambient field) for poleward motion, whereas this relation is

reversed for equatorward motion. The general algebraic relationship for a sheet with normal unit vector $\hat{\boldsymbol{n}}$ is

$$\mu_0 \boldsymbol{j} = \frac{\hat{\boldsymbol{n}} \times \dot{\boldsymbol{B}}}{V_n} = \frac{\hat{\boldsymbol{n}} \times \dot{\boldsymbol{B}}}{\hat{\boldsymbol{n}} \cdot \boldsymbol{V}} \tag{4}$$

For an ideal (infinitely extended) sheet of FACs we obtain

$$\mu_0 j_\parallel = \mu_0 \boldsymbol{j} \cdot \hat{\boldsymbol{B}}_0 = \frac{|\dot{\boldsymbol{B}} \times \hat{\boldsymbol{B}}_0|^2}{(\dot{\boldsymbol{B}} \times \hat{\boldsymbol{B}}_0) \cdot \boldsymbol{V}} \tag{5}$$

since $\dot{\boldsymbol{B}} \times \hat{\boldsymbol{B}}_0$ is aligned with $\hat{\boldsymbol{n}}$. Hence the FAC is positive/negative if the two vectors $\dot{\boldsymbol{B}} \times \hat{\boldsymbol{B}}_0$ and $\boldsymbol{V}$ form an angle smaller/larger than $180°$. Note that in the northern hemisphere, positive FACs are downward currents, and negative FACs are upward currents. In the southern hemisphere, negative FACs are downward currents, and positive FACs are upward currents.

In section 4 we show events with both polarward and equatorward crossing by Swarm spacecraft.

    When multi-spacecraft information is available, one can relax part of the assumptions involved in the single-spacecraft methods to compute the FAC density. For the case of the Swarm mission, the multi-point configuration is constructed by using the low orbit SwA and SwC spacecraft. By shifting the along-track positions one can build virtual quads which make an appropriate configuration for the computation of the FAC density. Based on their computation principle, we distinguish two

classes of dual-spacecraft methods. Finite differencing (FD) methods (Ritter et al., 2013; Ritter and Lühr, 2006) evaluate a discrete version of the boundary integral $j_\parallel = (\mu_0 A)^{-1} \oint \boldsymbol{B} \cdot \mathrm{d}\boldsymbol{s}$. Linear least squares (LS) estimators (Vogt et al., 2009, 2013) are constructed by projecting the dual-satellite measurements onto a local linear magnetic field model.

    While both FD and LS methods have obvious advantages over the single satellite methods, they are limited with respect to the scale resolution. The along-track separation can be varied in order to obtain squared quads configurations whereas the

cross-track is limited by the orbit separation. Thus, the cross-track separation defines the lower limit of the FAC scales in the cross-track direction whereas the limit in the along-track direction is determined by the along-track separation, provided that the FAC structure is quasi-stationary. The typical cross-track separation between SwA and SwC above the auroral oval is decreasing towards poles from $\sim$80 km to $\sim$50 km around latitudes of $\sim$60° to $\sim$70°, respectively. The along-track separation of about 10 s corresponds to some 70 km.

## 2.3 Multiscale FAC density scalogram

In order to characterize the small-scale FACs, one has to rely on single-spacecraft methods. Bunescu et al. (2015) introduced the multiscale FAC analyzer (MSMVA) to study the FAC signatures. The MSMVA technique extends the MVA analysis by providing continuous and multiscale information on the planarity and orientation of the observed FACs. The continuous character over the time domain is achieved by computing the MVA parameters (eigenvalues and eigenvectors) over a sliding window (width $w$). The multiscale character is achieved by repeating the procedure for an array of window widths, $w_k$, within a given range (resolution $dw$). The eigenvalues ($\lambda_\eta$, $\lambda_\xi$), eigenvectors ($e_\eta$, $e_\xi$), eigenvalues ratio, $R = \lambda_\eta / \lambda_\xi$, and the orientation, $\theta \equiv \sphericalangle(e_\xi, \hat{x})$, are thus 2-D quantities dependent on time and scale. Bunescu et al. (2015) showed that the derivative of $\lambda_\eta$ with respect to the length of the analysis window, $\partial_w \lambda_\eta$, provides the location (center) and scale (thickness) of the planar FAC structures. We note that the amplitude of $\partial_w \lambda_\eta$ depends on the scanning parameter $w$ which represents the along track scale. In order to obtain the amplitude corrected derivative we use the orientation information, $\partial_\xi \lambda_\eta = \partial_w \lambda_\eta / \cos(\theta)$. Here after in this work we only use the amplitude corrected derivative $\partial_\xi \lambda_\eta$.

The method was checked on simple synthetic FACs (infinite and finite structures) of both uniform and nonuniform FAC density and showed good performance in identifying FAC scales. The method was applied to Cluster data showing both large scale quiet arcs and locally planar and dynamic FAC structures (Bunescu et al., 2015), as well as for the analysis of conjugate Cluster/FAST observations (Bunescu et al., 2017).

The multiscale information provided by MSMVA can be used to compute other quantities, like the FAC density. MSMVA provides the scale dependent orientation that can be used to compute the FAC density in the FAC's own reference system. Combined with the MSMVA results this provides a consistent tool to analyze the FAC signatures. One can compute the FAC density at each scale by discretizing equation (2)

$$j_\parallel = \mu_0^{-1} \Delta B_\eta / \Delta \xi \tag{6}$$

where $B_\eta$ is computed as the projection of $B$ along the tangential direction, $B_\eta = B \cdot e_\eta$, whereas $\Delta \xi$ is the thickness across the structure in the normal direction. Assuming a certain velocity of the spacecraft, $v_{sc}$, $\Delta \xi$ can be computed by using the projection of $v_{sc}$ on $e_\xi$ and the spacecraft crossing times, $\Delta \xi = v_{sc} \cdot e_\xi \Delta t$. We note that equation (6) provides the amplitude corrected FAC density because at each scale $j_\parallel$ is computed by taking into account the perpendicular scale variation, $\Delta \xi$.

The amplitude of $B_\eta$ at each scale $w = \Delta t$ is estimated by fitting $B_\eta$ using a simple linear regression analysis. Thus, $\Delta B_\eta = B_\eta(t_b) - B_\eta(t_a)$, where $t_a$ and $t_b$ are the limits of the analyzing window, $w$, at the respective position (center $t_{cen}$ of $[t_a, t_b]$ interval). When the analyzing scale is centered on a certain FAC structure and has the width equal to the FAC thickness, $\Delta B_\eta$ approximates well the entire perturbation across the structure. When the analysis window is centered between two balanced FACs of similar amplitude, $j_0$, and thickness, $w_0$, the two FACs cancel each other and provide no contribution to the current at that position and scale, $\Delta B_\eta = 0$. In the case of unbalanced FAC structures, the FAC density depends on their respective amplitudes and thicknesses.

The ensemble of the resulting estimates $j_\parallel = j_\parallel(t_{cen}, w) = j_\parallel(t_{cen}, \Delta t)$ yields a multiscale representation of FAC density in $(t_{cen}, w)$ space. We refer to this graphical representation as the FAC density scalogram, in analogy to the terminology used for wavelet transforms (Torrence and Compo, 1998).

The multiscale information can be separated into invariant information, which depends only on quantities in the local $(\xi, \eta)$ frame, and non-invariant information, which depends also on variables in the $(x, y)$ frame. All multiscale information depends on $w$ which is the scale length along the spacecraft track ($(x, y)$ frame) and thus a non-invariant variable. In order to obtain the dependencies on the perpendicular scale (FAC thickness) one has to correct the scale array $w_k$ by projection along the $\hat{\boldsymbol{\xi}}$ direction, $w_k \cos(\theta_k)$. Regarding the amplitude of the MSMVA quantities we notice that invariant information is given by $R_\lambda$, and $\theta$, whereas non-invariant information by uncorrected $j_\parallel$ and $\partial_w \lambda_\eta$. Corrections to the scale are applied for the individual profiles (dependence at a certain time or position, see sections (3) and (4)) and not to the scalograms of MSMVA quantities. As long as both synthetic and observed FACs are essentially east-west aligned (Bunescu et al., 2015), the method cannot be properly tested and validated for inclined structures. In section 3 we perform tests on inclined synthetic FACs, whereas in section 4 we apply the method also to inclined FAC observations by Swarm.

### 2.4   Scale sampling schemes

We use two different FACs scanning procedures (scaling schemes) for the discretization of the FACs scale domain. The scheme implemented by Bunescu et al. (2015) implies a linear sampling of both scale and time domain, i.e. linearly varying width for scale space and sliding for the time space. At a given time the discretization of the scale domain is similar to the nested MVA analysis (Sonnerup and Scheible, 1998) used to study the stationarity of the MVA parameters. The minimum scale, $w_{min}$, is given by three points (one point on each side of the central point). Iteratively, the scale increases by adding equal number of points (depending on $dw$ resolution) to the sides of the previous scale, yielding thus an array of odd numbers, $w_k$=3,5,7.. for the highest resolution scanning. For the Swarm high resolution magnetic field data (section 4) we look in the range between $w_{min}$=0.1 s and $w_{max}$=5 min which for an ionospheric mapping factor of 1.1 corresponds roughly to an ionospheric scale of about $\sim$760 m and $\sim$2000 km, respectively. This scheme has the advantage that one can scan all the FAC scales present in the data and provided the high resolution needed in the FAC scale/position identification (Bunescu et al., 2015). As discussed in sections 3 and 4, this high resolution linear scanning introduces a large degree of correlation in the results. Indeed, for an infinite planar sheet of width $w_0$, this is sampled many times for all scales $w_k \leq w_0$. When searching for FAC scale/location this proved to be fine, since $\partial_w \lambda_\eta$ maximized at $w_0$ for essentially east-west aligned FACs.

The second FAC scanning scheme uses successive intervals that do not overlap at a certain scale; the length of the intervals is varied logarithmicaly to provide information at different scales. This scheme is similar to the one used in Haar wavelet decomposition. All scales (interval widths) spanning $w_k = 2^k$ data points, where $k$=2,N (N the highest power of two that fits into the data interval) are considered. When dealing with large scales one can use zero padding of the data interval. Practically, in an ideal auroral oval configuration with balanced R1/R2 FACs, the largest scale samples the entire oval, in the second-largest scale the interval is split into two and addresses separately R1/R2 regions. The segmentation of the data interval repeats down to the smallest scale $w_{min}$. For the case of Swarm events (section 4) we take $w_{min}$ =0.04 s (2 points)

and $w_{max}$=21.8 min corresponding to a total number of 16 decomposition levels. One sensitive point of this scheme is the centering of the data interval because in reality we do not have an ideal oval, e.g. one can have a tangential crossing through the oval. One can manually center the analysis interval on the border between R1 and R2 regions. The main advantage of this logarithmic scheme is that is much faster than linear scheme and provides a more intuitive understanding of the multiscale

FAC density. In each computation cell of width $\Delta\xi^{(k)}$ we have the current density $j_{\parallel}^{(k)} = \mu_0^{-1}\Delta B_{\eta}^{(k)}/\Delta\xi^{(k)}$ and the integrated current $J_{\parallel}^{(k)} = \mu_0^{-1}\Delta B_{\eta}^{(k)}$. The FAC density $j_{\parallel}^{(k)}$ reflects the slope of $B_{\eta}$ whereas $J_{\parallel}^{(k)}$ reflects the jump of $B_{\eta}$ over the respective scale, $w_k$. Both $j_{\parallel}^{(k)}$ and $J_{\parallel}^{(k)}$ offer complementary useful information. In the following we concentrate on $j_{\parallel}^{(k)}$, similar to linear sampling scheme.

As it is constructed, the multiscale FAC density provides estimates of the average FAC across scales, as well as an indication

of the dominant scales, given by peaks in $\partial_{\xi}\lambda_{\eta}$. Both scale sampling schemes rely on a non-orthogonal basis functions because the aim is to precisely infer the scale and location of the FAC as well as the respective current density. As a consequence, one cannot simply integrate over scales to obtain a global FAC density estimate that can be compared with the single- and dual-spacecraft FAC estimates - which provide convoluted information about the FAC scales larger than the discretization interval (single-spacecraft) or the virtual quad scale (dual-spacecraft). As compared to the orthogonal decompositions, e.g. orthogonal

wavelet decomposition, where the signal is recovered easily by integration over scales, in our case such an integration would require a proper weighting scheme of the multiscale information. This development is considered for a future study.

## 3   Synthetic FAC structures

In this section we apply the multiscale FAC density technique to synthetic structures consisting of superposed FAC activity. We define complex FAC structures by superposing FACs of different scales (thickness), amplitudes (FAC intensity), and directions

of the current flow (upward and downward). Additionally, we consider the orientation of the FAC structures in the plane perpendicular to $\boldsymbol{B}$. The total FAC density in the $(\xi,\eta)$ frame is given by:

$$j_{\parallel}(\xi) = \sum_{k} s^{(k)} j_{\parallel}^{(k)}(\xi, \sigma_{\perp}^{(k)}) \tag{7}$$

where $j_{\parallel}^{(k)}$ denotes the elementary current associated with a single FAC element; $s^{(k)}$ is the sign of the FAC element, -/+ for the upward/downward FACs. For the case of uniform FAC density structures $j_{\parallel}^{(k)}$ =const; $j_{\parallel}^{(k)}$ is parametrized below by thickness, position, intensity, and orientation.

In the following, we define $j_{\parallel}^{(k)}$ elements according to a nonuniform FAC density depending on $\xi$ by a Gaussian function in the $(\xi,\eta)$ frame.

$$j_{\parallel}^{(k)}(\xi, J_0^{(k)}, \sigma_{\perp}^{(k)}) = \frac{J_0^{(k)}}{\sigma_{\perp}^{(k)}\sqrt{2\pi}} e^{-(\xi)^2/\left(2\left(\sigma_{\perp}^{(k)}\right)^2\right)} \tag{8}$$

The parameter $J_0$ indicates the integrated sheet current (integral across the arc per unit of east-west length) of a FAC element; $\sigma_{\perp}$ is the standard deviation and controls the perpendicular scale of the FAC element. The Gaussian profile is consistent with the FAC structures observed in the auroral region. Studies on the FAC scales (Johansson et al., 2007; Karlsson and

Marklund, 1996) estimated the FAC density profile by a Gaussian function and the scale is approximated by the full-width-at-half-maximum (fwhm) estimate, fwhm=$2\sqrt{2\ln(2)}\sigma_\perp \approx 2.35\sigma_\perp$. The fwhm estimate is typically used also when estimating the auroral thickness from optical emissions intensity (Partamies et al., 2010). In section 3.1 and 3.2 we compare fwhm FAC thickness also with $w_{1\sigma} = 2\sigma_\perp$.

Equations (7) and (8) do not include the orientation since the FACs are defined in the ($\xi$,$\eta$) frame. By using the coordinate transformation (rotation and translation) to ($x$,$y$) defined as, $\xi = (x - x_0)\cos(\theta)$, we introduce $x_0^{(k)}$ and $\theta^{(k)}$ parameters which control the location and orientation of the FAC elements. Note that the relevant angle $\theta^{(k)}$ is made by the satellite trajectory with the direction normal to the current sheet. For simplicity, we consider here that the satellite trajectory coincides with the x axis (pointing North), therefore the angle $\theta^{(k)}$ is provided directly by MVA (otherwise, one should subtract the angle made by

the satellite trajectory with the $x$ axis).

$$
j_\parallel^{(k)}(x, J_0^{(k)}, \sigma_\perp^{(k)}, x_0^{(k)}, \theta^{(k)}) =
$$
$$
\frac{J_0^{(k)}}{\sigma_\perp^{(k)}\sqrt{2\pi}} e^{-\left((x-x_0^{(k)})\cos(\theta^{(k)})\right)^2 / \left(2(\sigma_\perp^{(k)})^2\right)} \tag{9}
$$

where the FAC density of each FAC element depends on a set of four parameters ($x_0$, $J_0$, $\sigma_\perp$, $\theta$).

The integration of the Ampere law (equation (2)) yields the magnetic field associated to the FAC density (equation (7)) given by $B_\eta = \mu_0 \int j_\parallel d\xi$. Considering the superposition of FACs (equation (7)), this yields $B_\eta = \sum_k B_\eta^{(k)}$, where $B_\eta^{(k)}$ is the magnetic field of the $k$ FAC element derived as the integral of the Gaussian function and expressed in terms of error function:

$$
B_\eta^{(k)} = \frac{\mu_0 J_0^{(k)}}{2} \, erf\left(\frac{\xi}{\sigma_\perp^{(k)}\sqrt{2}}\right) \equiv
$$
$$
\frac{\mu_0 J_0^{(k)}}{2} \, erf\left(\frac{(x - x_0^{(k)})\cos(\theta^{(k)})}{\sigma_\perp^{(k)}\sqrt{2}}\right) \tag{10}
$$

where the second and third term show the dependence in the ($\xi$,$\eta$) and ($x$,$y$) frame, respectively. In order to obtain the $B_x$ and $B_y$ components we rotate $B_\eta^{(k)}$ for each FAC element with the $\theta^{(k)}$ angle ($\sphericalangle(\lambda_\xi, \hat{\boldsymbol{x}}) \equiv \sphericalangle(\lambda_\eta, \hat{\boldsymbol{y}})$). A positive/negative angle indicates a tilt toward south/north. The MSMVA analysis is thus applied on the following components of $\boldsymbol{B}$:

$$
B_x = -\sum_k B_\eta^{(k)}\sin(\theta^{(k)})
$$
$$
B_y = \sum_k B_\eta^{(k)}\cos(\theta^{(k)}) \tag{11}
$$

We note that for synthetic data the magnetic field perturbation is defined as a function of the spatial coordinate, $x$, whereas for the Swarm data (section 4) as a time series. The computation of $j_\parallel$ for Swarm is done using equation (6) which includes the

amplitude correction due to the orientation. In the case of syntetic data the amplitude is also corrected, $j_\parallel = \partial_x B_\eta / \cos(\theta)$.

By using the above equations we construct two particular cases of synthetic structures. In the first case we consider a simple balanced FAC structure, consisting of an upward and a downward FAC elements of the same thickness and amplitude, but different orientation. The second case consists of superposed FACs, smaller scale FACs of different orientations are embedded

into larger FACs. We show how the multiscale FAC estimate can be used to visualize the FACs. The simple case of a pair of FACs resembles the large scale R1/R2 system, as well as, the basic cell of a multiple arc system (Gillies et al., 2014; Wu et al., 2017). In the second case, the embedded smaller scale superposition can be associated with the analysis of the auroral oval with embedded smaller scale FACs, e.g. multiple arc systems, or pulsating aurora.

## 3.1 FAC structure of balanced current

In the following we consider the current system consisting of the downward/upward (labeled FD/FU) current regions. The value of the thickness parameter, $\sigma_\perp$, for both FAC structures is 50 km. Typical values of $\Delta B$ for the auroral region are in the range of a few 100 nT. Each 100 nT in the measured $\Delta B$ corresponds to an integrated sheet current $J_0 \sim 0.1$ A/m. For this synthetic case we consider $J_0 = \pm 0.63$ A/m for the downward/upward current. The current elements are located at $x_0^{(1)}$=600 km and $x_0^{(2)}$=800 km. We introduce a variation of the orientation from $\theta^{(1)} = 0°$ at FD to $\theta^{(2)} = 40°$ at FU. According to observations (Gillies et al., 2014) the value of $\theta^{(2)} = 40°$ is a rather extreme case for a stable auroral arc.

Figure 1 shows the results of both linear and logarithmic scale sampling for this simple FAC structure. Panel (a) shows the input current density, $j_\parallel$, of FD (magenta), FU (blue), and the total current (black). Panel (b) shows the $B_x$ (blue) and $B_y$ (green) components of the obtained magnetic field (equation (11)). This magnetic field contains a superposed normal distributed noise signal with zero mean and sigma of 3 nT. The maximum FAC density at the center of the two structures is $\sim 5\mu$A/m$^2$. The results of linear MSMVA scanning of the FAC system are shown in panels (c), (d), (e), and (f) by the planarity $R_\lambda$, the derivative $\partial_\xi \lambda_\eta$, the orientation $\theta$, and the linear multiscale FAC density, respectively. The width array used in the linear MSMVA is between 1 km and $\sim$400 km with a step of $\sim$0.6 km. We note the smooth variation of all quantities specific to this sampling scheme. On each spectrum we indicate the position and scale or the input FACs by the black circles (diameter equal to $\sigma_\perp$). $\partial_\xi \lambda_\eta$ correctly identifies the scale of FD around fwhm=117 km. For FU we get a larger estimate because of the dependence of $\partial_\xi \lambda_\eta$ on the non-invariant $w$ variable (length along the track). The sections at the FAC centers shown below are represented as a function of the corrected scale, obtained by projection of the scale array on $\hat{\xi}$ using the orientation (section (2.3)). $\theta$ scalogram (panel (e)) correctly identifies the orientation, $\theta^{(1)} = 0°$ and $\theta^{(2)} = 40°$. We note that $R_\lambda$ shows a signature with a rather flat maximum extending to large scales, with the local maxima for FD/FU regions not coincident with $\partial_\xi \lambda_\eta$ maxima. This behavior is influenced by the smoothness of $\Delta B$ for each FAC and by the constant $\Delta B$ located before/after FD/FU FACs. The multiscale FAC density shows higher values at smaller scales, roughly up to the actual scale of the structure, and decreasing values above this scale - indicated by $\partial_\xi \lambda_\eta$. This can be understood by considering the simple example of a uniform current sheet: the current density remains constant for all the scales smaller than the sheet width and decreases asymptotically to zero for larger scales. The panel on current density provides scale information just qualitatively. The quantitative aspect is addressed in correlation with the corrected $\partial_\xi \lambda_\eta$ information shown in sections at specific times (see right panels of Figure 1).

Panels (g)-(l) show the results of logarithmic FAC scale scanning. For this case the analysis is centered in the middle of the FAC structure, indicated by the vertical black line. The sampled scale array covers 13 logarithmic levels from $w_{\min}$=0.2 km to $w_{\max}$=820 km. The logarithmic scheme shows a more discrete character due to the non-overlapping sampling intervals at

each scale. Qualitatively, we observe a good agreement with the linear scheme for the orientation (panel (k)) and FAC density (panel (l)). The multiscale FAC density (panel (l)) shows at the largest scale a close to zero current because the two structures have similar amplitudes and compensate each other. At around 100 km we observe the separation of the two branches of the current centered at 600 km and 800 km. The distinction between the two regions is very clear down to smaller scales of a few km. Higher FAC intensity is observed around the centers of the FACs for scales smaller than about ~50 km.

Quantitative estimates are obtained trough vertical cuts into the MSMVA scalograms shown in panels (m)-(p) of Figure 1. The black/red line shows the profiles in the center of FD/FU structures, whereas the solid/dashed lines indicate the results for the linear/logarithmic sampling scheme. The vertical dashed lines show the scales $w_{1\sigma}$=100 km and fwhm=117 km. As discussed in section (2.3), for all multiscale parameters we correct the scale variable (multiplication of the scale array by $cos(\theta)$) to get the dependence on the perpendicular scale. For both FACs $\partial_\xi \lambda_\eta$ (panel (n)) shows that the scale is more consistent with fwhm estimate. We notice that for this simple FAC system both the linear and logarithmic sampling scheme provide consistent results, the scale is precisely identified in both cases. The orientation of the two FACs (panel (o)) at fwhm scale is consistent with the input parameters, $0°$ and $40°$ for FD and FU, respectively. We note that the scale corrected $\partial_\xi \lambda_\eta$ does not depend on the FAC's orientation. The similarity of $\partial_\xi \lambda_\eta$ amplitudes for FD and FU indicates a good amplitude correction for FU structure. $R_\lambda$ profile (panel (m)) does not have a maximum at the same scale as $\partial_\xi \lambda_\eta$. This shift is dependent on the noise level since $R_\lambda$ contains also dependence on $\lambda_\xi$. The local FAC density (panel (p)) at FD and FU locations provides also quantitative indication about the FAC scale. Around the FAC scale we observe a slight change of the slope of $j_\parallel$ for the linear scheme and also a decrease for the logarithmic scheme. At a given FAC center $j_\parallel$ shows a rather constant plateau and starts to decrease when the scanning reaches its characteristic scale. This behavior is more evident for uniform FAC density structures (see section (4.2)). The FAC density for FD and FU FACs shows values of about $\pm 4.5 \mu A/m^2$ and $\sim \pm 3.5 \mu A/m^2$ for the linear and logarithmic sampling, respectively, i.e., 10% and 30% smaller than the input FAC density ( $5\mu A/m^2$).

## 3.2 Superposition of Gaussian FAC structures

We start again with a large scale current system similar to the previous synthetic case. Two FAC elements FD/FU with $\sigma_\perp^l$=50 km and $J_0^l = \pm 0.63$ A/m are placed at $x_0$=700 km and $x_0$=900 km. The orientation of FD and FU structures is $\theta_l^{(1)} = 0°$ and $\theta_l^{(2)} = 40°$. A number of three small-scale FACs are superposed on each large-scale FAC structure. We consider equal scales of the embedded FACs given by $\sigma_\perp^s$=5 km. The small scale FACs embedded into FD have $J_0^{(k)}$ parameters defined as $J_0^l/6$, $J_0^l/3$, $J_0^l/6$, alternatively positive and negative. Similarly the small scale FACs superposed onto FU have also a central more intense FAC of amplitude $J_0^l/4$ and two side FACs of intensities $J_0^l/8$. For simplicity, we consider that all small scale FACs have $\theta_s^{(k)}$=0°. The small-scale FACs introduce alternatively positive and negative amplitude changes of the current density of the large-scale FAC system.

Figure 2 shows the overall contribution of the two scales to a rather complex FAC density profile shown by the black line in panel (a) and the corresponding magnetic field perturbation in panel (b). The FAC elements are indicated in panel (a) with blue/magenta for the pozitive/negative FAC densities at both scales. The attenuation (compensation)/intensification (addition) of the local FAC density from the two FAC systems is reflected in slower/steeper gradients of $\Delta B$. We note that the superpo-

sition of scales (equation 11) affects the orientation and the scale information for both large- and small-scale FACs. Thus, in general we do not expect to find the exact input angles and scales. The superposed normal distributed noise signal has $\sigma=2$ nT in this case. In this example we perform the linear FAC scanning over the range between 1 km and 400 km, whereas the logarithmic scanning over the scale domain from 0.2 km up to 820 km. The total number of levels in the logarithmic scanning is $k=13$.

Panels (c)-(f) and (i)-(l) show the MSMVA decomposition into the linear and logarithmic scheme, respectively. We notice the same characteristics of the two schemes, namely smooth and coarse results in the linear and logarithmic scanning, respectively. Panel (c) shows a high decrease of the planarity level for FU as compared with the previous case (section 3.1). We note regions of high $R_\lambda$ at both large- and small-scale FAC systems. The relative combination of angles and amplitudes of $\boldsymbol{B}$ from the two scales leads to three signatures of high $R_\lambda$ for the small scale FAC system inside FD, whereas for FU $R_\lambda$ is high only for the central more intense small scale FAC. $\partial_\xi\lambda_\eta$ scalogram clearly shows the two FAC systems. Besides the signatures around expected scales we have also an intermediate false level of identified FACs caused by the combination of adjacent small-scale FAC elements. In the logarithmic scanning, $R_\lambda$ (panel (i)) and $\partial_\xi\lambda_\eta$ (panel (j)) provide consistent information with the linear scanning. The $\theta$ scalograms (panels (e) and (k)) show well the overall structure of the FAC system, with values consistent with $\theta_l^{(1)}=0°$ and $\theta_l^{(2)}=40°$ for the large scale FAC system. At small scales, the variations are related to the vector addition (equation 11). While the small scale FACs inside FD show consistency with the input, $\theta=0°$, for the FU region we have good agreement with the input only for the central small scale FAC, associated with a steeper gradient in $\boldsymbol{B}$. In the regions of FAC attenuation (weaker gradient) the angles are not consistent with the input orientations, in agreement with the weeker signatures in $R_\lambda$ and $\partial_\xi\lambda_\eta$.

The local FAC density scalogram in both scanning schemes (panels (f) and (l)) provides a consistent view of the input FAC density, with well delimited FAC elements of both the large- and small-scale FAC systems. In panels (m)-(p) we show vertical cuts through the scalograms at the centers of attenuation/intensification of the FD/FU FAC density by the superposition of the two scales, indicated by vertical dashed lines in panels (a)-(l). The profiles show a more complex situation with respect to the previous synthetic case. The input scales of the two FAC systems are indicated by the vertical black (large scales) and blue (small scales) lines at $w_{1\sigma}$ and fwhm. We observe a good correlation of $R_\lambda$ and $\partial_\xi\lambda_\eta$ maxima for the small scale FACs. $\partial_\xi\lambda_\eta$ shows well defined peaks for the small scale FACs consistent with the input scales, whereas for the large scale FACs rather broad maxima, also around the expected scales. The orientations are roughly consistent with the input setup, $\theta_l^{(1)}=0°$ for FD and $\theta_l^{(2)}=\sim37°$-$42°$ for FU. At small scales we have as well consistency, $\theta_s^{(1)}=0°$ and $\theta_s^{(2)}=\sim5°$ for the small scale FACs inside FD and FU, respectively. The local FAC density for FD/FU is $4\mu\text{A/m}^2$/$-4.5\mu\text{A/m}^2$, in good agreement with the input of $\pm5\mu\text{A/m}^2$. For the small scale FACs centered on FD/FU we have $-10\mu\text{A/m}^2$/$-15\mu\text{A/m}^2$, which is roughly consistent with the input FAC density of $\sim-16\mu\text{A/m}^2$/$-12\mu\text{A/m}^2$. We get higher/lower deviations for the small scale FACs centered in FD/FU, in agreement with their weaker/stronger signatures in $\partial_\xi\lambda_\eta$.

In the case of superposed FACs the signatures of both large- and small-scale FACs are qualitatively reflected by the MSMVA information. The results also show some limitations of the method. One cannot expect to find a perfect decomposition of the FAC system, because of: a) The use of piece-wise linear functions of a certain length (scale) with a corresponding FAC density

profile given by a step function, which is not fully suitable for the smooth Gaussian functions. b) The results are actually dependent on the relative parameters (e.g. intensities, orientations, scales, locations) of the superposed FAC elements.

The combined use of $R_\lambda$, $\partial_\xi \lambda_\eta$, $\theta$, and $j_\parallel$ scalograms allows the identification of the geometry, scales, orientations, and estimates of the local FAC densities present at the respective scales. The linear approach shows a high precision in the identi-
fication of both FAC scale (panel (d)) and local FAC density (panel (f)). The logarithmic scheme lacks resolution in the FAC scale identification and subsequently gives a poor estimate of the local current. However, this scheme provides quick results that capture qualitatively similar features. More advanced data processing can include, e.g., filtering $\partial_\xi \lambda_\eta$ by the planarity $R_\lambda$, to remove non-planar FAC structures, and applying a similar mask to current density.

A more systematic study of superposed FAC sheets is required, e.g. by varying the relative parameters of a FAC system
consisting of broad and narrow FAC sheets. In this context, we note that a better approach might be to iteratively identify the FACs based on their intensity and to apply MSMVA to the successive residuals obtained by separating the identified FAC signatures (fitting the data at each iteration by model FAC functions, e.g. planar FACs, as indicated by the MSMVA parameters). However, the problem might not be uniquely determined, and before engaging in such a development, we rather apply the present procedure to several real events, three of which are detailed in the next Section.

# 4  Auroral region crossings by Swarm

The FAC density scalogram introduced in Sect. 2 and the other components of the multiscale FAC analyzer framework are now applied to three auroral crossings of the Swarm satellites, namely, a stable linear east-west aligned current sheet, an auroral pattern with sharp changes in inclination, and small-scale auroral structures embedded in a large-scale current.

## 4.1  Instrumentation and basic data processing

The Swarm mission (Friis-Christensen et al., 2008; Olsen et al., 2013) consists of three spacecraft equipped with identical instruments and placed on polar orbits. The primary objective of the Swarm mission is to study the Earth magnetic field, e.g. mapping, modeling, separation of the different sources of the measured field. The satellites are equipped with both a vector field magnetometer (VFM) and an absolute scalar magnetometer (ASM) (Hulot et al., 2015) which provide high accuracy and high resolution magnetic field measurements. ASM data are used mainly for the calibration of VFM.

In this work we mainly use the VFM measurements to study the FACs. Because we address the multiscale aspect of the FAC signatures and in order to have a good statistics also at smaller scales, we use the highest resolution data provided by VFM, namely the 50 Hz data (0.02 s sampling). The resolution of the data is directly related to the scale of the structures that can be resolved by MSMVA. For a minimum scale of 5 points in the MSMVA analysis we obtain an along-track scale mapped to ionosphere of about 700 m (spacecraft velocity of 7.6 km/s and linear mapping factor of ∼1.09).

One major point of the Swarm constellation is its orbital configuration. Two spacecraft, SwA and SwC, are flying side by side at 460 km altitude with a cross-track separation (longitudinal separation) of $1.4^o$ which amounts to about 50-80 km above the auroral oval. The measurements provided by these satellites are combined in the two-satellite methods to estimate the FAC

density (Ritter and Lühr, 2006; Ritter et al., 2013). The other spacecraft, SwB, is flying at higher altitude and periodically forms a close three-satellite configuration with the lower pair. When this is the case, it is possible to compute the FAC density by using also a three-spacecraft method (Vogt et al., 2009). In the following, for each event we cross-check the local FAC density provided by MSMVA with the single- and dual-spacecraft estimates.

5  The single- and dual-spacecraft FAC estimates provided by ESA (part of the Swarm L2 products available at ftp://swarm-diss.eo.esa.int/) are based on the FD approach and available with 1 s resolution. The single-spacecraft FAC density corresponds to a resolution of the mapped ionospheric scale of ∼7 km. The computation of FD dual-spacecraft FAC estimate is done with a filtered magnetic field perturbation. The filtering is used to remove the FACs with scales smaller than ∼20 s, corresponding to along-track scales smaller than ∼150 km (Lühr et al., 2016). Thus, we expect a good agreement between the single- and

10 dual-spacecraft FAC density estimates for scales larger than 150 km.

  The second type of data used in this study is provided by the THEMIS ASI ground network. THEMIS ASI network (Donovan et al., 2006; Mende et al., 2009) was installed to complement spacecraft observations, in particular by the THEMIS mission, related to substorms and, more generally, to auroral phenomena. With a number of 22 stations, the network covers a large region of northern Canada, Alaska and Greenland. The THEMIS ASI locations were chosen based on an earlier statistical study (Frey

15 et al., 2004) of the auroral substorm onsets inferred from IMAGE spacecraft. Each ASI provides frames of 256×256 pixels at a time resolution of 3 s (exposure time 1 s). All ASI are based on fish-eye lenses that provide wide angle optical observations. Due to the fish-eye lenses the pixels at the center cover a smaller sky surface element as compared to the pixels located towards the edges. Thus, the best resolution is at the center, of about 1 km. The events included in this study make use of optical data from Sanikiluaq (SNKQ), Rankin Inlet (RANK), and Fort Smith (FSMI).

20 One basic operation is the mapping of the spacecraft orbit into the image plane, done by using the THEMIS TDAS software (http://themis.ssl.berkeley.edu/) where the field line tracing is implemented by different versions of the Tsyganenko magnetic field model. In this paper we use Tsyganenko T04 model (Tsyganenko and Sitnov, 2005) with the solar wind parameters provided by OMNI (http://omniweb.gsfc.nasa.gov) and DST index from WDG at Kyoto (http://wdc.kugi.kyoto-u.ac.jp/). The footprints of Swarm are projected onto the optical frames provided by the THEMIS ground stations.

25 The measured magnetic field is transformed to the MFA reference system. The magnetic field perturbation, $\Delta B$, is obtained by subtracting a model magnetic field from the measured data. The internal magnetic field parametrization is taken from CHAOS-6 (Olsen et al., 2014; Finlay et al., 2016) whereas the lithospheric (e.g. crust and upper-most mantle) and external magnetospheric (e.g. ring current) contributions are taken from the Pomme 10 (Maus et al., 2006, 2010) model. The results obtained for various events in different geomagnetic conditions showed good consistency when using this setup. Ideally, after

30 the subtraction of the magnetic field model we should remain with the perturbation caused by the large scale R1/R2 currents, the embedded mesoscale and small-scale FACs, as well as the influence of the ionospheric current systems. Another option is to separate the embedded small-scales FACs from the large-scale FACs (R1/R2) by filtering the data. Bunescu et al. (2015) computed a model magnetic field proxy from the measured field using an average over a sliding window (with tapering at the ends). This procedure excludes roughly the scales larger than a certain percent of the sliding window width (depending on the

tapering extent). The disadvantage of this approach is that it can introduce additional low amplitude fluctuations. Thus, in the following we analyze $\Delta B$ obtained by subtracting the model magnetic field.

## 4.2 Stable east-west aligned aurora of constant FAC density

On 17 February 2015 the Swarm spacecraft crossed the auroral oval toward north over the FoV of the SNKQ station. The
event is observed around 03:25 UT at ∼1 h after an intermediate substorm intensification/onset following ∼6 h of quasi-steady magnetospheric convection. The AE index is ∼200 nT, whereas DST∼-26 nT.

    Figure 3 shows the ionospheric footprints of the spacecraft (mapped at 110 km altitude) superposed on the SNKQ optical observations. The optical frames are mapped to geographic coordinates and show rather stable and east-west elongated arc structures. We distinguish two large-scale upward FACs located northward and, respectively, southward of the station. Between
these two upward FACs we observe a mesoscale upward FAC with an east-west extent covering the westward FoV of SNKQ. Swarm is crossing along the westward edge of the ASI over all three visible arcs. While the thick northward and the narrow mesoscale structures are highly planar, the thick southward structure looks curled around the spacecraft tracks. Because Swarm is crossing near its center, the magnetic field perturbation for this strucure looks similar to that of a planar FAC.

    Figure 4 shows SNKQ keogram, Swarm $\Delta B$, FAC density estimates (L2 products), and the hodogram representation of
$\Delta B$. The keogram (panel (a)) is obtained by stacking in time the central column (meridian) of pixels from the optical frames. The combined analysis of optical frames and of SNKQ keogram confirm the stability of the aurora over the entire interval. The intermediate arc appears in the center of ASI around 03:25 UT. The measured $\Delta B_\perp$ by SwA and SwC are shown in panels (b) and (c). $\Delta B_\perp$ from both spacecraft shows similar structures, with a small difference in amplitude, consistent with optical data. $\Delta B_\perp$ from SwA shows a shift (within 10 s) with respect to SwC crossing earlier. SwB (not included) is not properly located,
its footprint is outside the ASI's FoV. The vertical dashed lines indicate distinct regions of the FAC system. The black, blue, green, and red indicate the beginning of upward FACs labeled U1, U2, U3, and U4 whereas magenta and cyan indicate the downward regions in-between, labeled D1 and D2. One observes some small imbalance between the upward and downward current, presumably caused by a cross-polar cap current system, or by the imprecision of the magnetic field model in the polar region.

Panel (d) shows different FAC density estimates. The green and red line shows the L2 single-spacecraft FAC density obtained using the un-filtered magnetic field data from SwA and SwC, respectively. The L2 single-spacecraft FAC estimate (Ritter et al., 2013) with 1 s resolution (∼7.5 km ionospheric scale) is computed with the assumption that the main magnetic perturbation is in the east-west $B_y$ component. The dual-spacecraft FAC density that combines the information provided by SwA and SwC using the FD method of Ritter et al. (2013) is indicated by the black line. This estimate is computed over the filtered data that
removes scales smaller than 150 km. The two-spacecraft method shows an average of the FAC density over the quad area and does not capture small-scale FACs. Both single and dual-spacecraft FAC estimates are used as a qualitative reference for our multiscale FAC density technique.

    Panel (f) of Figure 4 shows the hodogram representation, $B_y$ as a function of $B_x$, for SwA. The hodogram is represented with the time interval running from blue to red (rainbow color scale). On this trace we indicate the FAC segments with the

same color used in panels (a)-(e) to mark the beginning of the respective time interval. We observe different regions of the hodogram that consist of linear segments which indicate FAC structures of constant orientation (linear polarization of $\Delta B$). The U1, U2, U3, and U4 FACs are indicated by the black, blue, green, and red lines, respectively. The MSMVA is used to find and characterize such segments of linear polarization of $\Delta B$.

5 The left part of Figure 5 shows the results of the linear MSMVA for SwA. The planarity, shown by $R_\lambda$ scalogram (panel (b)), indicates regions of high planarity for several large/small-scale FACs, e.g. U1-3 and D1-2. The scalogram of $\partial_\xi \lambda_\eta$ (panel (c)) shows the location and thickness of FAC structures whereas their orientation (panel (d)) confirms the optically observed alignment of the normal with the north direction, $\theta \approx 0°$. Some typical threshold value of $R_\lambda$ associated with planar structures is about 10-30 (for 3D MVA). Because we use the 2D MVA ($B_\perp$ perturbation), $R_\lambda$ shows larger values, consistent with a reduced

10 variance. We note that the investigation of the relationship between the longitudinal extension of FACs and the $R_\lambda$ ratio can actually be done by using correlation analysis of the two longitudinally separated Swarm spacecraft. One expects that $R_\lambda$ can provide a more quantitative indication about the FAC east-west length. This topic is considered for a future study.

 Panel (e) shows the newly introduced linear multiscale FAC density (section 2). We can easily see the different regions of upward and downward current at different scales, e.g. large-scale R1/R2 FACs at scales larger than 100 s, better visible

15 in the logarithmic sampling, and smaller scale FACs (U1-3, D1-2) at lower values, better visible in the linear sampling. The negative/positive large-scale trend is associated with upward/downward FACs, consistent with the statistical FAC model (Iijima and Potemra, 1976b) around 22 MLT. An alternative identification of the large-scale FACs is done by Wu et al. (2017) directly with single-spacecraft FAC density by computing the ratio of the upward and downward current to the total current.

 These representations provide a new visualization of the FAC currents dependent on scale. The linear scanning of FACs uses

20 a large number of scales sampled at high resolution. As already mentioned, one limitation in the integrated FAC estimate for this approach is that it does not rely on an orthogonal basis and thus the integration over scales does not provide a global FAC density similar to the single- and dual-spacecraft FD methods. In order to partially improve the analysis towards an orthogonal basis we computed the same parameters also for the logarithmic scanning procedure (section 2). Panels (f)-(j) show MSMVA quantities for the logarithmic scanning. In this case, the scale range extends to higher values ($\sim$1000 s=16.6 min) and from

25 about 200 s (1381 km mapped to ionosphere) up one can see a close to zero net current. While the resolution is not suitable to obtain precise information on the scale dependence of these quantities, the results are in good qualitative agreement with the linear scanning.

 Figure 6 shows a more quantitative comparison of the MSMVA quantities, including FAC density given by the two scanning schemes. We show the scale dependence of $R_\lambda$, $\partial_\xi \lambda_\eta$, $\theta$, and $j_\parallel$ at the center of the FACs as identified by $\partial_\xi \lambda_\eta$ and indicated

30 by the solid lines in Figure 5. The selected times are $t_{U1}$=03:24:43 and $t_{D1}$=03:25:00, associated to U1 and D1, respectively. All quantities are represented as a function of the corrected scale similar to the synthetic data (section 3) and neglecting the small inclination of the Swarm trajectory with respect to the $x$ axis (direction pointing North). One can see that all quantities have local maxima around the same scale indicated by the vertical dash lines at 22 s and 10 s for U1 and D1, respectively. These scales correspond to about 153 km and 70 km in the ionosphere. $R_\lambda$ shows a high planarity at these two scales, with

35 values larger than 100 (threshold indicated by the horizontal blue line) for both FACs in the linear sampling. The logarithmic

sampling shows smaller values, with a smoothing of the linear profile and values below the threshold for D1. For the logarithmic sampling $\partial_\xi \lambda_\eta$ shows a similar scale, 16 s (110 km at ionosphere), for both U1 and D1 FACs. The orientation is consistent for both linear and logarithmic sampling, $\theta$=10° for U1 and ~2° for D1. In the case of rather uniform FAC density (U1 and D1) we observe that the maxima of $\partial_\xi \lambda_\eta$ is almost aligned with local maxima in $R_\lambda$ which is consistent with the intuitive expectation

that the planarity of a sheet like FAC structure maximizes around the scale (thickness) of the sheet.

The FAC density at U1 and D1 is around -2.7$\mu$A/m$^2$ and 4.5$\mu$A/m$^2$, respectively, for the linear sampling. In the case of logarithmic sampling, $j_\parallel$ (dashed lines in panel (d)) shows roughly similar results where the respective scales are properly sampled. We have agreement for U1 (~-2.5$\mu$A/m$^2$) and a close to zero FAC density for D1. The zero estimate of the current for D1 in the logarithmic scanning is caused by imperfect centering at that scale with respect to the linear scanning. Most likely,

it is evaluated between U1 and D1, where we have a compensation of the currents from the two FACs. The profile of $j_\parallel$ for D1 corresponds to the same scale, but it is evaluated at a different point with respect to $t_{D1}$. For the logarithmic scheme, precise comparison with the linear scheme can be obtained at the centers of the sampled intervals (panel (j) in Figure 5).

Due to the high planarity and relatively large thickness, U1 and D1 structures satisfy the assumptions of the single- and dual-spacecraft methods. The FAC density in the single-spacecraft approximation (panel (d) in Figure 4) at $t_{U1}$ and $t_{D1}$ shows

values of -2.37$\mu$A/m$^2$ and 4.02$\mu$A/m$^2$, respectively, whereas the dual-spacecraft FAC estimate indicates values of -3.21$\mu$A/m$^2$ and 2.58$\mu$A/m$^2$. These values indicate deviations of the local FAC density (linear) with respect to single-spacecraft FAC (100·$(j_\parallel^{MSMVA} - j_\parallel^{sc})/j_\parallel^{sc}$) of about 14% and 12% for U1 and D1, respectively. The same estimates with respect to the dual-spacecraft FAC density are -15% and -74%. The main characteristics of U1 and D1 FACs, including the percentage differences between the FAC density estimates (multiscale, single- and dual-spacecraft) are summarized in Table 1. The deviation of the

local FAC with respect to the dual-spacecraft FAC density is consistent with the scale information, low/high deviation for large/small-scale FACs. While U1 scale (153 km) is close to the resolution limit (150 km) of the dual-spacecraft method, the scale of D1 is below this limit. Considering the uncertainties in the scale definition and estimate of the FAC density, we consider that the differences between the local FAC density and the single-spacecraft estimate (<15%) indicate a good agreement.

**Table 1.** Comparison of the FAC estimates for 17 February 2017. Colums show: perpendicular scale, FAC inclination, FAC density from multiscale, single- and dual-spacecraft, and the relative differences between the FAC densities.

|  | $w_\perp$[km] | $\theta$[$^o$] | $j_\parallel$ (MSMVA) [$\mu$A/m$^2$] | $j_\parallel$ (1SC) [$\mu$A/m$^2$] | $j_\parallel$ (2SC) [$\mu$A/m$^2$] | MS-1SC[%] | MS-2SC[%] |
|---|---|---|---|---|---|---|---|
| U1 | 153 | 10 | -2.7 | -2.37 | -3.21 | 14 | -15 |
| D1 | 70 | 2 | 4.5 | 4.02 | 2.58 | 12 | -74 |

Through the continuous and multiscale MSMVA analysis we identify the discrete FAC elements associated to the measured

magnetic field perturbation. The sections in the MSMVA scalograms quantify how much current one has at the respective FAC structure. The results show the difficulty of dealing at the same time with a meaningful local FAC density estimate at a given scale and the need for orthogonality in the MSMVA basis functions. While FAC density is correctly inferred locally, one cannot compute a global FAC density estimate by integration over scales due to the lack of orthogonality of the basis functions. The

sections shown in Figure 6 were selected around the local maxima of $\partial_\xi \lambda_\eta$. The sharp maxima of $\partial_\xi \lambda_\eta$ for U1 and D1 agree with structures of constant current densities (Bunescu et al., 2015), also expected from the $\Delta \boldsymbol{B}$ profile. The gray shaded area in Figure 6 shows the range of scales for which $R_\lambda$ is below an arbitrary reference level of 100. This indicates the possibility to clean MSMVA quantities based on the planarity level. Such an option is needed for a multi-event or statistical study on the scale dependence of FACs characteristics. Overall, the sections into the MSMVA scalograms indicate consistent results, since all quantities show roughly the same scale. One can also note that the linear scheme is better suited for scale analysis.

A comparison of the regular single-satellite FAC density with the MVA corrected FAC density product, albeit without scale dependence, is included also in Gillies et al. (2015) for 9 events of pulsating aurora. Gillies et al. (2015) found consistent results between the two estimates at the edges of the patches associated to $R_\lambda = \lambda_{\text{int}}/\lambda_{\text{min}} > 10$ for which the infinite FAC sheet approximation was considered valid, whereas, within the patch the criteria $R_\lambda > 10$ was fulfilled only for 5 out of 9 events.

The multiscale FAC density benefits from the orientation computed at each scale. For the case of east-west aligned FACs, this may have less influence, even though one cannot exclude that some FAC elements, in a certain range of scales, are not east-west aligned. The more so, one can expect differences for events of inclined FACs. Typically, the quiet aurora during the growth phase has the normal direction aligned with the north direction. By using the multiscale approach one can check if this is true also for the embedded small-scale FACs. During the onset, expansion, or early recovery phase the aurora is typically dominated by 2D forms, possibly including locally planar small-scale FACs. By using the multiscale estimates one can better quantify the FACs with respect to their orientation as a function of scale. This might help to quantify whether the embedded FACs are forced to have the same orientation as the large-scale FACs and, further on, possible relationships between the respective mechanisms. The FAC density scalogram combined with the other information of MSMVA provides a more intuitive and visual representation that can help to search the data for particular information.

### 4.3 Inclined auroral structures

This event was observed by Swarm and RANK station of the THEMIS ASI network on 15 January 2015 around 07:39 UT. The event was observed after a long quiet period, during the growth phase of a substorm with maximum ∼1 h later and, possibly, during/after pseudo-breakup activity. The AE index is ∼70 nT and DST between -5 and -8 nT. The optical frames under the spacecraft track (07:39:27-07:39:54) are shown in Figure 7. The optical frames from the southward pass of Swarm over RANK were not included since the structures are not clearly visible. $\Delta \boldsymbol{B}$ shown below indicates locally planar FACs also in this region. Overall, the optical data show a larger scale structure inclined with respect to east-west direction (the angle between the normal to the FAC and north is about -20°). Embedded smaller scale FACs with a limited east-west extent are visible in the central region at slightly different orientations. In the center of RANK FoV SwA/SwC are crossing different structures. The two planar FACs are about parallel as shown also by the magnetic field data below. The RANK keogram (panel (a) in Figure 8) shows a patchy character related to the structuring of aurora. While not detailed here, a more consistent display of the time evolution of aurora can be obtained through the satellite-aligned keograms SAK (Gillies et al., 2015) obtained by stacking in

time the line of pixels along the spacecraft trajectory. This is particularly useful for small-scale structures, e.g. pulsating auroral patches (Gillies et al., 2015).

Figure 8 shows Swarm measurements of $\Delta\boldsymbol{B}$, FAC density estimates, and $\Delta\boldsymbol{B}$ hodogram. Consistent with the inclination of the FAC structures, we have a stronger northward $B_x$ component of $\boldsymbol{B}_\perp$ up to about 100 nT. One can expect that the calculation of the typical single-spacecraft FAC density that neglects $B_x$ component would lead to an underestimation of $j_\parallel$. $\Delta\boldsymbol{B}$ (panels (b) and (c)) indicate similar FAC structures observed by SwA and SwC. Without optical data one could think that the two spacecraft are crossing the same structures, because of the similarity of $\boldsymbol{B}$ signatures, possibly with same dynamics considering that $B_x$ component is varying. The L2 single-satellite FAC density (1 s resolution), shown in panel (d), indicates an oscillatory signature, associated with crossing a sequence of upward and downward FACs. The oscillations are also shown by the two-spacecraft FD estimate ((black line)). One can expect that the two-spacecraft estimate is rather not suitable to describe the internal structure observed optically for this event because the assumption of uniformity over the quad surface is likely not well satisfied, e.g. in the central region of RANK's FoV. The two-satellite method can average over different structures. In this respect, the scanning of FACs by using MSMVA can help to visualize and characterize the observations of geometry ($R_\lambda$) and orientation ($\theta$). For completeness, panel (e) shows the hodogram for this event. The intervals and the color code assignment is the same as for the previous event. Moving towards higher latitudes, SwA is crossing successively several upward and downward FAC segments colored by black, magenta, blue, cyan, green, yellow, red, and black in the hodogram. We label the delimited upward FACs by U1-U4 and the downward regions by D1-D4. The magenta interval shows also a substructure of three FACs. The difference with respect to the previous case is that for this event we have a more complex current system with embedded mesoscale FACs superposed mainly on the large-scale upward FAC, consistent with the optical data.

Figure 9 shows the results of the multiscale analysis for SwA. The left/right plots show the comparison of linear/logarithmic scanning schemes. $R_\lambda$ (panel (b)) shows high values for some of the mesoscale FAC structures in the southern part of RANK location, not well visible optically. Higher values are also associated with the crossing of the FAC system in the center of FoV. By comparing $R_\lambda$ values with the previous event we observe a decrease of planarity level by half, consistent with the sub-structuring of aurora, finite east-west aligned FACs. We observe also the alternance of high and low planarity regions, well correlated with regions of upward and downward current, respectively, in the mesoscale range. High planarity at small-scale FACs is embedded also in the downward current regions. The scalogram of $\partial_\xi \lambda_\eta$ (panel (c)) shows high intensity for U4 and D4 regions. The scale associated with U4 and D4 region is around 10 s (70 km). The orientation (panels (d) and (i)) at these scales is $\sim$-20° and $\sim$0°, respectively, qualitatively consistent with the optical data. The $j_\parallel$ scalograms (panels (e) and (j)) show well the embedded regions of upward and downward directed current. One can zoom into this display to get information at smaller scales, e.g. the region adjacent to the equatorward part of the track.

Similar to the previous event, in Figure 10 we show sections into MSMVA scalograms to infer quantitative estimates of the scales and current densities for a few selected FAC elements. The times of the sections are 07:38:41 (blue), 07:39:09 (green), and 07:39:33 (red). These times, indicated by the solid lines in Figure 9, are all located in upward current regions, U2, U3, U4 intervals. As before, for all profiles we show the dependence on the corrected scale (taking into account the inclination). $R_\lambda$ shows values larger than 100 for all selected upward FACs. The maxima of $\partial_\xi \lambda_\eta$ at larger scales correspond to remote FAC

elements, e.g. U4 and D4 (see Figure 9). We note the slight shift between $R_\lambda$ local maxima and the maxima of $\partial_\xi \lambda_\eta$ and $j_\parallel$. We have a good agreement between the linear and logarithmic sampling for the identification of the scale for U2 and U4, whereas U3 is not properly sampled by the logarithmic scheme. We note ~~a similar~~ scales of $\sim$12–14 s (84–98 km ionospheric scale) for ~~all the~~ three FACs. The scale dependence at these sections shows again clearly that a masking procedure based on $R_\lambda$ would

be effective in removing the features associated with remote FACs crossed earlier or later. The orientation (panel (c)) shows an inclination of about -~~20~~ 18° for U2 (blue), $\sim$~~0~~ 4.5° for U3 (green) and -25° for U4 (red), with roughly similar values in the two sampling schemes and consistent with the optical data.

The values of the FAC density at these FAC segments are about -0.6$\mu$A/m$^2$ for U2, -~~0.7~~ 0.67$\mu$A/m$^2$ for U3, and -~~1.5~~ 1.49$\mu$A/m$^2$ for U4 in the linear sampling. Roughly similar currents are obtained in the logarithmic sampling. The FAC density

given by the single-spacecraft L2 estimate (Figure 8) for U2, U3, and U4 is -0.23$\mu$A/m$^2$, -0.58$\mu$A/m$^2$, and -0.66$\mu$A/m$^2$, whereas the dual-spacecraft FAC estimate is -0.35$\mu$A/m$^2$, -0.52$\mu$A/m$^2$, and -0.87$\mu$A/m$^2$, respectively. For U2, U3, and U4 we have deviations of the local multiscale FAC density of ~~160~~ 161%, ~~20~~ 15%, and ~~127~~ 126% with respect to single-spacecraft L2 estimate.

MSMVA FAC estimates, selected based on $\partial_\xi \lambda_\eta$, correspond to the overall (average) current at the mesoscale U2-U4 FACs.

When simply compared with the instantaneous values of the L2 single-sc FAC density, the differences are significant (e.g. U2 and U4) due to the mismatch of the compared scales. In order to properly compare the FAC estimates they should correspond to similar scales. Thus, we computed also an estimate of the current at a scale similar to the U2 and U4 thickness by simply smoothing the L2 single-sc current using a boxcar running average of 12 s width (see Table 2). The comparison of MSMVA FAC density with the average L2 single-sc current leads to a decrease of the relative percentage differences to about $\sim$8-46% for

the selected FACs and thus confirms that the initially larger differences are due to the comparison of mesoscale FAC currents with currents associated to the internal structure of the respective FACs. The small difference between the local multiscale FAC and the single-spacecraft estimate for U3 is consistent with the east-west alignment of this structure ($\theta < 5°$). The ~~large~~ remaining deviations for U2 and U4 are ~~partially~~ presumably related as well to the neglect~~ing~~ of the orientation, contribution from $B_x$, in the computation of the single-spacecraft FD FAC estimate. A further inclusion of the orientation in the average L2

single-sc FAC product would probably make the agreement with the MSMVA result even better.

Comparison of the local multiscale FAC density with the dual-spacecraft estimate gives 71%, ~~34~~ 29%, and ~~72~~ 71% for the three FACs. Thus, we have again a lower difference for U3 and higher for U2 and U4. The percentages for U2 and U4 are still smaller than when comparing with the single-spacecraft estimates. Part of the differences is probably related to the resolution limit of the dual-spacecraft FAC estimates, larger than our scale of $\sim$83 km. The comparison between the FAC density estimates

is summarized in Table 2.

Both the linear and logarithmic sampling provide consistent information. We have similar results for the orientation and the local FAC density, whereas the scale identification can sometimes be missed in the logarithmic sampling (e.g. U3) due to the limitations of this scanning by non-overlapping intervals. This event indicates that care is needed when designing an automatic procedure for the analysis of FACs on a statistical basis. The two-spacecraft methods can average over different structures,

moreover some assumptions of the methods are possibly not fulfilled.

**Table 2.** Comparison of the FAC estimates for 15 January 2015 (same format as Table 1). This table shows additionally the average L2 single-sc FAC density (at 12 s scale) and the percentage deviation obtained by comparison with the MSMVA estimate. All FAC density estimates are given in $\mu$A/m$^2$, similar to Table 1

|  | $w_\perp$ [km] | $\theta[^o]$ | $j_\parallel$ (MS) | $j_\parallel$ (1SC) | $j_\parallel$ (1SC AVG) | $j_\parallel$ (2SC) | MS-1SC [%] | MS-1SC AVG [%] | MS-2SC [%] |
|---|---|---|---|---|---|---|---|---|---|
| U2 | ∼84 | -18 | -0.6 | -0.23 | -0.49 | -0.35 | 161 | 22 | 71 |
| U3 | ∼98 | 4.5 | -0.67 | -0.58 | -0.62 | -0.52 | 15 | 8 | 29 |
| U4 | ∼84 | -25 | -1.49 | -0.66 | -1.02 | -0.87 | 126 | 46 | 71 |

## 4.4 Small scale auroral observations embedded into a large scale current

The relation between multiple arc systems and their FAC signatures was addressed recently by Wu et al. (2017) based on Swarm/THEMIS ASI observations. Wu et al. (2017) selected events with clearly identifiable stable arcs and separated the observations into two categories, unipolar (multiple arcs embedded into a single large upward FAC) and multipolar events (a collection of multiple arcs and related pairs of upward and downward FACs). Arcs associated with multipolar FAC events were found to be broader and more separated than those associated with unipolar FAC events. In this section we perform MSMVA analysis for an unipolar event investigated by Wu et al. (2017).

The event occurred on 27 September 2014 around 06:00 UT, in the evening sector (∼22 MLT) and was observed simultaneously by Swarm and FSMI ASI in Canada. The event was observed during a very active period, with multiple substorms and an average AE of ∼500 nT over the hours around the event. The AE index is ∼550 nT and DST=-23 nT.

The mapped optical frames and the superposed spacecraft tracks, shown in Figure 11, indicate the crossing towards equator of a thick auroral structure (∼05:59:48-06:00 =12 s) followed by some small-scale less intense arcs and an intense structure around 06:00:09-06:00:12. Since the crossing is near the edge of the ASI FoV, in the following we do not attempt to make a one to one matching between the optical observations and $\Delta B$ or FAC signatures.

The MLT location of the event and the optical data indicate the crossing near the Harang discontinuity region. Following Swarm track (north to south) the statistical model of FACs (Iijima and Potemra, 1976b) indicates the crossing of the large-scale downward, upward, and downward FACs. Figure 12 shows again $\Delta B$, FAC density, and $\Delta B$ hodogram. $\Delta B$ for both SwA and SwC (panels (b) and (c)) show the three large scale FACs with embedded smaller scale structures. The FAC density estimates are shown in panel (d). We note the high fluctuation level in the single-spacecraft estimates caused by small-scale FACs. The two spacecraft method shows likewise the large-scale FACs and does not capture small-scale currents, which are dynamic, since the related signatures on the two spacecraft, separated by ∼10 s in latitude, are different. The small-scale features are captured by the single-satellite FAC estimates, but it is difficult to quantify their characteristics based only on this information. Comparison of $\Delta B$ from SwA and SwC shows more clearly small-scale perturbations on SwC in the green interval. Thus, in the following we perform MSMVA analysis on SwC.

The hodogram (panel (e)) shows the typical characteristics observed for the previous events. The interval color is given by the color of the left vertical dashed line which for this case are related to $\Delta \boldsymbol{B}$ from SwC. The prevalence of $\Delta B_y$ indicates a close alignment of the arcs with the east-west direction. We note that in this case the relationship between the FAC direction and the slope of $\Delta \boldsymbol{B}$ is opposite with respect to the other events, consistent with the eqatorward crossing of Swarm. The U/D labels are associated to positive/negative slope of $\Delta \boldsymbol{B}$ and negative/positive $j_\parallel$ (see section 2.2, equations (4) and (5)). Embedded small-scale FACs segments are seen in the red (D2) and the second green (D3) intervals which show as well a rotation in the hodogram, specific to wave activity.

Figure 13 shows the results of the linear and logarithmic FAC scanning. The intensity of the scalograms for the linear scheme is also shown in logarithmic scale to emphasis the small-scale FACs. The highly planar FACs at small-scales are confirmed by $R_\lambda$ scalogram (panels (b) and (g)), consistent with the hodogram. The general description from the previous events applies also here. In the following, we select and analyze in more detail a few small-scale FACs, indicated by the solid vertical lines. The black/green color indicates downward/upward FACs. Here we do not distinguish between these small-scale downward FACs and just infer a range of the parameters. Figure 14 shows the sections in the MSMVA scalograms at the respective times. We observe again that remote FACs have a smaller impact on $R_\lambda$ as compared to $\partial_\xi \lambda_\eta$. All selected small-scale FACs show a high degree of planarity. The dependence of $\partial_\xi \lambda_\eta$ on scale indicates a range of scales between 1.8 s (12.4 km) and 4 s (27.6 km) (shown by the vertical red lines) for the selected FACs. The orientation shows values from $\sim$-40$°$ to $\sim$10$°$. The FAC density has values of about -7.5 $\mu$A/m$^2$ for the upward region (green) and between 4 $\mu$A/m$^2$ and 6 $\mu$A/m$^2$ for the four downward FACs. For this event we can make just a qualitative comparison with the single-spacecraft FAC estimates since the scales are well below the resolution of the dual-spacecraft estimate. The single-spacecraft FAC density is -4.24 $\mu$A/m$^2$ for the upward FAC and between 3.84 $\mu$A/m$^2$ and 6.08 $\mu$A/m$^2$ for the downward FACs. Thus, we have roughly similar values for downward FACs with small inclination (<10$°$) and higher deviations for the highly inclined FACs, e.g. upward FAC element. The detailed analysis (not shown) indicates that the selected times are associated with local maxima of the single-spacecraft FAC density and this indicates the consistency of $\partial_\xi \lambda_\eta$ information at small scales.

When going to smaller scales, non-stationary effects become more important and can be characterized by using the nested MVA analysis. This procedure is implicitly included in the MSMVA technique since at each point we perform a nested MVA in the linear scanning. The standard nested MVA (Sonnerup and Scheible, 1998) is applied in 3D and investigates the scale dependence of orientation and projections of $\boldsymbol{B}$ on the eigenvectors. In this study, we extended MSMVA analysis by the density scalogram and showed quantitative estimates of the local current density for FACs observed by Swarm.

## 5   Discussion and summary

A good fraction of the FAC signatures above the auroral oval consists of rapidly varying FAC features, associated with time dependent discrete aurora, superposed on slowly varying FAC structures (R1 and R2 currents). Using a fixed window analysis approach to study the FACs which occur at different scales has limitations. Instead, one can use varying window sizes to capture both the fast and slow varying FACs. The long/short analysis windows are appropriate for large/small-scale FACs.

The MSMVA technique was previously applied to auroral oval crossings by Cluster and FAST spacecraft. The main goal of Bunescu et al. (2015) was to introduce the technique for the scale identification capability by $\partial_w \lambda_\eta$. Bunescu et al. (2015) showed large-scale planar and stable arcs as well as more dynamic aurora (locally planar), but did not address in detail the superposition of scales or the inclined FACs. The magnetic field was filtered such that the large scale R1 and R2 FACs were practically removed. Thus, the method was effectively showing the sequence of crossings of mesoscale FACs, whereas the small scale FACs were not analyzed in detail in terms of localization and orientation.

In order to explore all scales, the analysis was previously applied using a linear scale sampling, covered typically with high resolution. While some small scale structures are seen, as they should, in their scale range, they also contribute to the variance at large scales. One large scale is identified as planar at any scale smaller than its thickness. By using this method one can self-consistently derive various information on the planarity, scale, and also on the current density, which is particularly useful when optical data are not available in event studies.

In this study we compared the local multiscale FAC density estimates with well established methods used routinely for the computation of the FAC density. The goal was to show that the multiscale FAC density provided results consistent with other methods, in particular Ritter et al. (2013). For the case of synthetic FACs (section 3) the comparison of the input parameters with the local output of the MSMVA parameters indicated specific limitations of the method (related, e.g., to the accuracy of resolving different scales and the respective orientations), to be explored closer by upcoming work.

The analysis presented in this work offers a new visualization tool for the FAC density that helps to explore current structures embedded into larger scale FACs. The main goal of the paper is to enable the visualization of the multiscale FAC density. Based on this framework we can easily visualize the discrete constituents of a measured FAC signature. $R_\lambda$ dependence on scale in the center of FAC structures allows to separate the instantaneously crossed FACs from remote FACs. Thus we can separate the near field FACs from the far field FACs. The accuracy of the identification depends on the relative distance between the FACs and their planarity. The complex FAC signatures can be thus deconvoluted into a discrete sequence of FAC elements.

The extended MSMVA framework, and the FAC density scalogram in particular, can be compared with other spectral techniques offering spectral resolution together with time localization. The most prominent examples are the dynamical Fourier spectra produced by a windowed Fourier transformation (WFT), and wavelet techniques. In a so-called orthogonal or discrete wavelet transform (DWT) such as the Haar transform or the Daubechies transform (Daubechies, 1992), the signal is represented using a family of mutually uncorrelated (hence orthogonal with respect to the canonical scalar product) basis functions. The orthogonality condition facilitates signal reconstruction but puts severe constraints on the selection of scale and time parameters that are then usually arranged in a manner similar to the logarithmic scale sampling scheme chosen for the FAC density scalogram (subsection 2.4). A so-called continuous wavelet transform (CWT) does not aim at a compact signal representation and hence can be based on a function family that is not constrained by orthogonality conditions. A CWT produces a (redundant) set of signal correlations with basis functions that depend on scale and time, e.g. Gaussian wave packets in the Morlet wavelet transform. In contrast to DWTs, the flexible choice of time and scale parameters in CWTs allows for a smooth representation of the time-varying scale dependence of the signal.

The FAC density scalogram of the extended MSMVA framework introduced in this paper takes into account scale-dependent current structure information such as sheet inclination, reflected in both perpendicular components of the magnetic field perturbation $\Delta B$, and thus yields a more comprehensive FAC representation than straightforward wavelet transforms. Selected elements of wavelet transform are adopted in our constructions of FAC density scalograms, e.g., logarithmic scale sampling,

and the construction of time series segments for multiscale MVA. At the largest scale we sample the entire auroral oval, a separate sampling of R1 and R2 at the second scale and then a progressive decrease of the analysis window appropriate for small-scale FACs, without overlapping segments. The logarithmic scheme is faster and consumes less computational resources but the analysis intervals cannot be expected to properly capture the location and extent of FAC structures.

The model functions implicitly employed to represent the magnetic field measurements are piece-wise linear functions of

a certain length $w$, interpreted as the scale of the underlying current structure. The corresponding FAC density profile is a step function of the same width $w$, and centered at the same reference time $t_{cen}$. This approach is compatible with established FAC estimators based on finite differencing. Actual magnetic profiles in the auroral zone are quite similar to these underlying piece-wise linear model functions, at least closer than perfectly smooth functions such as the ones employed for producing the synthetic data in section 3 (which are preferred there because of analytic tractability). Hence, we assume that our FAC

scalogram performs actually better on real data than on the synthetic examples. Nonzero correlations among different piece-wise linear model functions lead to the non-orthogonal behavior. The overall implications, however, depend on the particular subset of model functions associated with the chosen sampling scheme: (a) If for a given scale $w$ all available center times $t_{cen}$ are used, model functions with neighboring $t_{cen}$ are strongly correlated, resulting in a highly redundant and very non-orthogonal representation. This scale sampling scheme we call linear. (b) If for a given scale $w$ the chosen center times $t_{cen}$

are separated by the scale $w$, the model functions are only weakly correlated, resulting in a representation that is much less redundant and closer to orthogonality. This scale sampling scheme we call logarithmic. The underlying logic is the same as for the Haar wavelet transform. By comparing the results of linear versus logarithmic scale sampling for synthetic data, one finds that localization of center time/location and scale is more accurate with the linear sampling scheme. In logarithmic sampling, the center location of a current structure is heavily constrained by the scale $w$ that thus effectively constitutes the uncertainty

of the $t_{cen}$ (note also the uncertainty relation in wavelet analysis). Here our emphasis is on constraining FAC scales and center locations using a visualization tool, not on a full reconstruction of the FAC profile, thus we prefer to use a highly redundant set of model functions instead of an orthogonal and thus non-redundant one. Since the synthetic data are smooth profiles, and the scales are the widths of Gaussian profiles, we cannot expect that the piece-wise linear model functions identify the parameters perfectly.

Stasiewicz and Potemra (1998) made use of DWT analysis to study the multiscale properties of magnetic field gradients and plasma density perturbations observed by Freja. Another option to get scale information is by filtering the measured perturbation or the FAC density (obtained within the constrains of the methods). Using Swarm data, the study of McGranaghan et al. (2017) separates the contributions from scales $\sim$50 km, $\sim$150 km, and $\sim$350 km based on filtering (Hanning window) of the FAC density with window lengths of 8 s, 20 s, and 48 s, respectively. As compared to these techniques that rely on filtering

– assuming the variations of the magnetic field perturbation (or the FAC density) are approximated by certain basis functions

– we here compute the FAC density without removing the relative influence of large/small scale FACs on the small/large FACs that are present in the measured (or simulated) magnetic field perturbation. To distinguish between the scales we mainly rely on the $\partial_\xi \lambda_\eta$ information.

Because MSMVA is based on the statistical MVA analysis, it can be affected by two types of error, namely the statistical and the discretization errors. At short analysis windows, well-suited for the fine scale FACs, the MVA is affected by the increase of the statistical error (noise level). Longer analysis windows, suitable for the mesoscale and large scale FACs, are associated with a lower statistical error in the MSMVA, because the analysis window includes a large number of measurements. However, for long analysis windows there is an increase of the discretization error. The discretization error is caused by the use of analysis windows larger than the FAC signature, in which case the FAC is not well sampled. The error analysis of MSMVA is not the subject of this paper and it will be addressed in a future publication.

The dual-spacecraft FD and LS, estimates can be unreliable in the case of dynamic and/or inclined FACs with embedded smaller scale structures, as shown in section (4.3). Gillies et al. (2015) also pointed out that the two-spacecraft products can be compromised in regions of diffuse aurora, typically observed near the equatorward border of the auroral oval, around the midnight sector following substorms. When applied to pulsating aurora (Gillies et al., 2015), the dual-spacecraft approach does not precisely identify the boundaries of the auroral patches associated to FAC reversals, whereas the single-spacecraft precisely identifies these boundaries. One region of interest to apply MSMVA is adjacent to the polar-cap. This region is known for the high variability of the FAC geometries, typically filamentary. For such dynamic and non-planar events two-spacecraft methods are used to derive the FAC density (Lühr et al., 2016). By using MSMVA one could obtain visual information on the consistency of these results as well as on (non)planarity.

## 6 Conclusions and outlook

The technique presented in this paper extends the multiscale framework of Bunescu et al. (2015) and provides a multiscale version of the single-spacecraft FAC density estimate. The main goal of this technique is to assist the studies of aurora by an improved visualization of the FAC structures.

The MSMVA scalograms can be used to visualize and characterize the spacecraft measurements of the auroral field-aligned current structures. One can separate the planar FACs through the $R_\lambda$ scalogram and check their orientations by the $\theta$ scalogram. Using the local information about the magnetic field perturbation, the along-track FAC thickness and the orientation we obtain the FAC density scalogram. One can intuitively (visually) distinguish between the currents of different orientation (upward and downward) and their dependence on scale (e.g. get information about the FACs extent in time domain or the along-track thickness from the scale domain). The time and scale dependence of FAC density and $\partial_\xi \lambda_\eta$ scalograms are compared to obtain the local average current at the characteristic scales of the FAC signatures. Technically, the computation of the FAC density scalogram can be summarized in a few steps: a) Setup of the scanning parameters (e.g. scale range, discretization scheme, scanning steps in the time and scale domains) that suit the observed FACs. For the logarithmic scheme one needs to define the smallest scale, the number of scales, and the center time of the largest scale; b) Computation of MVA over the time and

scale grid defined in (a) to obtain the 2D data structures with information about e.g. planarity, orientation, eigenvalues. The local average FAC density is computed at each iteration on the same time and scale grid used for MVA; c) Computation of the scale derivatives of the 2D quantities, e.g. $\partial_\xi \lambda_\eta$; d) Quantitative information about selected FACs is derived for related cuts of the scalogram by looking at the dependence of the amplitude corrected quantities (e.g. $\partial_\xi \lambda_\eta$, $j_\parallel$) as a function of the corrected scale.

The application of the technique to measured Swarm data showed that the multiscale FAC density can provide results that are consistent with the typically used methods. The local multiscale FAC density shows good consistency with the dual-spacecraft FAC estimate, with deviations within 15% for larger scales (>150 km), which are well resolved by the dual-spacecraft estimate. In the mesoscale and small-scale range deviations can be larger than 70%. The comparison of the local multiscale FAC density with the single-spacecraft estimate shows better consistency also at small scales, since the single-spacecraft estimate has a resolution of ∼7.5 km (to be compared with the local multiscale FAC estimate resolution of 0.7 km, when high frequency magnetic field data are used). We observe higher deviations in the case of inclined structures, since the single-spacecraft estimate neglects the north-south component of $\Delta \boldsymbol{B}$.

We applied the MSMVA technique for the computation of multiscale FAC density using two scale scanning procedures, linear and logarithmic. The logarithmic sampling scheme shows consistent information for both synthetic and observed FACs. We observe that the orientation and the local FAC density are typically in good agreement with the linear sampling scheme. However, the location and scale information provided by $\partial_\xi \lambda_\eta$ is affected by the non-overlapping of the intervals in the logarithmic scheme and can provide inconsistent information at scales that are not properly sampled, e.g. if the scale is not centered on the FAC element.

Future work will address an error analysis of the multiscale information. We plan to use the bootstrap method to evaluate the impact of the error level of the input magnetic field perturbation on the output multiscale information. The multiscale approach offers a good setup to study the distribution of the statistical error (noise level), predominant at small scales, and of the discretization error (imperfect sampling of the FACs), predominant at large scales, where the analysis window becomes larger than the FAC signature. We expect thus to properly separate between different error sources of the multiscale information, e.g. multiscale FAC density, FAC localization, thickness, and orientation.

So far, the method does not properly take into account the geometry of the FAC structures. At this point one can select thresholds in the planarity and, accordingly, apply a mask to other quantities. Masked results were not included since they affect the overall structure of the displayed quantities. However, the masks are suitable to select a certain type of FACs, e.g. planar or non-planar. Further improvement might address a thorough study on finite structures to properly quantify the influence of the scale and how the planarity can better weight the results. In this respect we plan to extend the method to a dual-spacecraft multiscale analysis by using Swarm observations. With two satellites one can correlate the quantities, e.g. planarity, in longitudinal direction.

Swarm provides an appropriate platform to quantify and check the planarity as derived by eigenvalue ratios, based on the similarity of the results obtained from the two longitudinally separated measurement points. This analysis is particularly

useful for the fine structure of the aurora which cannot be addressed by other dual-spacecraft methods due to the spacecraft configuration limitations.

At present, the scale dependence of FACs properties can be investigated using Swarm, FAST, and Cluster high-resolution
5 measurements. The technique can be also adapted to other more recently launched missions, like MMS, particularly for conjugate measurements, e.g. MMS/Swarm.

*Acknowledgements.* We acknowledge the support from the Deutsche Forschungsgemeinschaft (DFG) through grant VO 855/4-1 MuSICAL in the context of the DFG Priority Programme SPP 1788 DynamicEarth, SIFACIT ESA contract 4000118383/16/I–EF, and STAR EXPRESS contract 119/2017 with Romanian Space Agency. We acknowledge the use of the conjunction finder interface available at https://swarm-aurora.com/conjunctionFinder/. We acknowledge the use of the Swarm magnetic field data as well as the L2 products provided by ESA
10 at ftp://swarm-diss.eo.esa.int/. We acknowledge NASA contract NAS5-02099 and V. Angelopoulos for the use of data from the THEMIS Mission, specifically, S. Mende and E. Donovan for the use of the ASI data (http://themis.ssl.berkeley.edu/ data/themis/thg/), the CSA for logistical support in fielding and data retrieval from the GBO stations, and NSF for support of GIMNAST through grant AGS-1004736. We acknowledge the use of THEMIS Data Analysis Software available at http://themis.ssl.berkeley.edu/. The Tsyganenko magnetic field model involves the use of the DST and OMNI data. The DST data are provided by the World Data Center for Geomagnetism, Kyoto
15 (http://wdc.kugi.kyoto-u.ac.jp/). The OMNI data were obtained from the GSFC/SPDF OMNIWeb interface at http://omniweb.gsfc.nasa.gov.

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

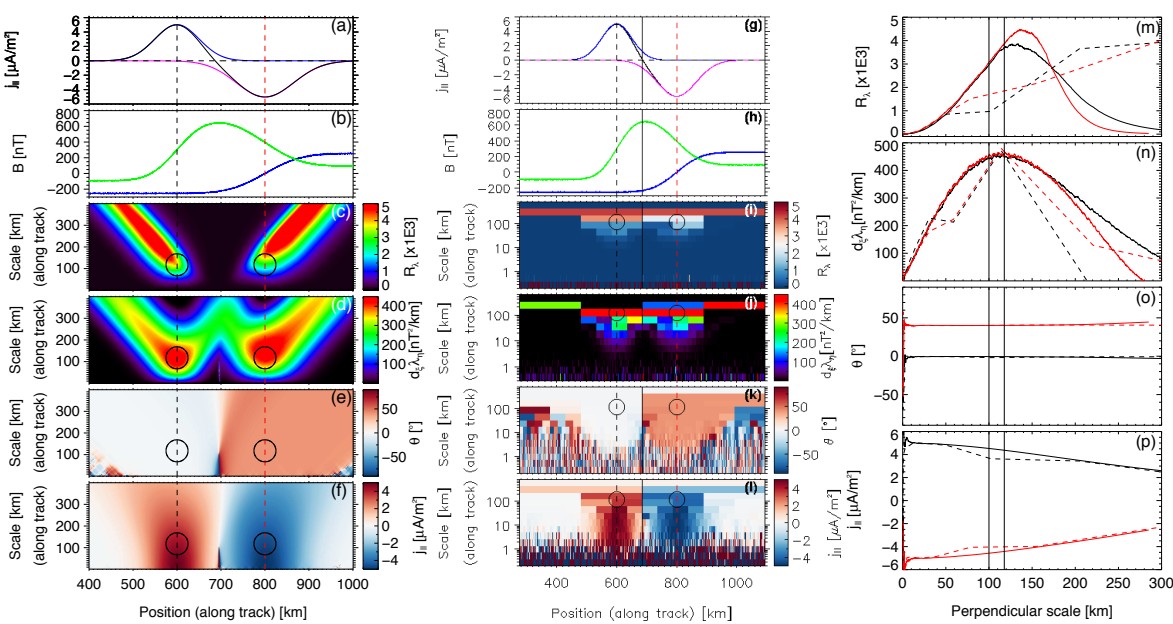

**Figure 1.** MSMVA analysis for the linear and logarithmic scheme. a) Input FAC density for FD (magenta), FU (blue) and the summed contribution from both FACs (black); b) Magnetic field perturbation in the $(x, y)$ frame with $B_x$ (blue) and $B_y$ (green); c) Planarity, $R_\lambda$; d) FAC scale/location, $\partial_\xi \lambda_\eta$; e) Orientation, $\theta$; f) Multiscale FAC density; Panels (g)-(l) show the same quantities for the logarithmic sampling scheme. Panels (m)-(p) show the profile of $R_\lambda$, $\partial_\xi \lambda_\eta$, $\theta$, and $j_\parallel$ at the center of FD/FU structures indicated by the vertical black/red dashed lines in panels (a)-(f) and (g)-(l). The vertical black dashed lines indicate $w_{1\sigma}$ and fwhm scales discussed in the text. Solid/dashed lines indicate the profiles for the linear/logarithmic scanning.

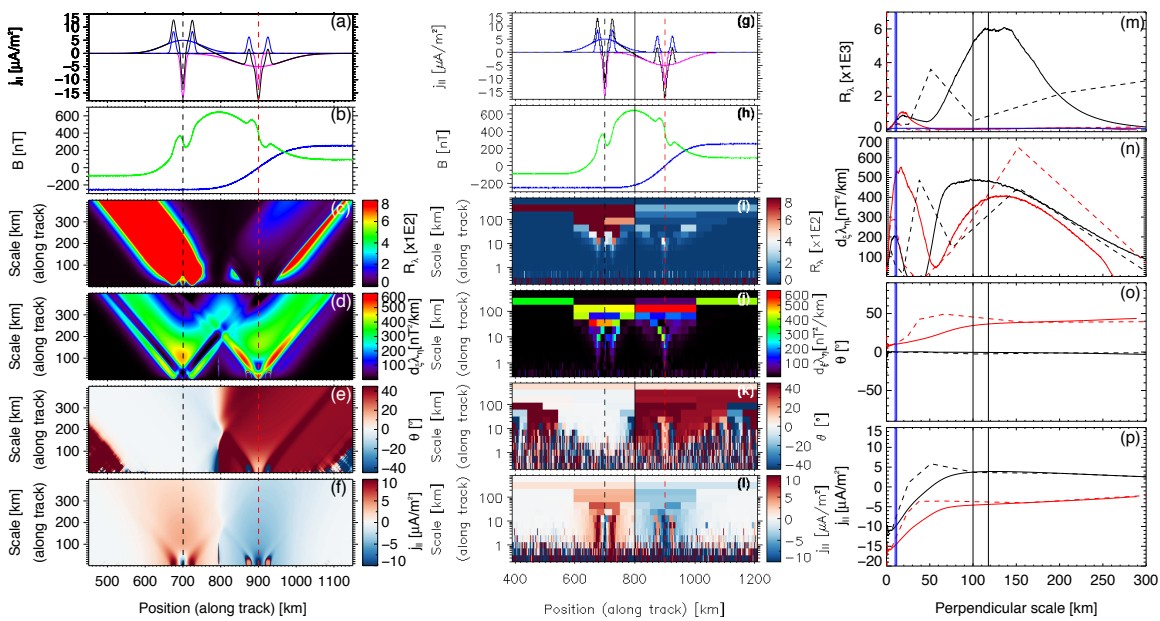

**Figure 2.** Same panels as in Figure 1.

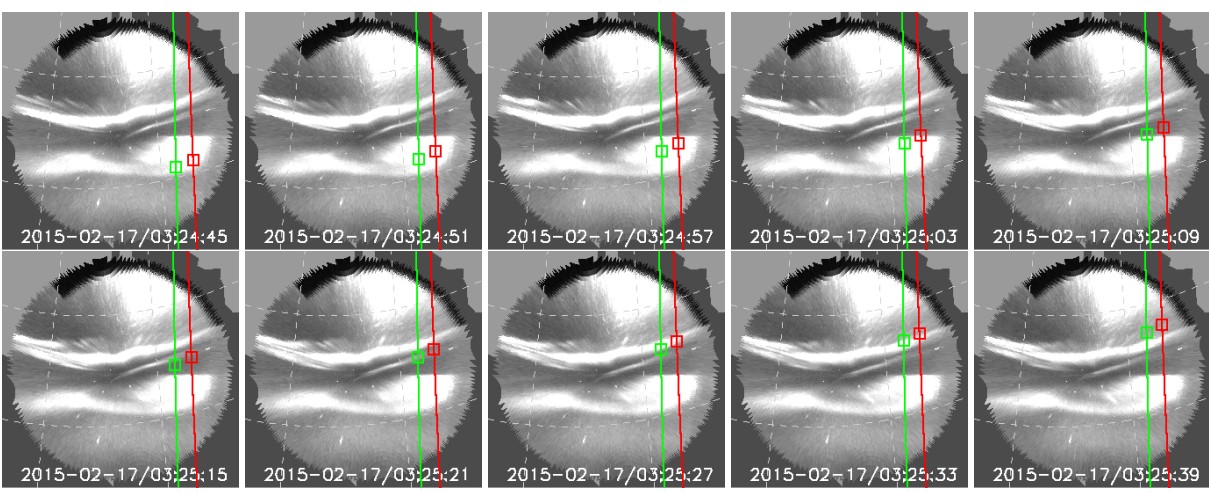

**Figure 3.** Optical frames from SNKQ station mapped in geographic coordinates. The tracks show the ionospheric projection of SwA (green) and SwC (red). At the time of the frame the spacecraft mapped position is shown by the square symbols. The time is overplotted on each frame and covers the interval from 03:24:45 to 03:25:39.

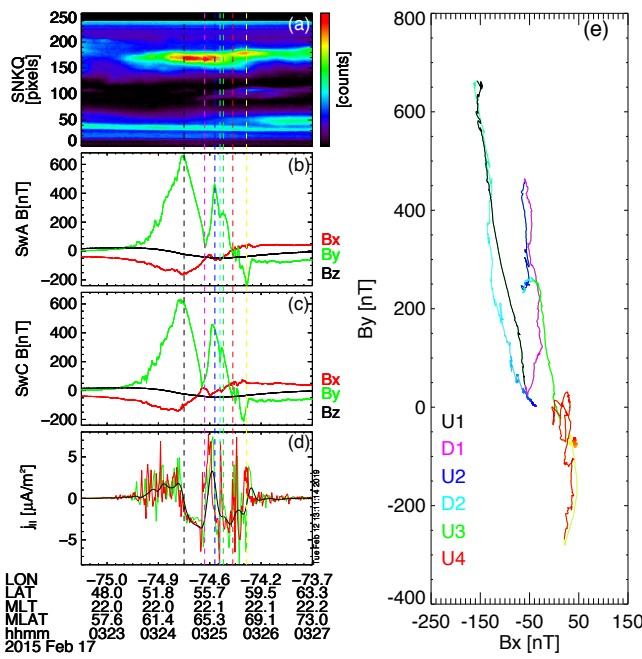

**Figure 4.** Left: a) Keogram from SNKQ station; b-c) Magnetic field perturbation from SwA and SwC. d) Single-spacecraft FAC density estimated by the FD method (L2 product) on SwA (green) and SwC (red); The FAC density estimate based on the two-spacecraft FD method (L2 product) is shown by the black line. The vertical dashed lines indicate the beginning of various FAC elements. e) Hodogram representation of $B_\perp$. The hodogram is first represented in a rainbow color scale (blue to red) on which we superpose a layer of identified FAC intervals using discrete colors associated to the labels. For each FAC segment we use the same color as in the left panels to indicate the beginning of the respective FAC element. The U and D labels indicate upward and downward FACs with the same color code.

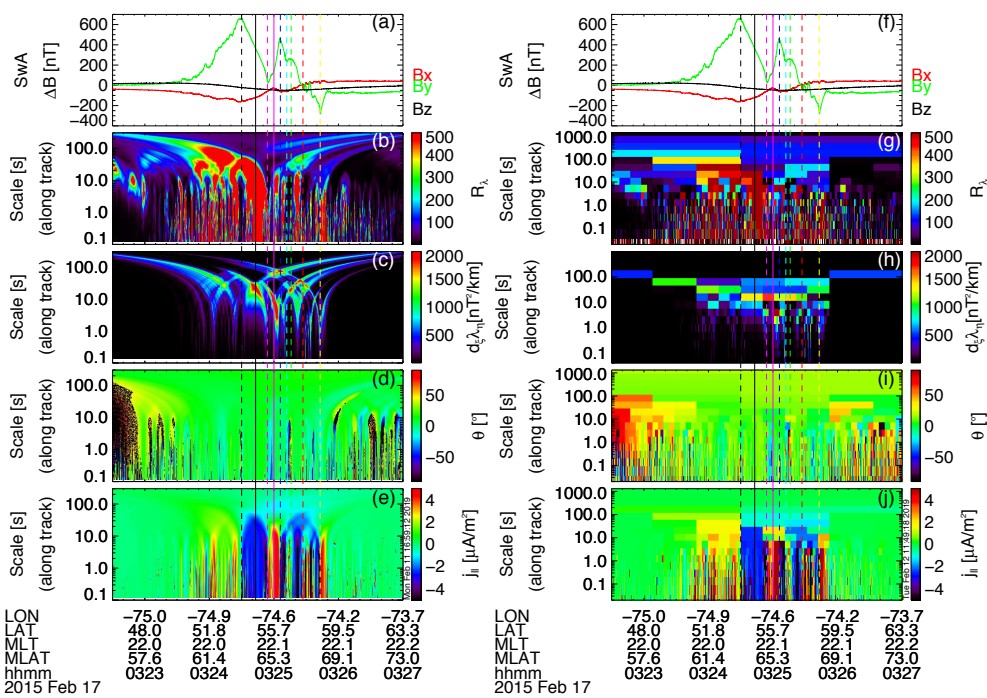

**Figure 5.** MSMVA analysis for the linear (left) and logarithmic (right) scheme. Left: a) Magnetic field perturbation; b) Planarity $R_\lambda$; c) FAC location and characteristic scale $\partial_\xi \lambda_\eta$; d) Orientation; e) Multiscale FAC density; Right: Panels (f)-(j) show the same quantities for the logarithmic scale sampling. MSMVA paramters represented as a function of the along track scale. The vertical dashed lines delimit FAC segments as shown in Figure 4. The vertical solid lines indicate the times for which we show the sections in Figure 6.

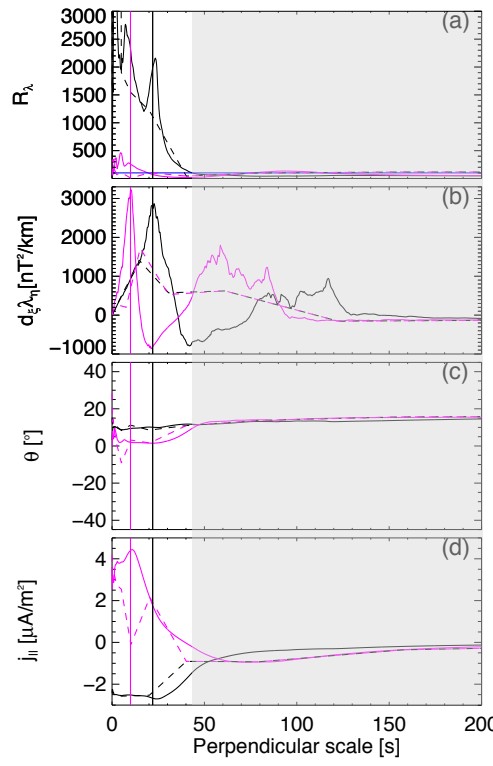

**Figure 6.** Sections in the MSMVA scalograms showing the dependence of the parameters as a function of the perpendicular scale (scale corrected). a) $R_\lambda$; b) $\partial_\xi\lambda_\eta$; c) $\theta$; d) $j_\parallel$. Solid/dashed lines indicate the profiles for the linear/logarithmic scale sampling scheme. The profiles are taken in the middle of the upward and downward FACs located at 03:24:43 and 03:25:00, respectively. These times are indicated by the vertical solid lines in Figure 5 (same color code). The vertical black/magenta lines indicate the scales of these FAC elements as identified by $\partial_\xi\lambda_\eta$. The horizontal blue line in (a) indicates a reference level, $R_\lambda$=100, discussed in the text. The marked gray area indicates the region where $R_\lambda <100$ for the selected sections.

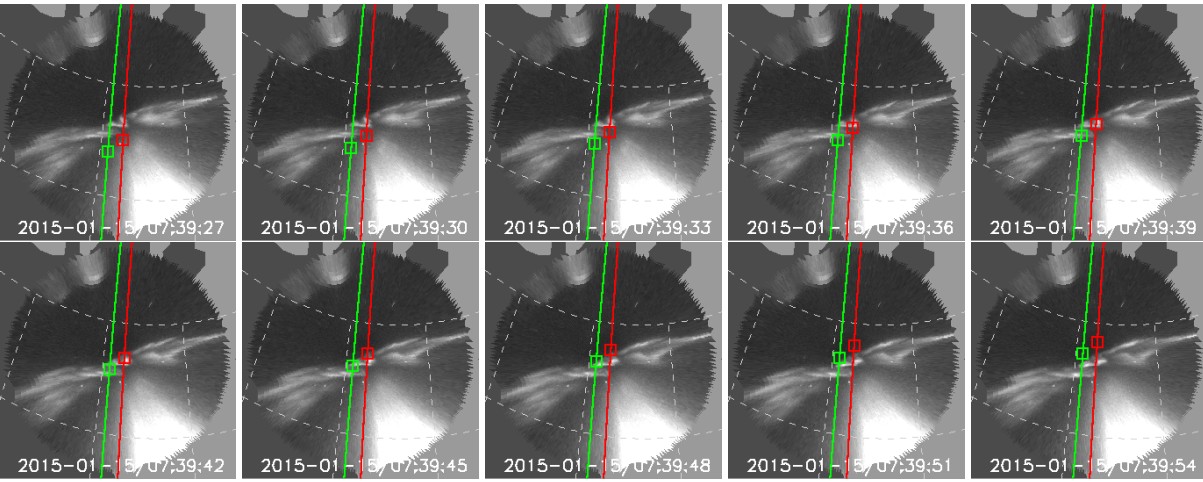

**Figure 7.** Mapped optical frames in geographic coordinates from RANK station. The tracks show the ionospheric projection of SwA (green) and SwC (red). At the time of the frame the spacecraft mapped position is shown by the square symbols. The time is overplotted on each frame and covers the interval from 07:39:27 to 07:39:54.

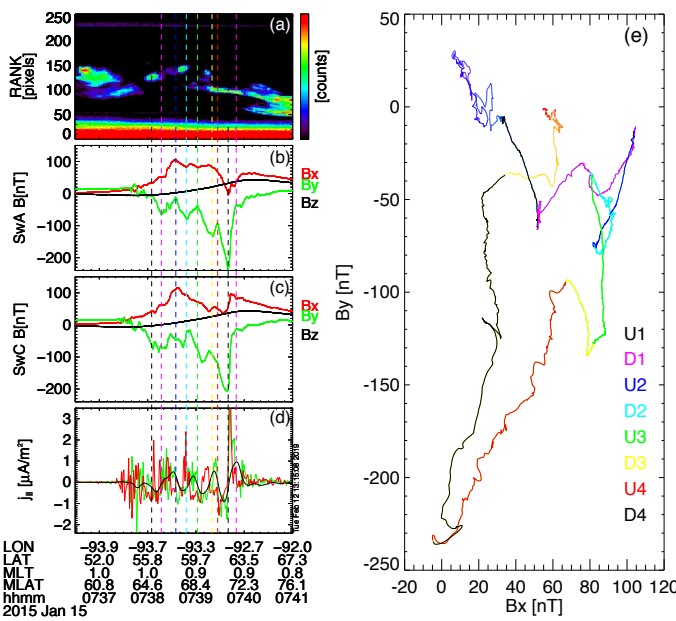

**Figure 8.** Same panels as in Figure 4. We note that some of the delimited FACs have also an internal structure, e.g. first magenta interval labeled D1.

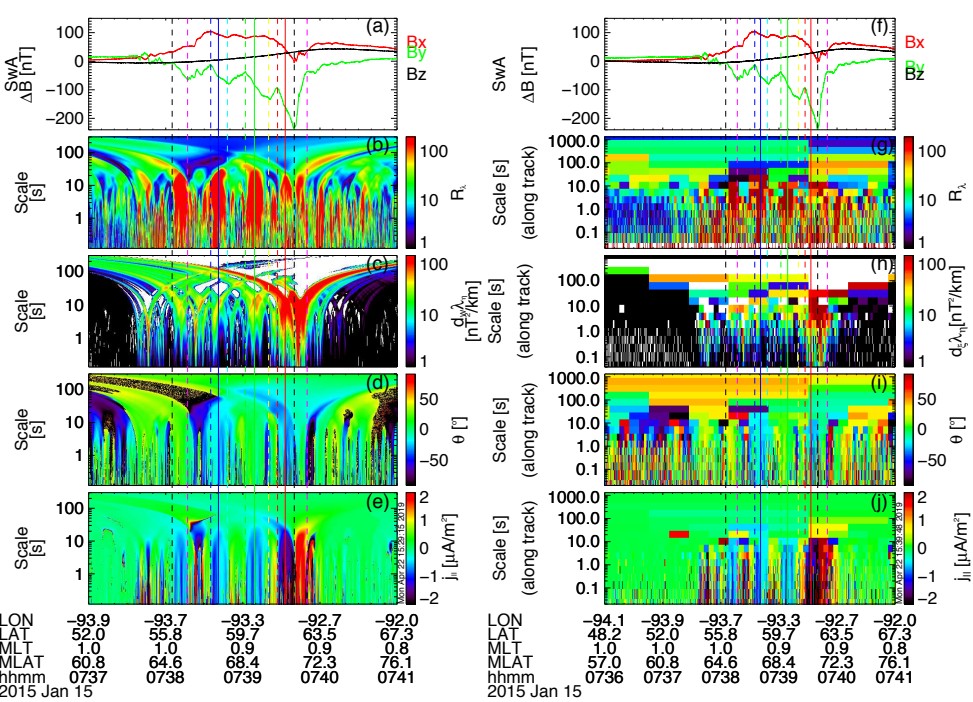

**Figure 9.** Same panels as in Figure 5. The vertical solid lines indicate the center times of U2 (blue), U3 (green), and U4 (red) FACs for which we show the sections in Figure 10.

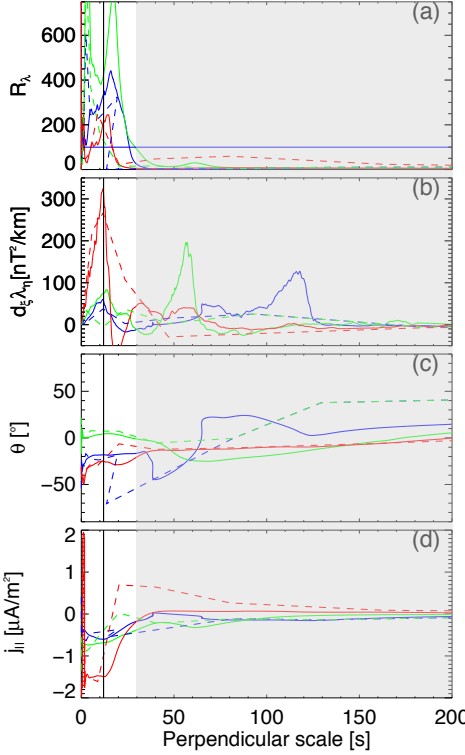

**Figure 10.** Same panels as in Figure 6. Solid/dashed lines indicate the profiles for the linear/logarithmic scale sampling scheme. The profiles are taken in the middle of the upward FACs located at 07:38:41 (blue), 07:39:09 (green), and 07:39:33 (red). These times are indicated by the vertical solid lines in Figure 9.

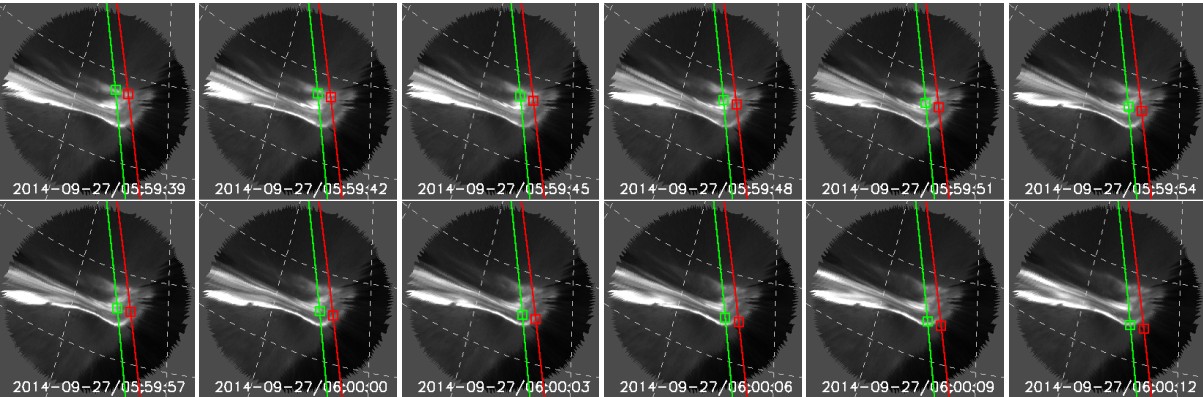

**Figure 11.** Mapped optical frames in geographic coordinates from FSMI station. The tracks show the ionospheric projection of SwA (green) and SwC (red). At the time of the frame the spacecraft mapped position is shown by the square symbols. The time is overplotted on each frame and covers the interval from 05:59:39 to 06:00:12.

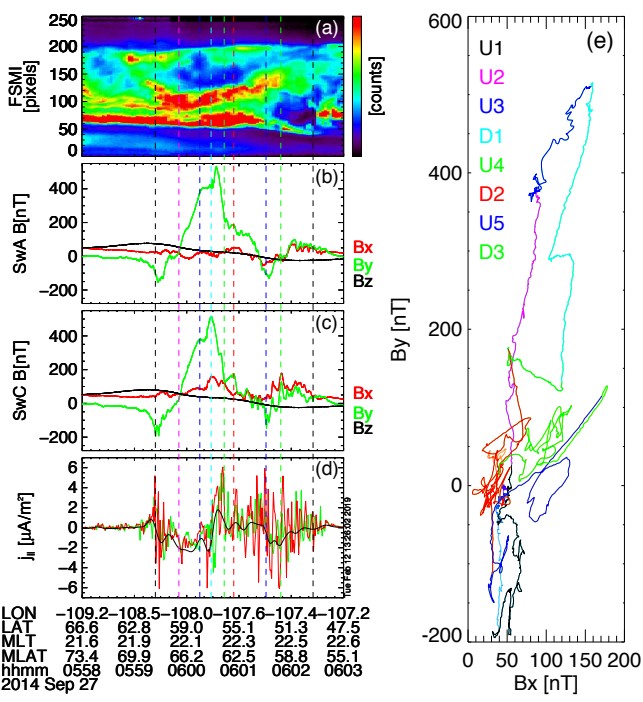

**Figure 12.** Same panels as in Figure 4. The vertical lines indicate individual FACs, e.g. first black, magenta and blue intervals, or larger intervals with small-scale FAC signatures, e.g. red, second blue, or second green intervals.

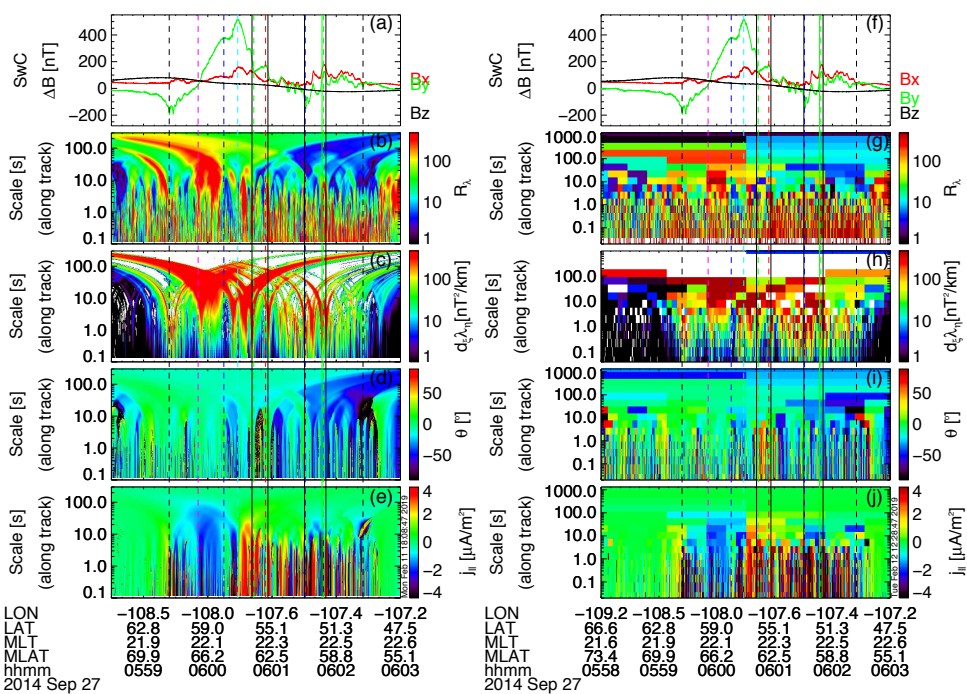

**Figure 13.** Same panels as in Figure 5. The vertical dashed lines indicate the FAC segments or intervals of small-scale FACs. The vertical solid lines indicate the central times of small-scale downward (black) and upward (green) FAC elements for which we show the sections in Figure 14.

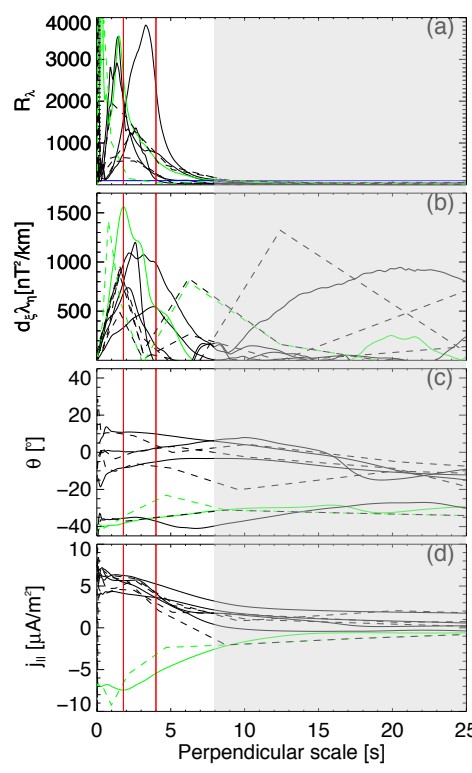

**Figure 14.** Same panels as in Figure 6. Solid/dashed lines indicate the profiles for the linear/logarithmic scale sampling scheme. The vertical red lines indicate the domain of scales identified by $\partial_\xi \lambda_\eta$ for the upward (green) and downward (black) FACs.

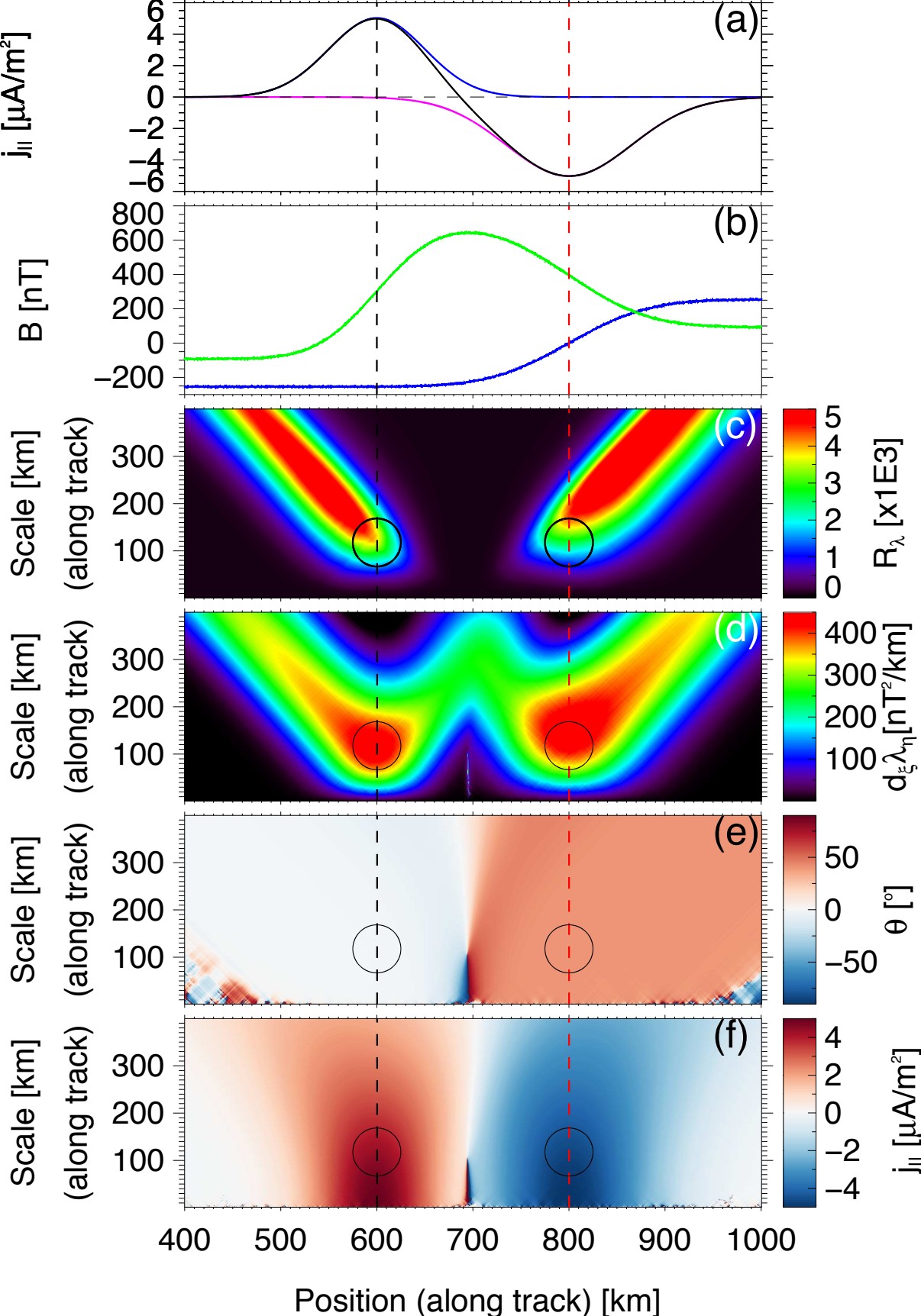

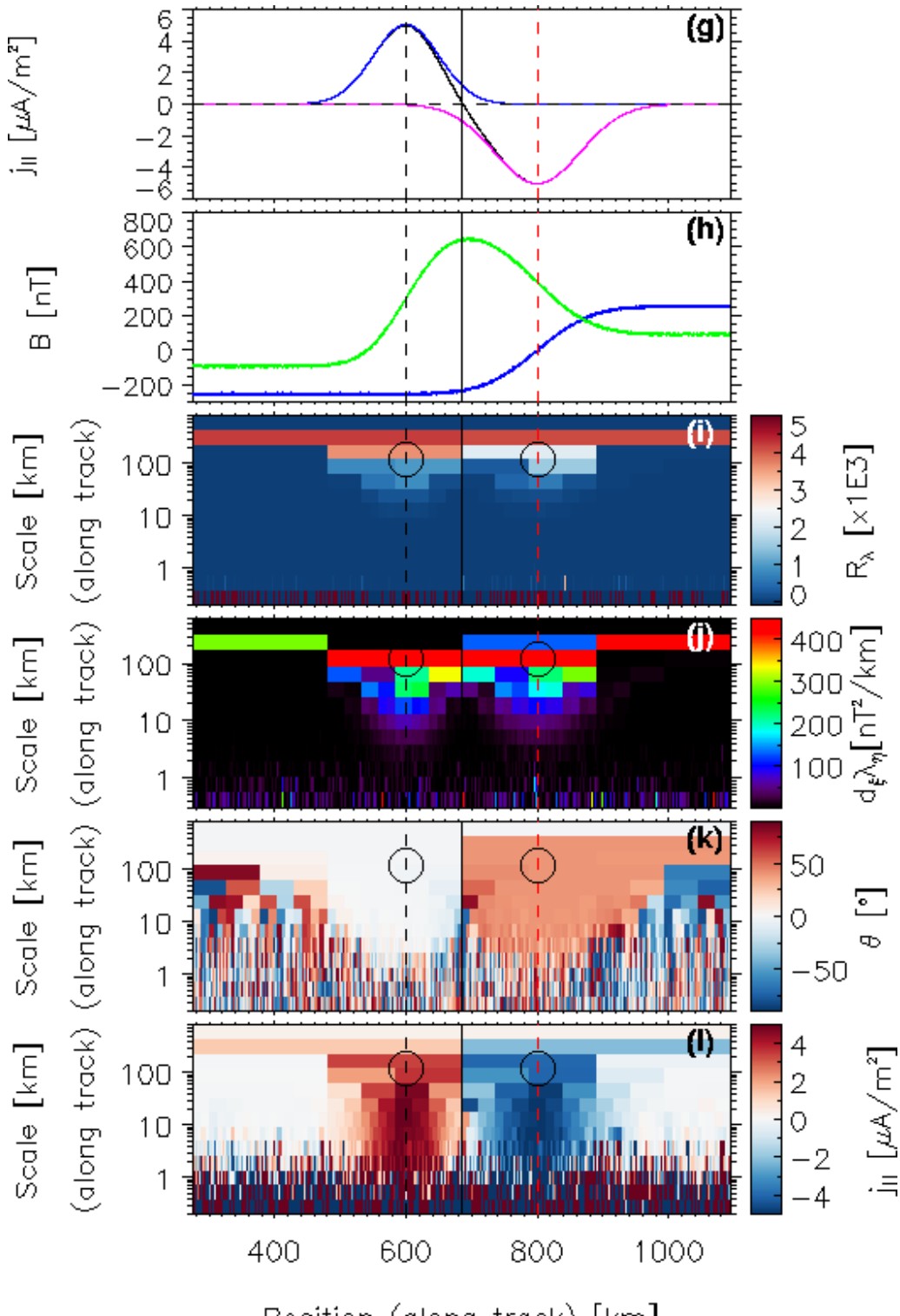

Position (along track) [km]

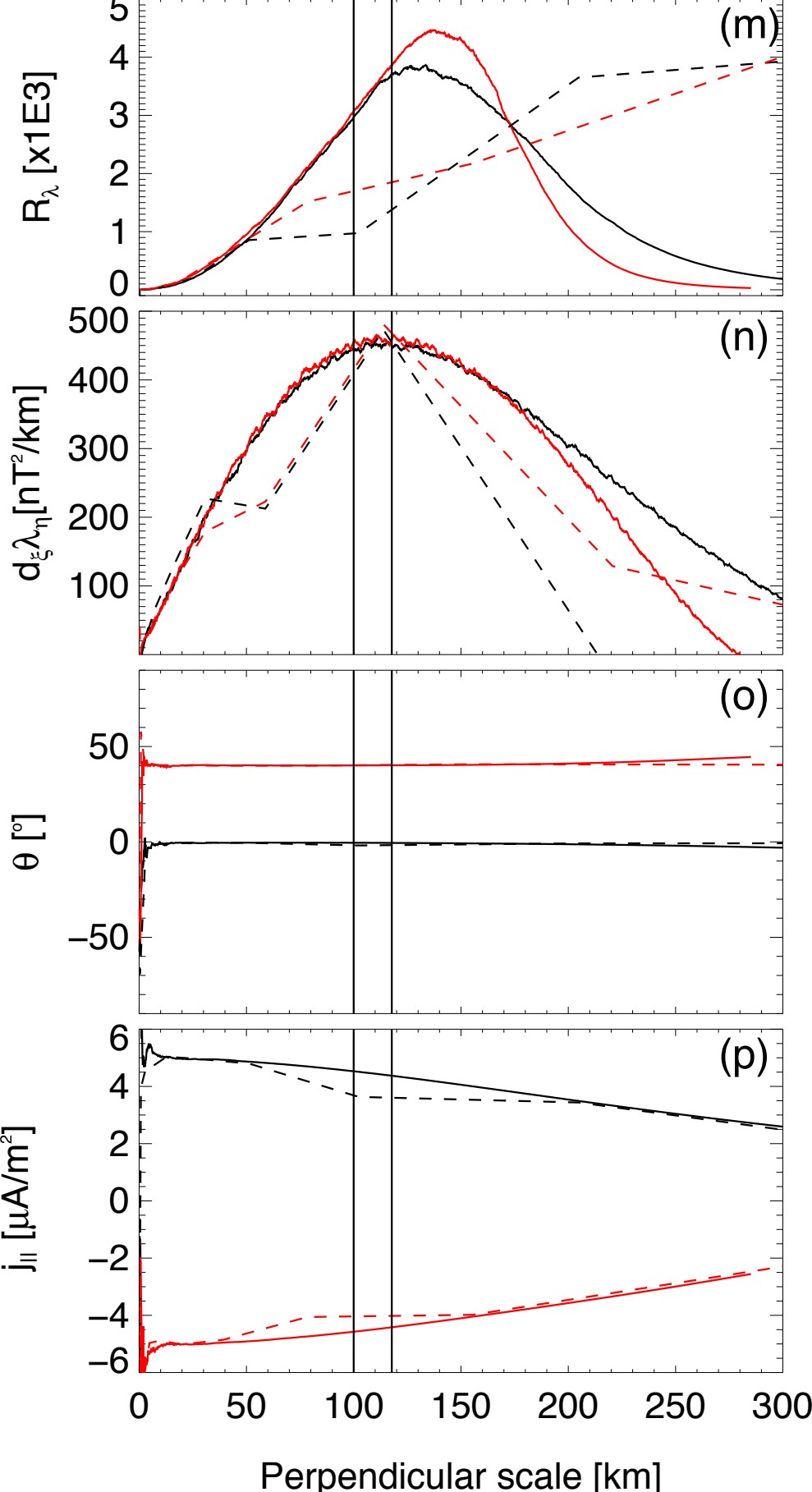

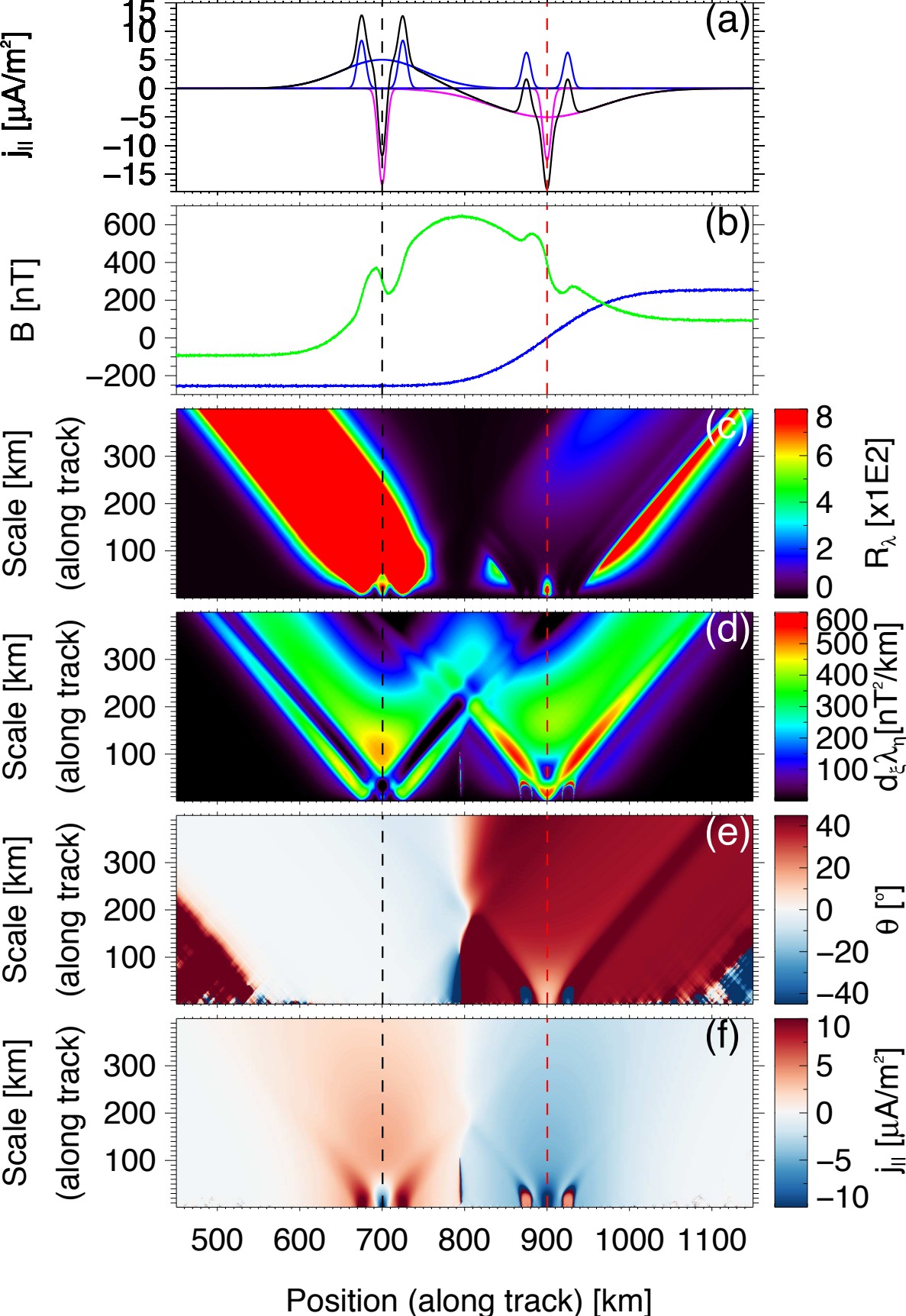

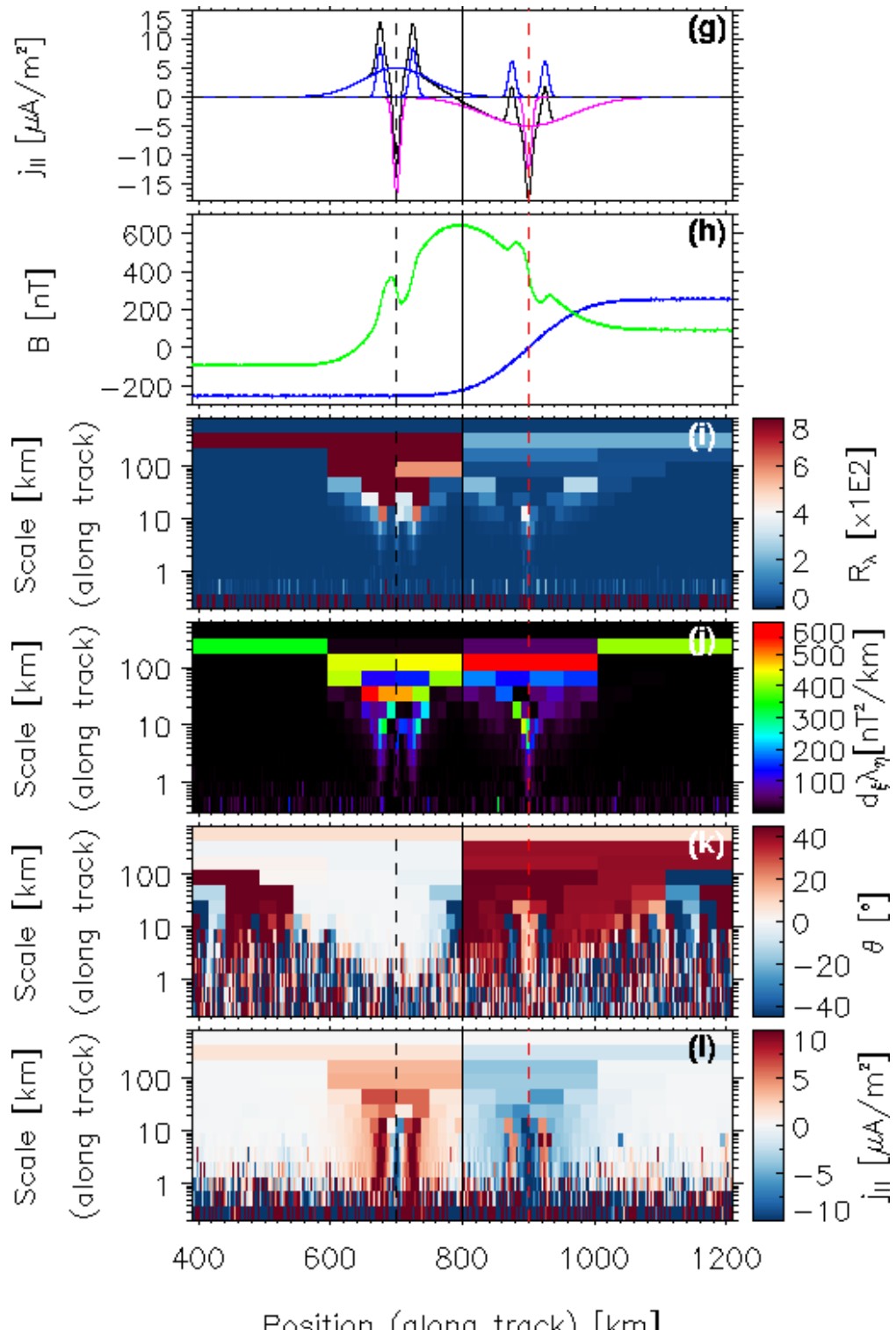

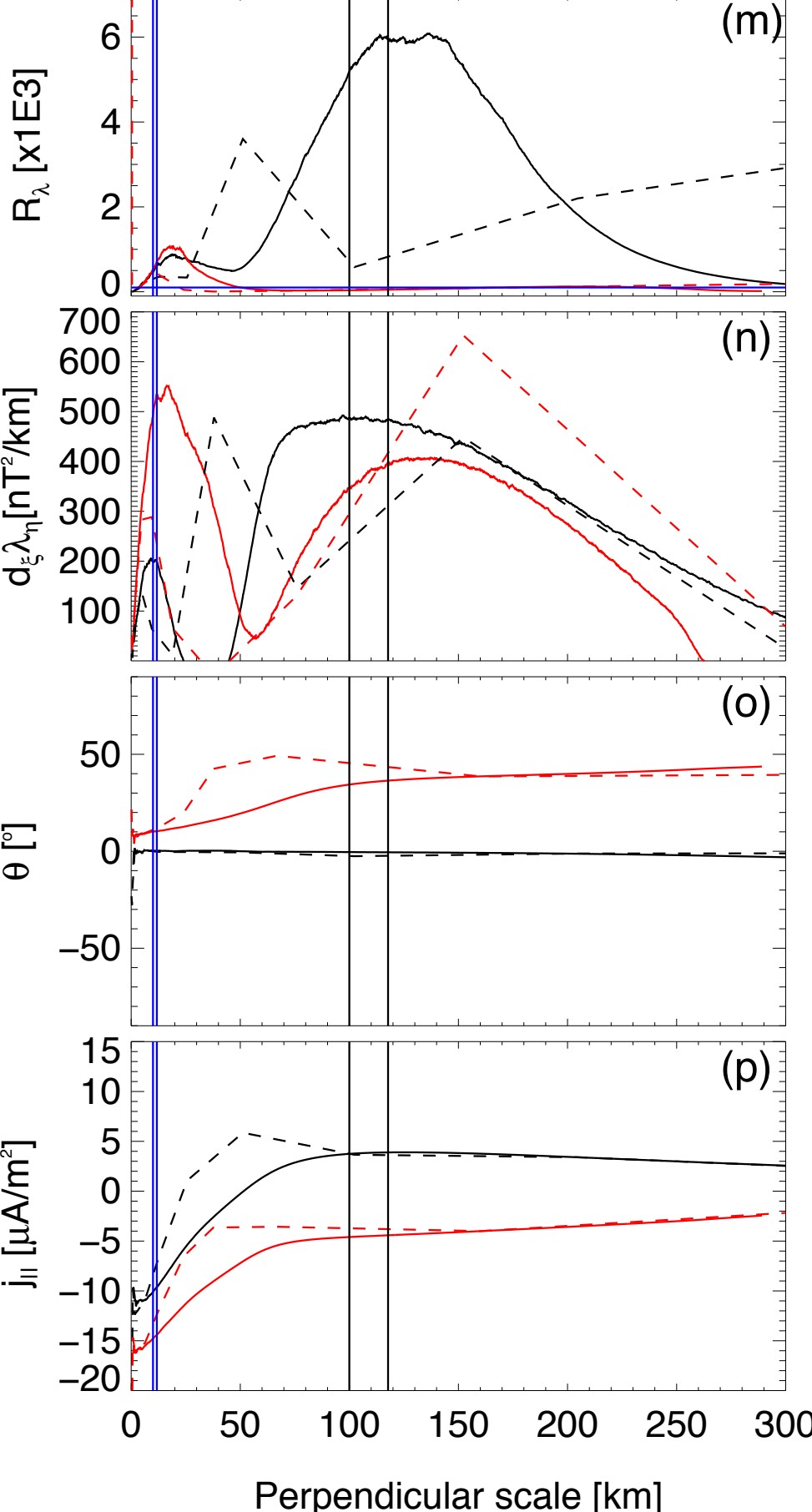

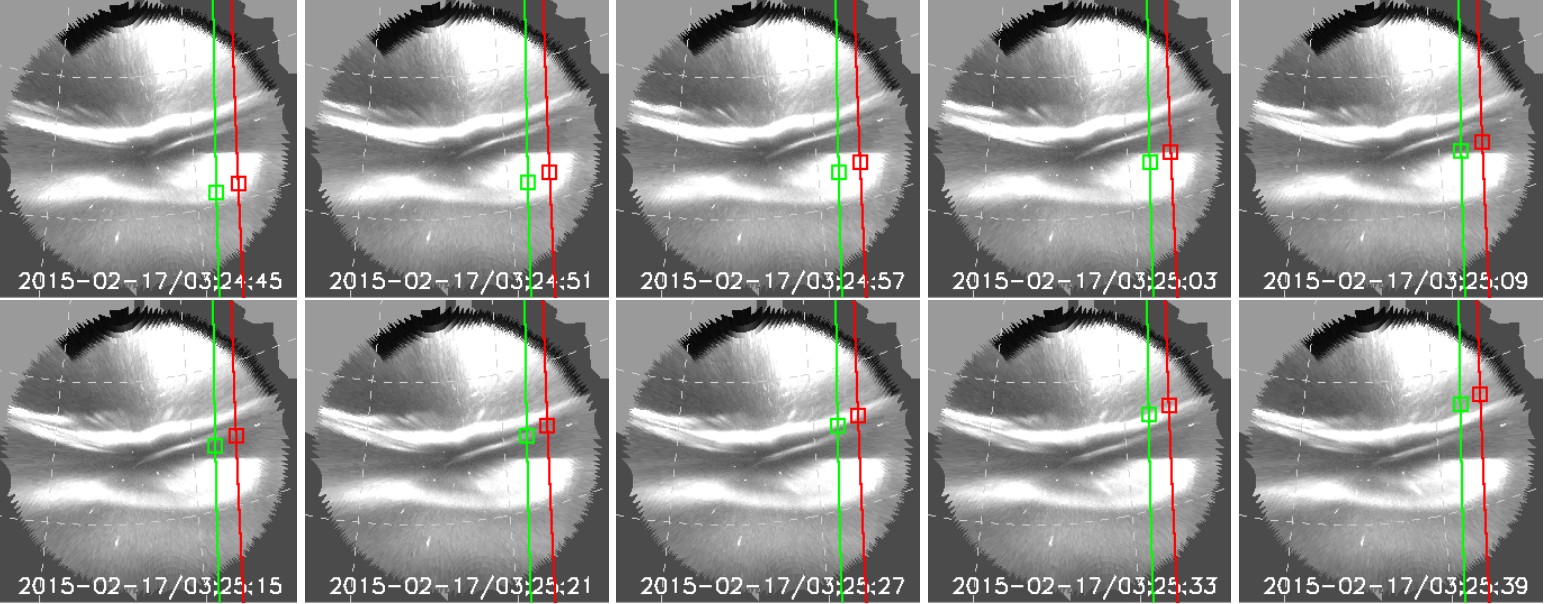

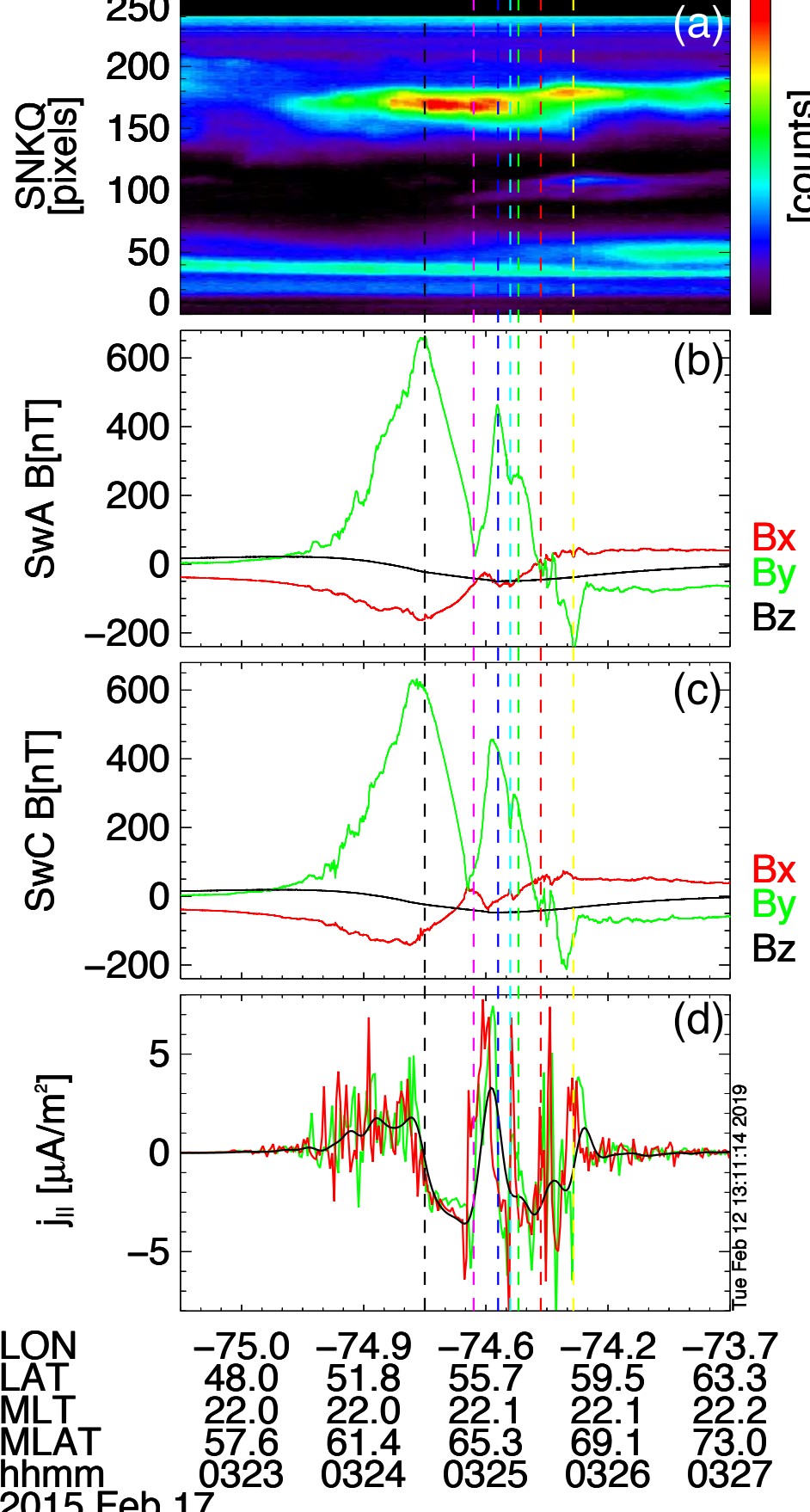

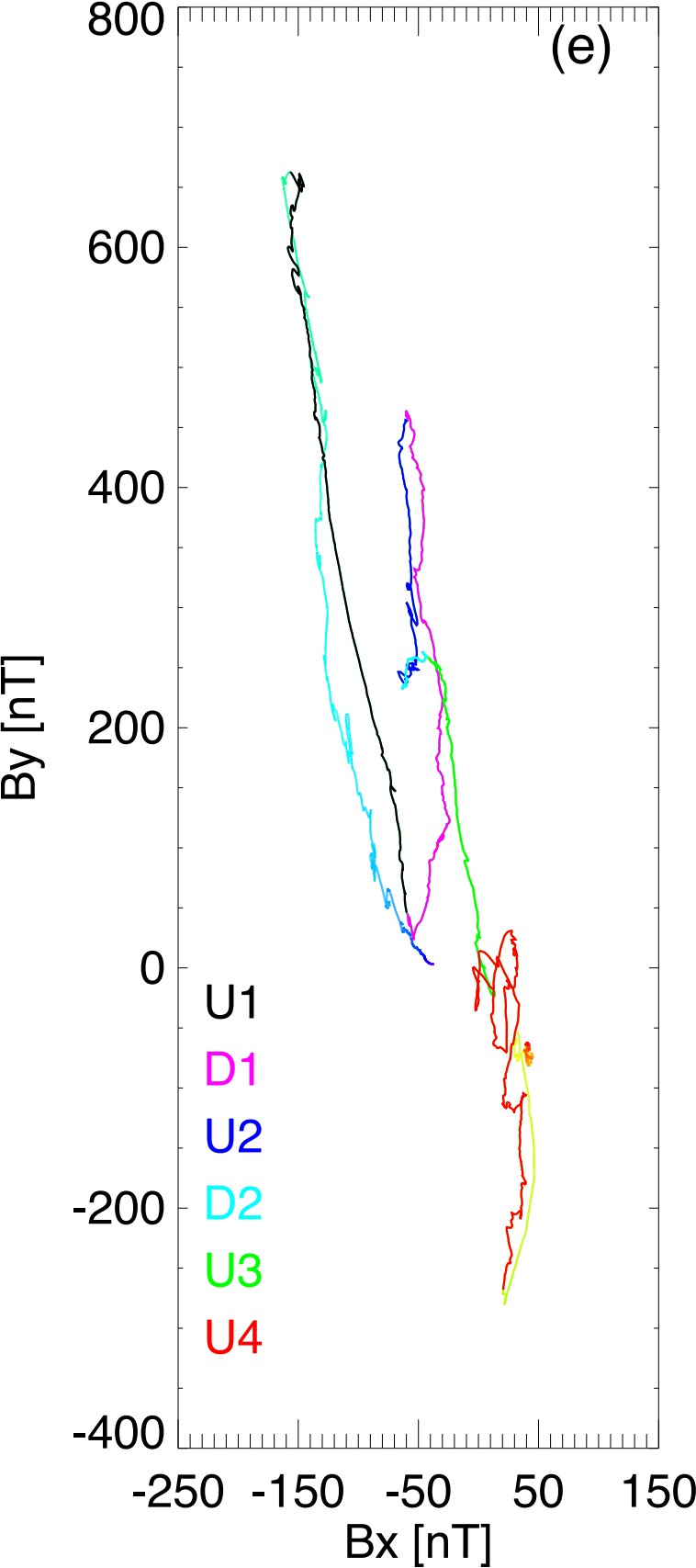

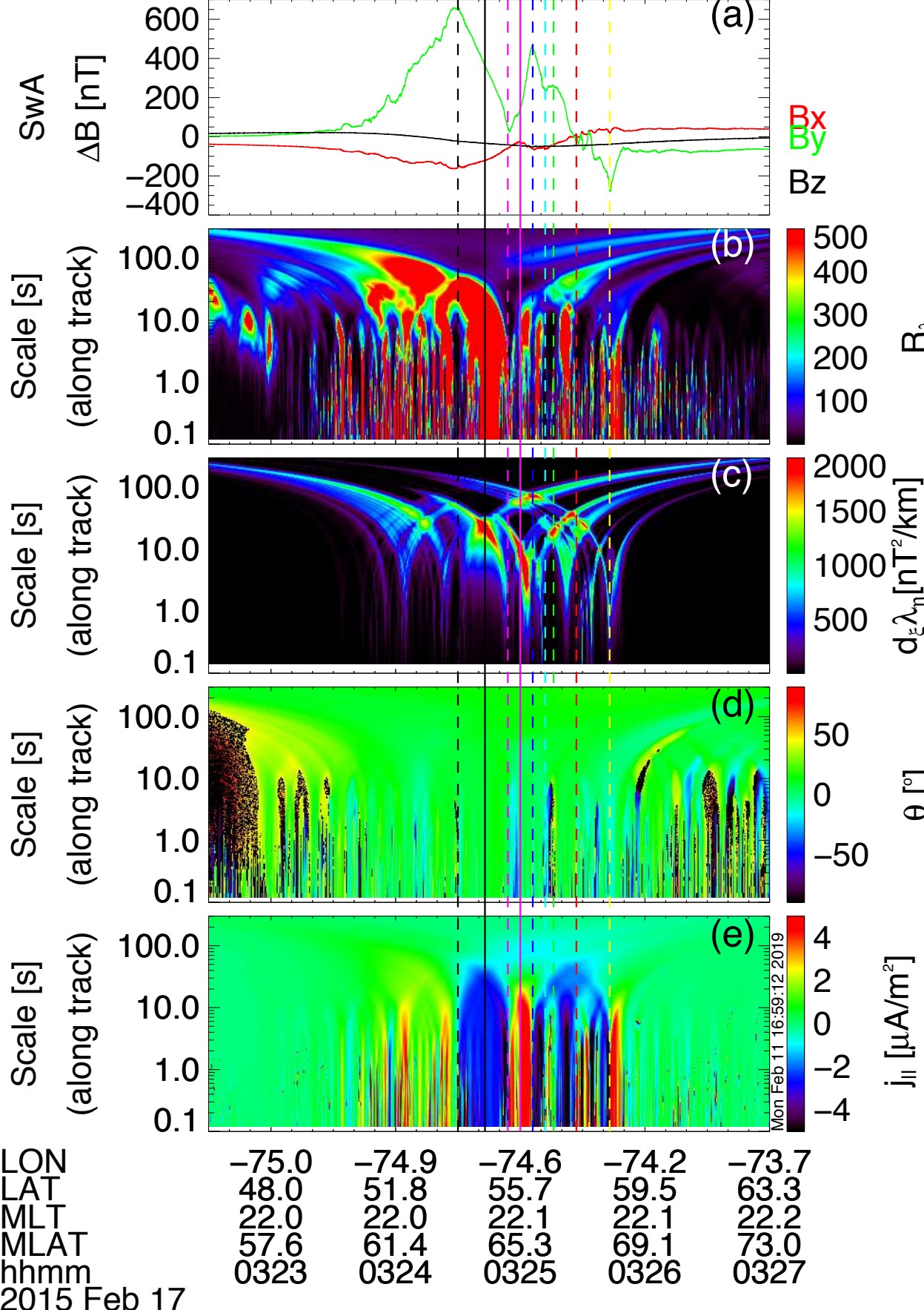

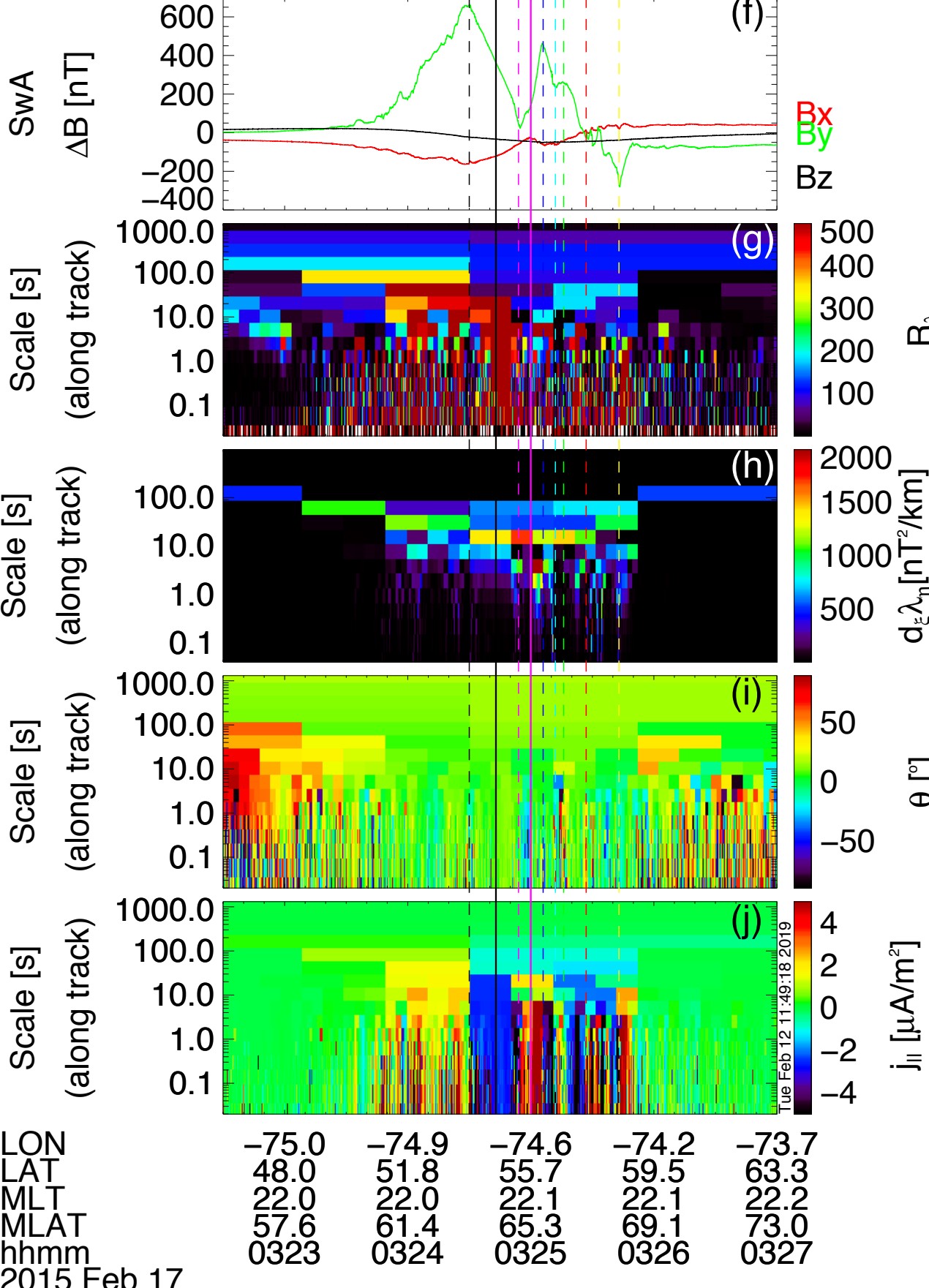

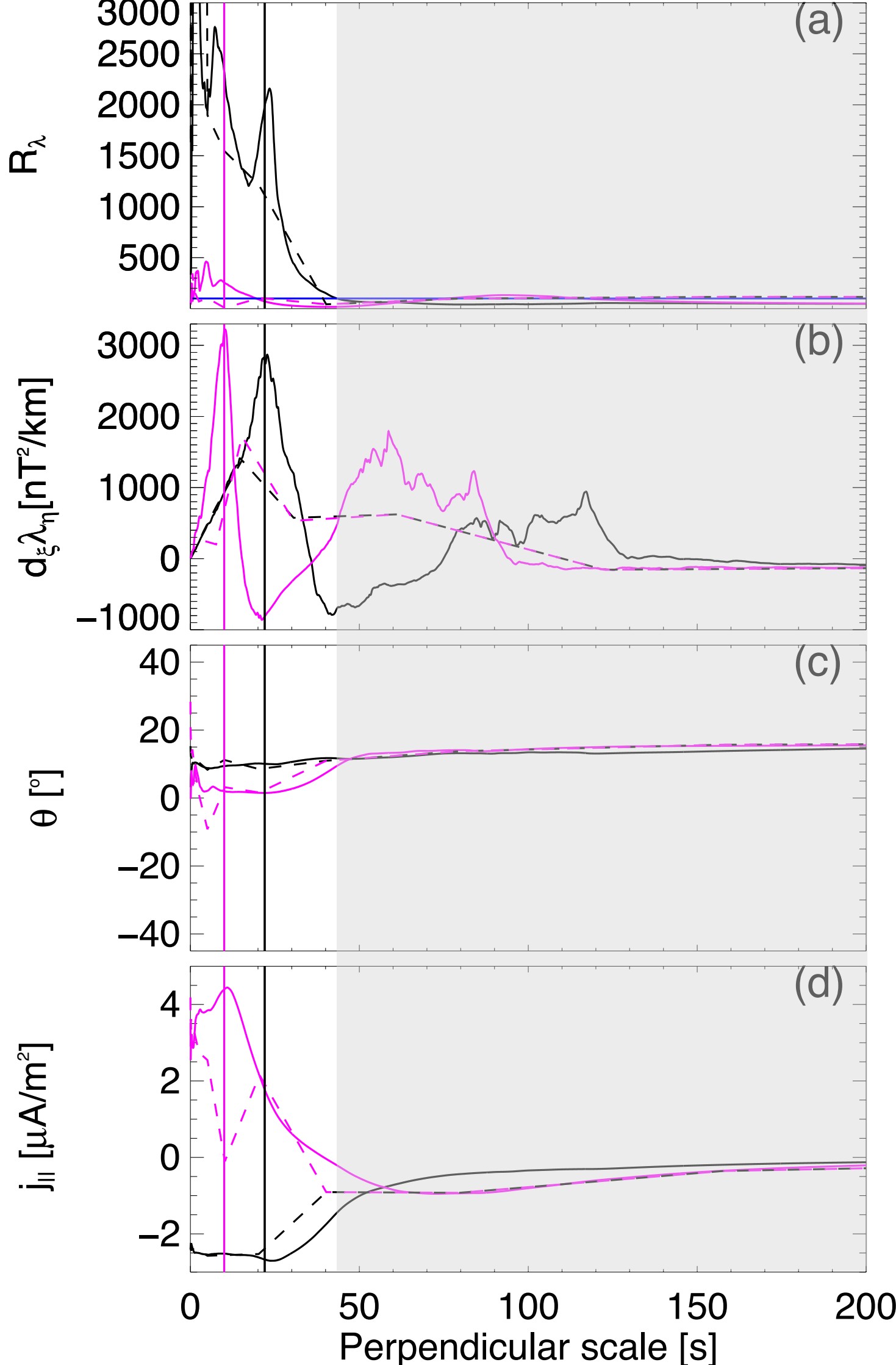

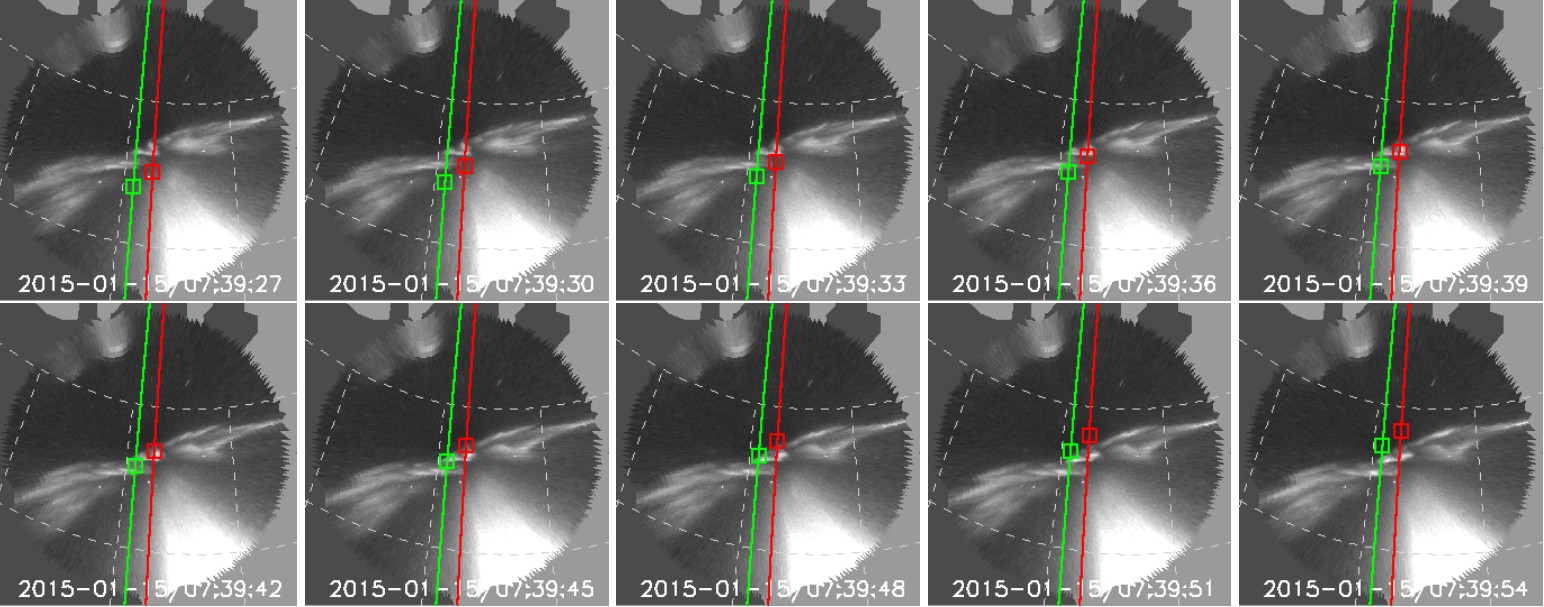

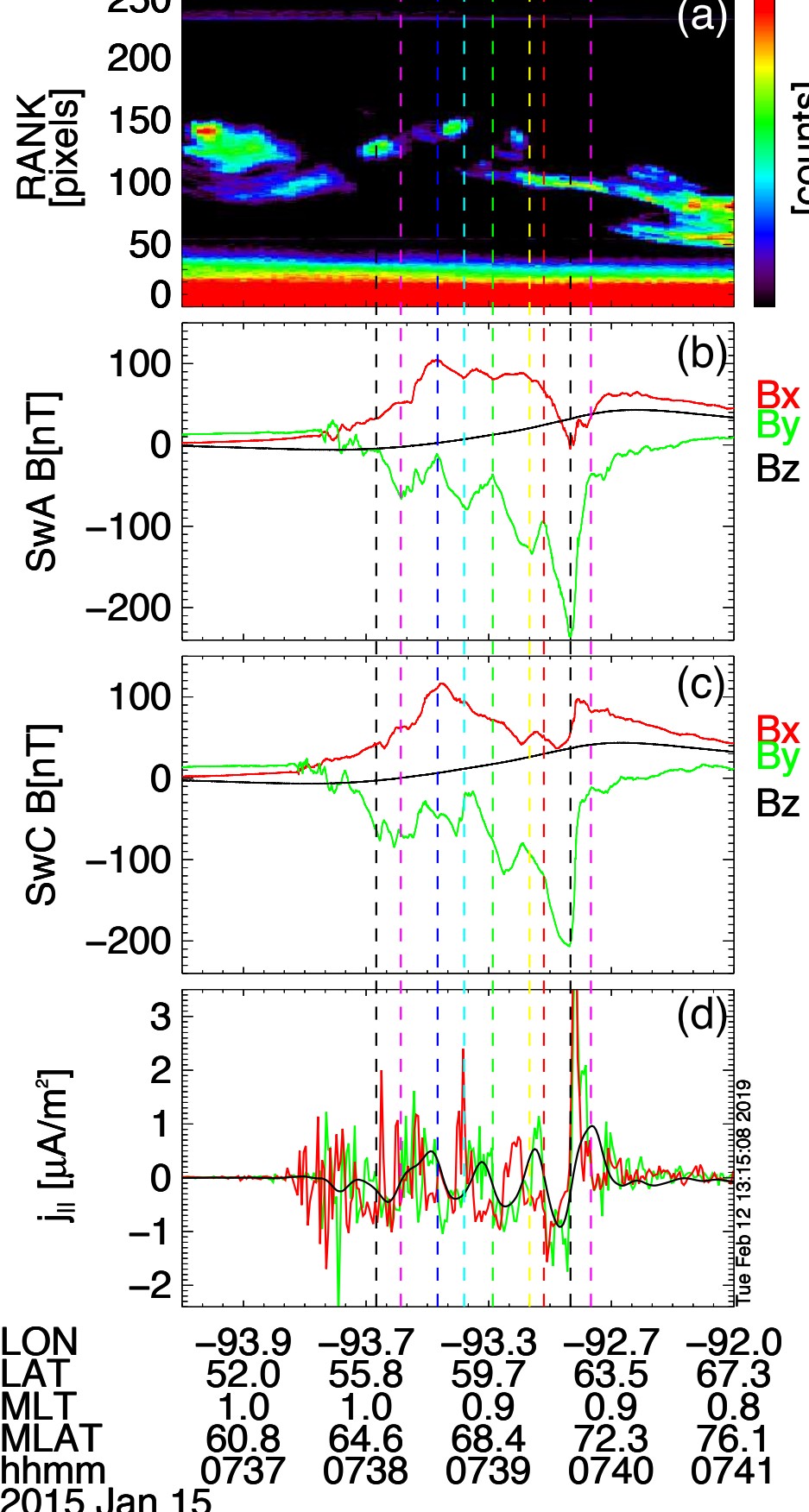

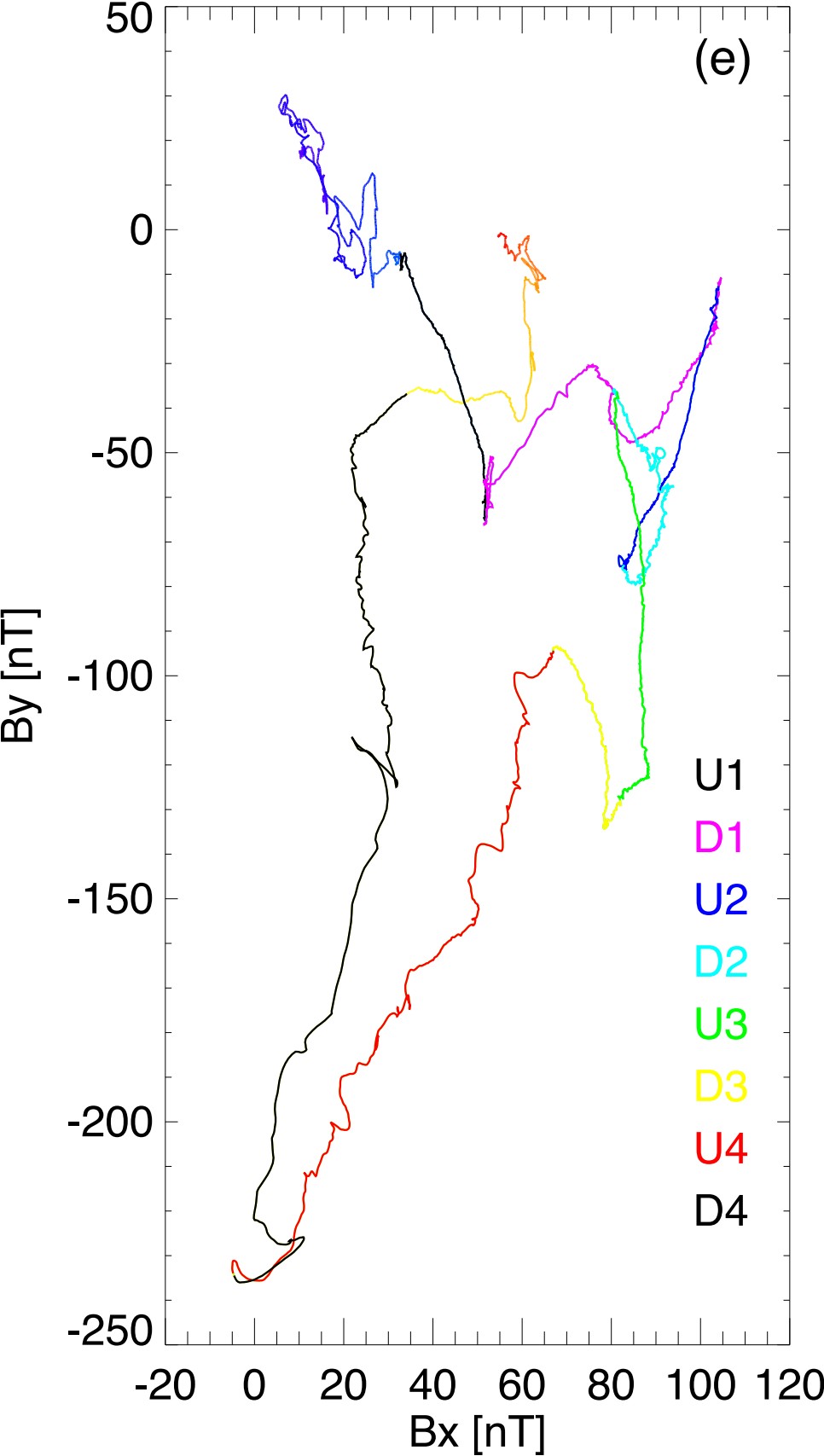

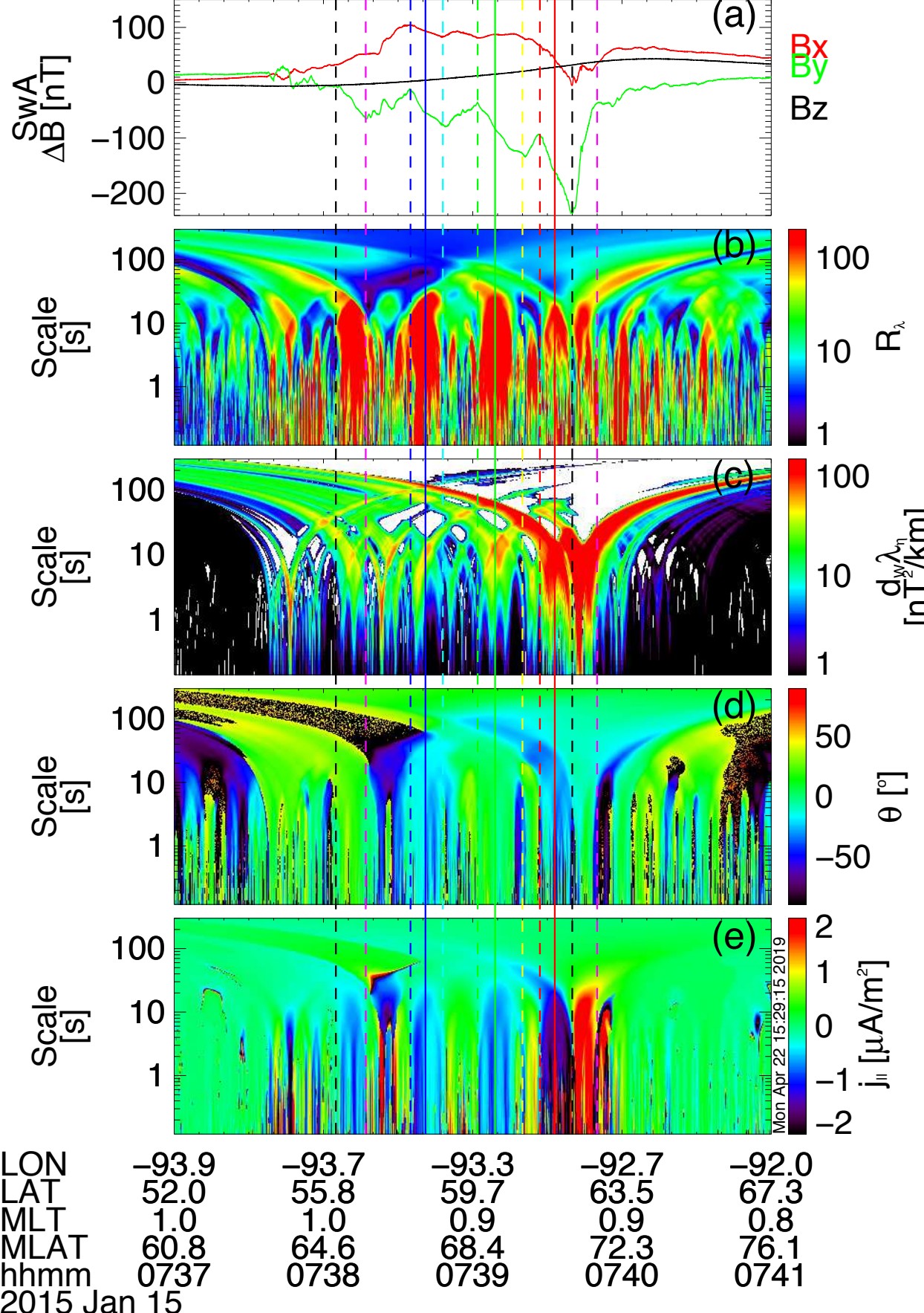

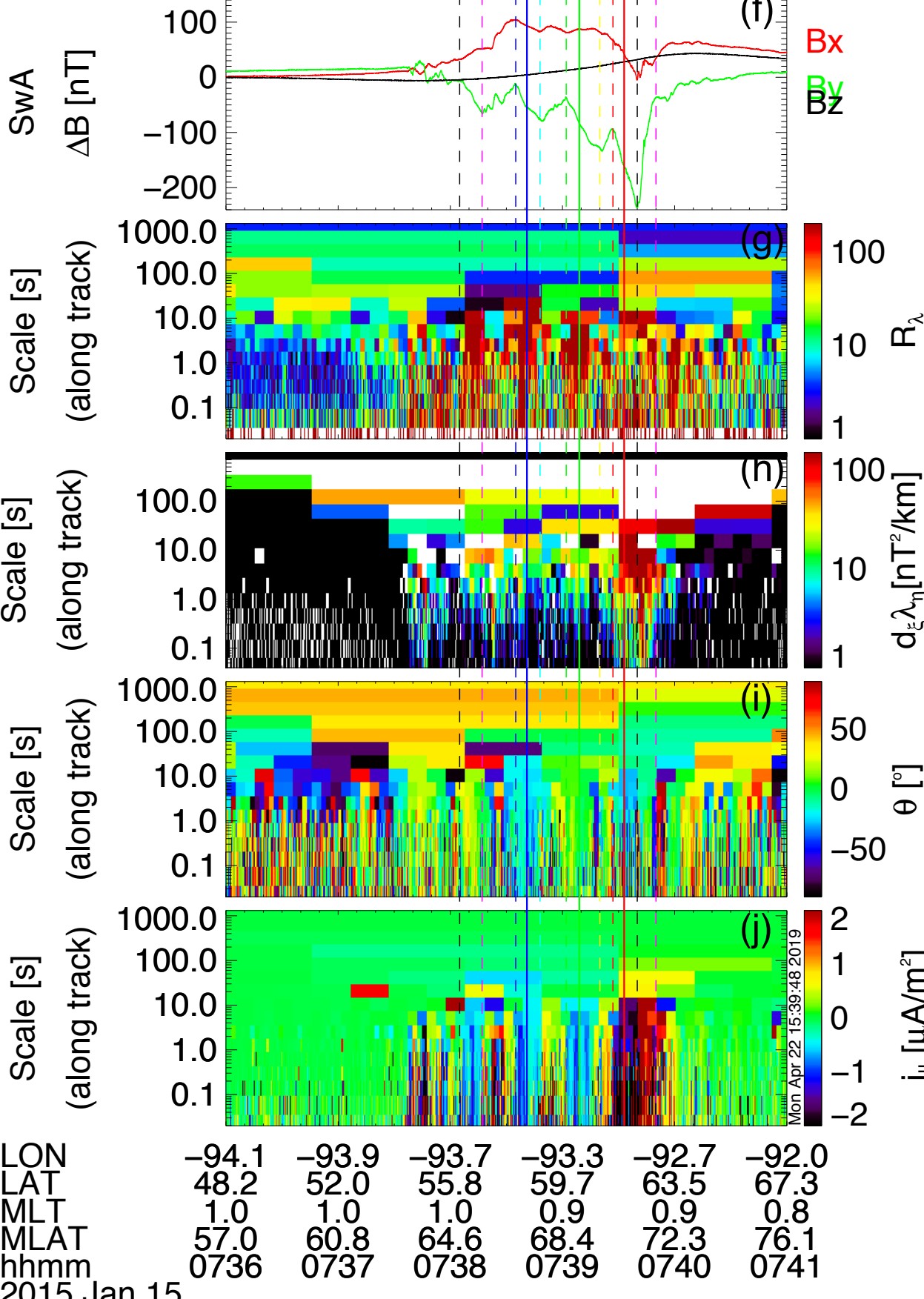

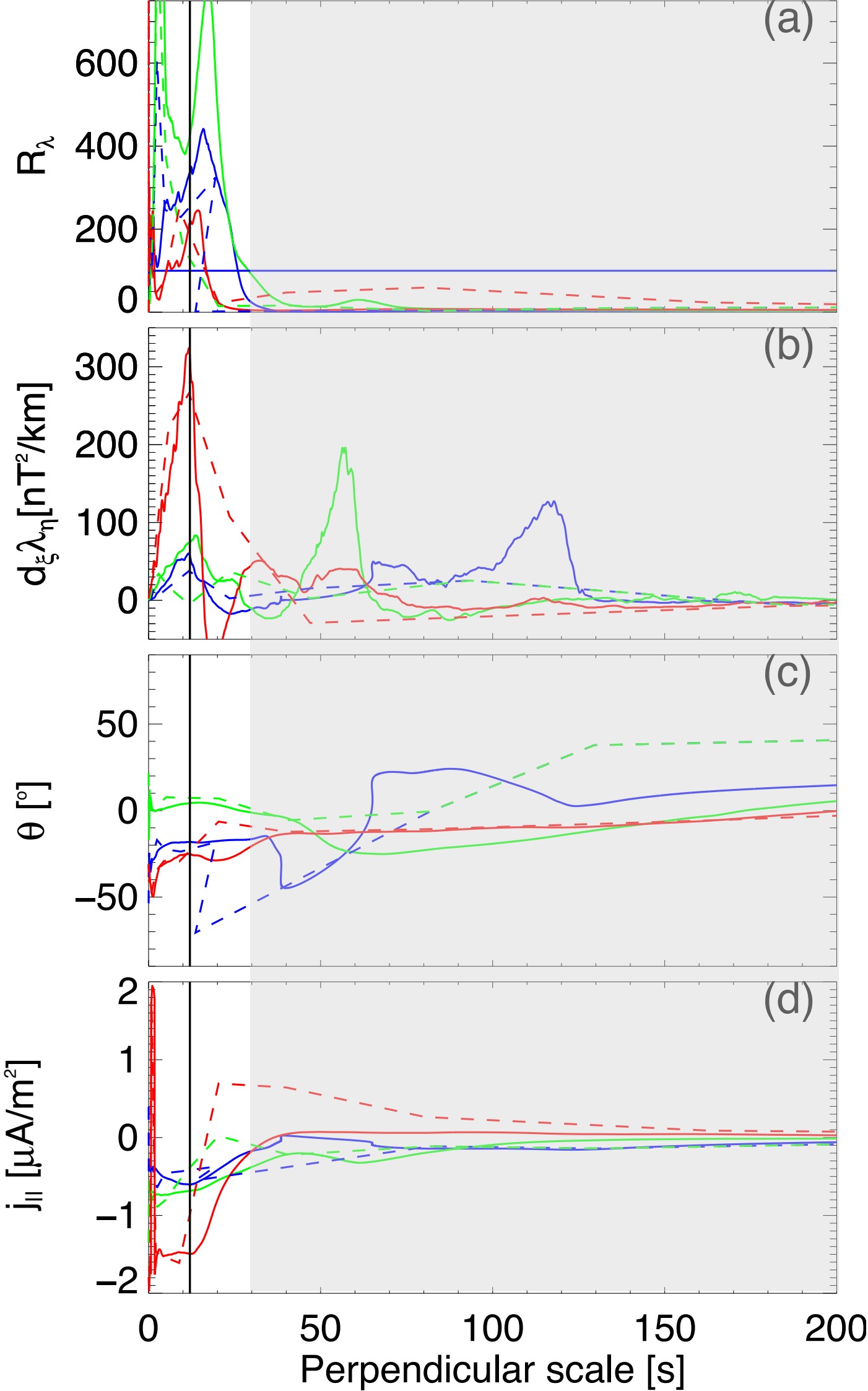

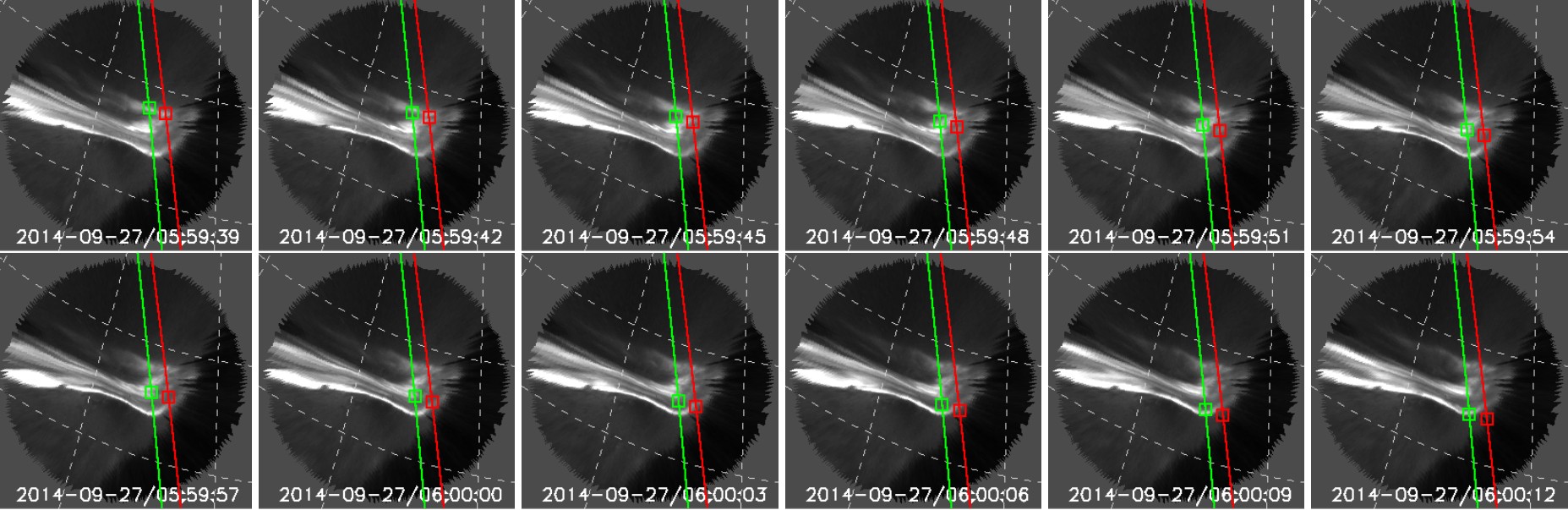

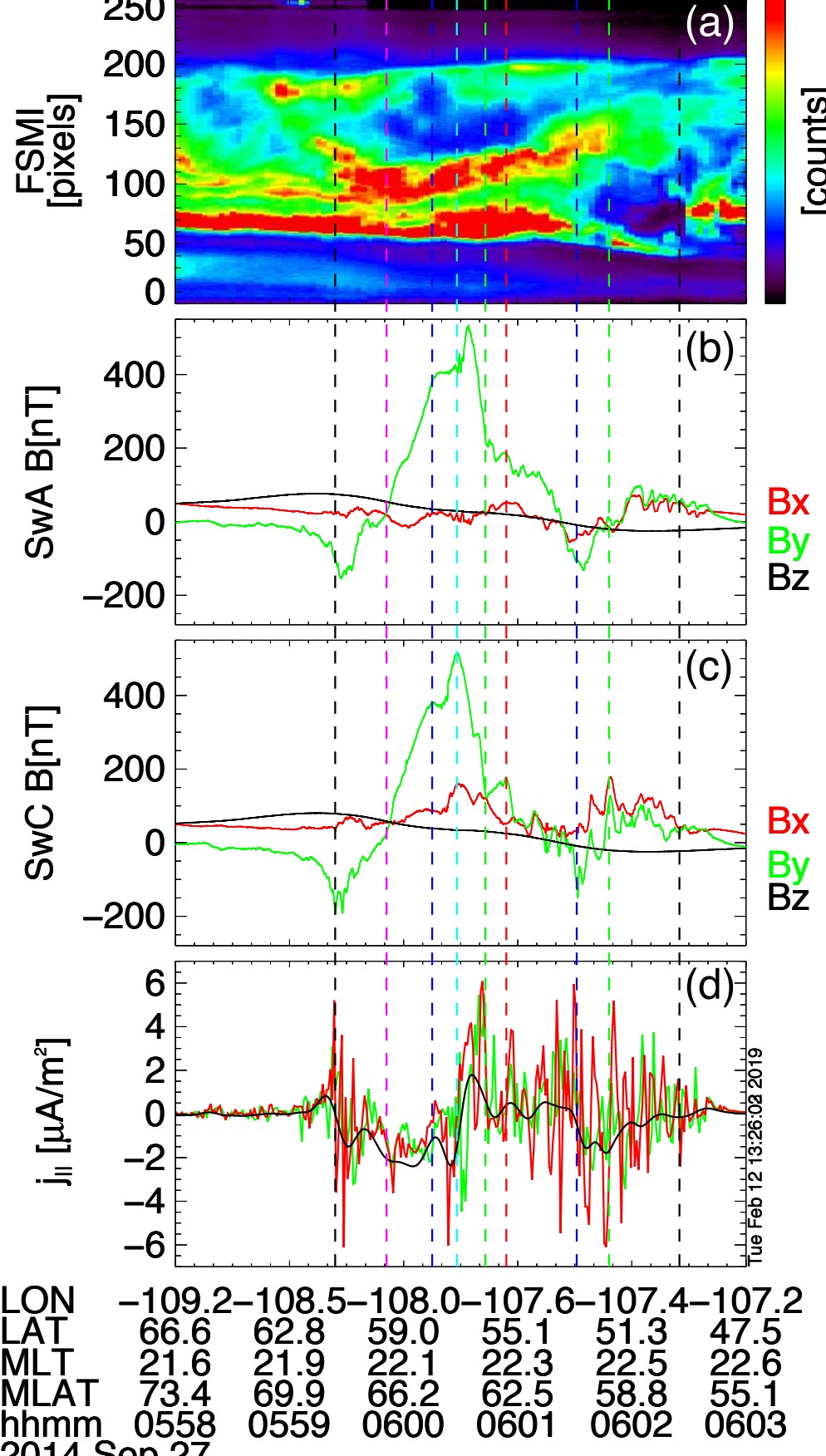

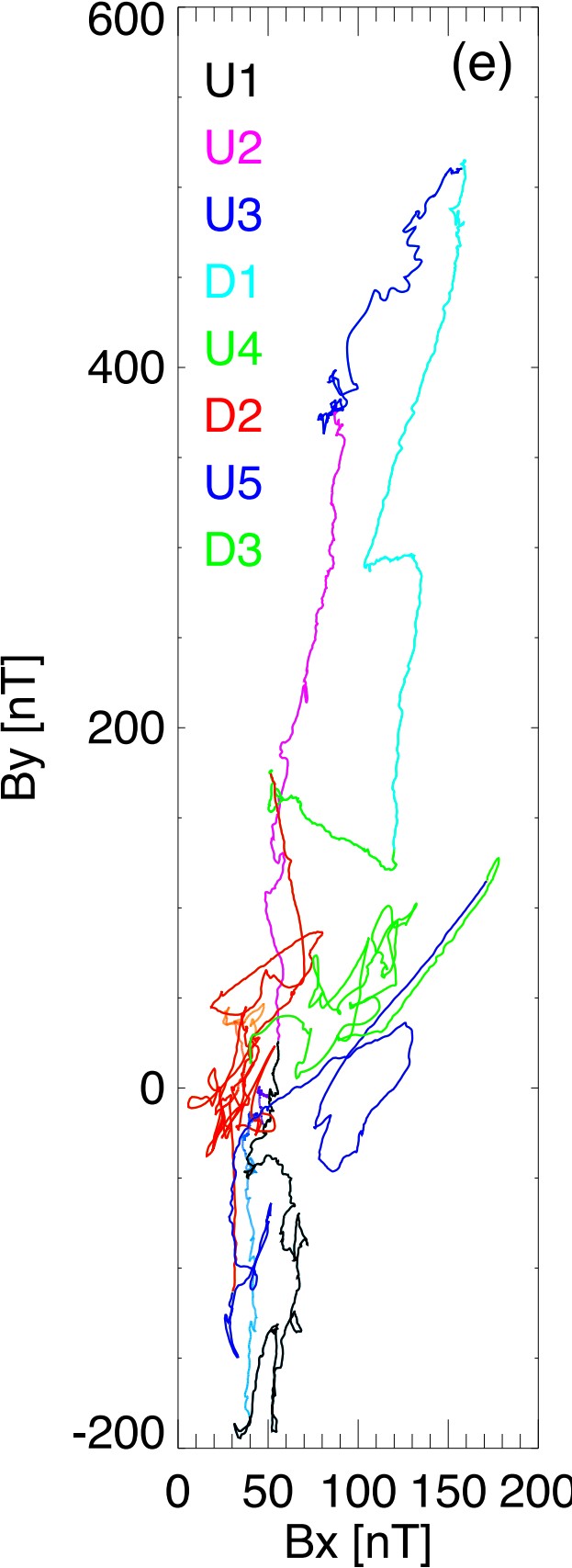

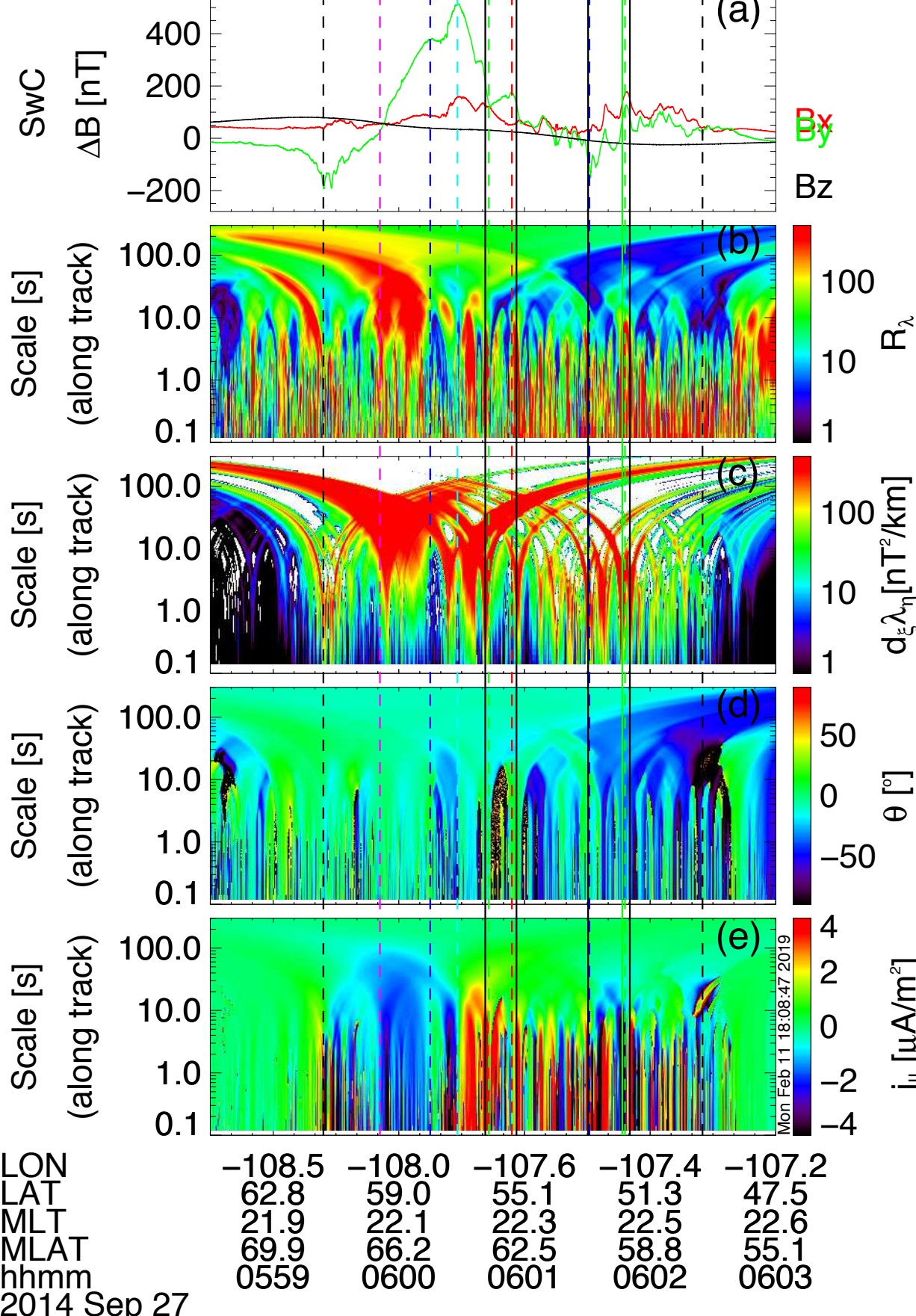

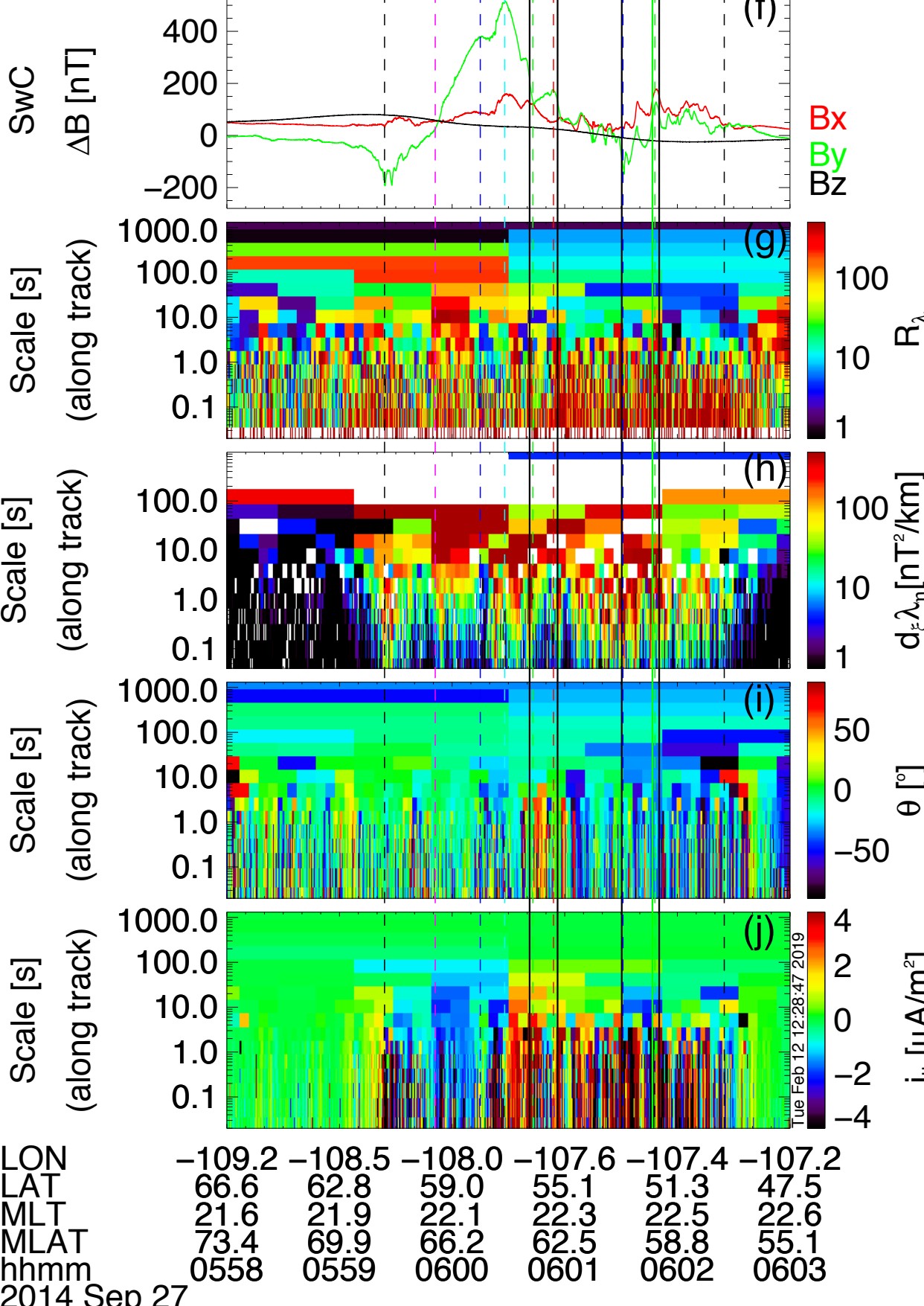

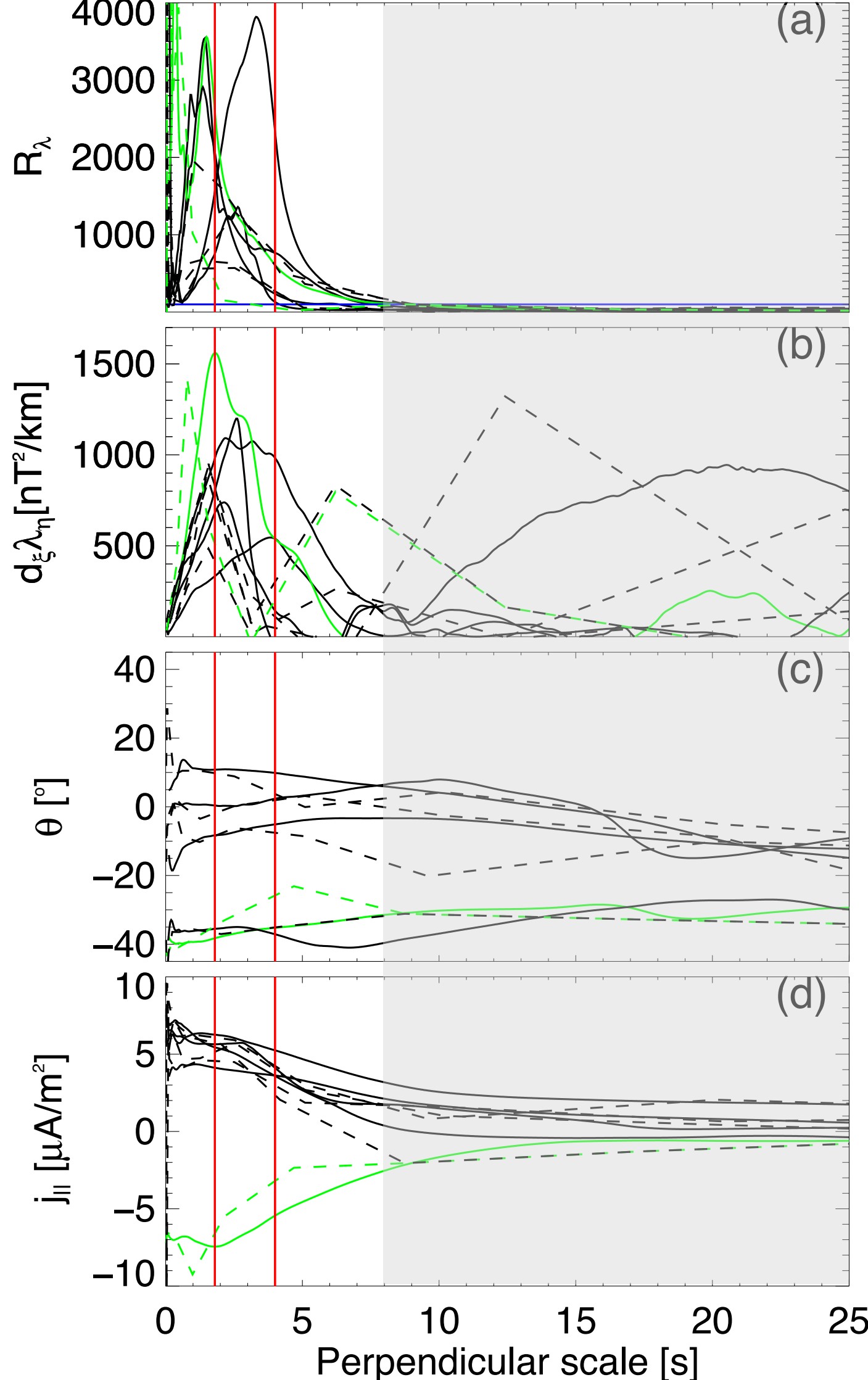