# Peer review of "Multiscale estimation of the field-aligned current density"

_Annales Geophysicae, 2018_

## Referee Comment (RC1) · Anonymous Referee #1 · 13 Aug 2018

Summary This manuscript presents a novel technique for assessing field-aligned currents across a range of scales, extending upon previous work present by Bunescu et al. [2015]. That previous work applied minimum variance analysis in sliding windows across a range of scales to determine the planarity and orientation of field-aligned currents. Using this technique, the field-aligned current density for each scale is determined by calculating the current from the gradient of the maximal variance magnetic field perturbation. While there is merit in this idea, the described technique does not sufficiently address the non-orthogonality of the scales used, which limits this techniques usefulness and how the results may be interpreted. As such, it is my recommendation that the authors revise this.

I would note that the manuscript does provide a good level of detail in relatively acces-

sible language and notation, which is a credit to the authors.

Specific comments There is much potential merit in the analysis technique described. However, in my opinion there is a potential underlying flaw that drastically limits the usefulness of this technique; the scales examined are not independent. As described, the minimum variance technique is applied to a collection of increasing scales by simply varying the window length over which the analysis is performed. As such, this does not isolate fluctuations on these scales and there is potential for the scales to 'bleed' into one another. The manuscript does discuss this in a limited fashion, noting that the scales are not orthogonal, however this does not go beyond a discussion. As a result, the calculated FACs at each scale are comparable to the total FAC, particularly at the smaller scales and the sum of the FACs across all scales is not the total FAC.

It is not clear to me exactly how to address this. The technique described in Bunescu et al. [2015] potentially enables different scales to be determined at different times by determining local maxima in delta_w Lambda_max, so some iterative process which identifies the relevant scales, filters the data at those scales, then runs the minimum variance analysis on those may be appropriate. Alternatively, filtering at a select number of scales will remove some of the 'bleed' between scales. These additions will not remove the non-orthogonality problem (discrete wavelet analysis or similar would be needed for that), band-pass filtering to attempt to isolate given scales should improve the results of the current calculation and remove the need to apply the weighting functions.

I note that in order to attempt to correct for the issue of the total multi-scale FAC, the manuscript describes three ways to weight the data: either taking the mean of the FACs across all scales; or multiplying by either the window width or one over the window width. These are somewhat contradictory to the aims of the paper as they either equally weight all scales or weight the to the larger or smaller scales. However, the principle of this analysis is to determine the most important scales. I believe that by applying the appropriate filtering, the need for these weightings will be removed.

Technical comments Figures should have panels labelled. While the panels are described in the captions, none are actually labelled.

The figures all appear to be fairly low resolution. For multi-panel figures, this makes them hard to examine in detail. Please provide higher resolution figures.

U1, 2 etc. are not labelled in Figure 4.

Figures 5, 9 and 13 all have a mis-labelled Y-axis in the top left plot (this should be "Magnetic Field (nT)" or similar)

The caption for Figure 6 does not described the coloured traces. Furthermore, the dashed lines only appear to be in two panels.

Each hodogram is missing the label for the Y-axis

In general, the description of the MSMVA panels in the text should be improved – it is somewhat hard to follow e.g. panel 9e4 etc. I would recommend unique letters for each panel.

P1. Line 11 - the abstract notes that the multiscale FAC is compared with input data and Swarm data, but gives no indication of how good or bad the comparison is.

P1. Line 17 – while I agree that solar wind-magnetosphere coupling is a key driver, there is an element of ionospheric feedback into the system which should not be ignored.

P1. Line 24 – above the ionosphere, one tends to measure magnetic perturbations due to the in-situ field-aligned currents rather than the ionospheric Pedersen currents

P2. Line11 – I suggest you reword this – it reads as though the maximum width was around 400-500 m but the average was greater than that. I believe you mean the peak of the distribution was 400-500 m

P2. Lines 21-32 – please be clear as to whether these scales are in-situ, in which case

the height they were measured is important, or mapped to some common altitude

P.18 Line 8 – You suggest that your technique is useful for comparing SwA and SwC data, but do not then go on to make this comparison. It would be interesting to see that (or remove this comment).

P.19 Line 14 – is this event a unipolar or multi-polar event from Wu et al. [2017]

The authors may also be interested in a study by Peria et al [2000, doi: 10.1029/GM118p0181] who examined used MVA to statistically examine auroral zone crossings by FAST.

---

## Referee Comment (RC2) · Anonymous Referee #2 · 14 Aug 2018

The authors describe a technique which aims to disentangle the characteristics of FACs present in the auroral region at multiple scales. The scheme is a development of earlier work by Bunescu et al. (2015) and uses analysis of pre-processed magnetic field data with sliding windows at multiple scales to conduct analysis on observed disturbances in the time domain, using MVA to calculate characteristics such as current sheet orientation and to ultimately identify the relative contributions of various scales to make up a FAC system. The system is tested first on simple simulated FACs and then on real data observed using the Swarm satellite mission.

General Comments

The system performs remarkably well when tested on simulated data and I believe that the development path is a promising one – in particular the time domain nature

of the method promises the potential of improved accuracy vs. frequency domain approaches. However when operating in the time domain, it is vitally important to somehow separate signals at various scales e.g. by zero-phase band-pass filtering. At the very least this direction should be explored and the results reported on. As things stand and as evident in Figures 1 and 2, even when running the system on simple test inputs, without appropriately suppressing large scales, small-scale parameters such as variance, dB etc. will be dominated by whatever is happening at those large scales. This is evidenced by the large error in calculated FAC in Figure 1 as the authors themselves admit to on page 11 lines 11-12 of the manuscript. When real data with multiple scales is analysed (e.g. Figure 5) the FAC contributions at various scales appear to bleed into each other to such a degree that disentangling the contributions from various scales becomes very difficult.

The authors attempt to correct for this as far as FAC density calculations are concerned by using weighting factors. In my opinion this can be potentially dangerous as then there is a risk of pre-supposing assumptions. Ultimately, we still do not know which scales are relatively the most important – this is what the methodology is designed to find out. By forcing weightings on scales there is a risk of the method presupposing its own conclusions. It is true that with the weighting factors the system does a good job of reconstructing the observed time series – however the aim is not to reconstruct the time series but to decompose them in a way that reflects the truth.

I would thus recommend a revision of the manuscript where some way of separating the scales, perhaps by selective zero-phase band-pass filtering, is carried out with the results reported on. I believe that simple additions such as these may greatly improve the system's capacity and look forward to seeing the results of the developments.

Specific comments:

Page 2 lines 3-4 "median of the scale distribution around 230 m in the range of fine and small scale auroral arcs (10 m – 1 km)." – please specify where the 10 m to 1

km numbers are from, since they do not appear to be present in the reference (the minimum scale in the reference is 70 m).

Page 2 line 7 "large sampling frequency difference, maximum at about ∼25 Hz for TV and ∼ 0.3 Hz for ASI" – for clarification it would be sensible to add something to the effect of "as any arcs which did not exhibit quasi-stationarity at the exposure timescales would likely have had their optical signals smeared and integrated to appear as larger-scale structures".

Page 2 lines 12-13 "Overall, the results of all these studies indicate a rather continuous scale spectrum" it may be better to write "Overall, the results of all these studies together indicate a rather continuous scale spectrum" since some of the individual studies alone certainly do not seem to indicate that!

Page 2 lines 23-24 "Karlsson and Marklund (1996) found a median scale of about 4.6 km for the diverging electric fields observed by Freja" – please specify where the referenced paper mentions that number, since I am unfortunately unable to find it

Page 4 lines 1-2 "Gillies et al. (2015) pointed out that the single-spacecraft FAC density provides better identitifaction of the boundaries of auroral patches." – Is this "better" as compared to the dual-spacecraft FAC density? Please specify. As I understand this paragraph advocates for the need to analyse multiple scales including small scales which may not be well-captured by the dual-spacecraft method or indeed by the 1 Hz single-spacecraft FAC product. In this context a valuable reference to add would be Miles et al. (2018) who studied an intense discrete auroral arc crossing observed by Swarm and e-POP, the paper vividly demonstrates the limitations of both the dual spacecraft FAC density product and the small-tilt assumption when analyzing small-scale auroral structures.

Page 5 lines 26-27 the authors state the Equation (3) was used for many single-spacecraft missions but then proceed to give only 1 reference (to Freja data). Please either amend the wording (e.g. "Equation (3) was used successfully to obtain partial

estimates of the FAC density for some single-spacecraft missions") or else add more references to where the single-spacecraft product was used (e.g. Swarm).

Page 5 line 21 "Thus, the cross-track separation defines the lower limit of the FAC scales" – authors should specify that the lower scale limit is in the cross-track direction. The scale limit in the along-track direction is determined by the inter-spacecraft along-track separation and the degree of quasi-stationarity of those FACs.

Page 15 lines 5-6 (also Figure 4) – the origin of the time shift is explained fine but it is visually difficult to compare the two results with the time shift in place. I would recommend factoring it out so that the two estimates are overplotted directly and the differences can be more clearly seen.

Technical corrections:

Page 8 line 4 please close the 2nd bracket

Page 15 line 27 "slightly" Please replace "associated to" to "associated with" (e.g. in lines 22-23 in page 1)

Figures:

Please label the plots in the figures (a1, b1 etc… are described in the figure captions but are not actually labelled by the sides of the plots). In Figures 1, 2, 5, 9 and 13 planarity (on the y-axis) appears to be mis-labelled as 'scale' in the plots.

References

Miles, D. M., Mann, I. R., Pakhotin, I. P., Burchill, J. K., Howarth, A. D., Knudsen, D. J., … Yau, A. W. (2018). Alfvénic dynamics and fine structuring of discrete auroral arcs: Swarm and e-POP observations. Geophysical Research Letters, 45. https://doi.org/10.1002/2017GL076051

ANGEOD

Interactive
comment

---

## Author Comment (AC1) · 30 Nov 2018

**1 Summary**

This manuscript presents a novel technique for assessing field-aligned currents across a range of scales, extending upon previous work present by (Bunescu et al., 2015). That previous work applied minimum variance analysis in sliding windows across a range of scales to determine the planarity and orientation of field-aligned currents. Using this technique, the field-aligned current density for each scale is determined by calculating the current from the gradient of the maximal variance magnetic field perturbation. While there is merit in this idea, the described technique does not sufficiently address the non-orthogonality of the scales used, which limits this techniques usefulness and how the results may be interpreted. As such, it is my recommendation that the authors revise this.

MSMVA technique is a time domain analysis based on the procedure described in (Bunescu et al., 2015). Besides the multiscale information on planarity and orientation, the main point of MSMVA is the determination of characteristic scales of the measured FAC elements and their locations, provided by $\partial_w \lambda_\eta$. In this manuscript, we use the MSMVA analysis to compute the local FAC density and we find consistent results for the synthetic generated FACs and also for the Swarm events analyzed in the manuscript. We agree with the referee that there is room for improvement, but we plan this for a future study. The main criticism from both referees is related to the non-orthogonality of the basis functions. As explained below, we regard this as a feature of the technique and not as a week point. We note that various other spectral methods were already tried in other studies, e.g. Stasiewicz and Potemra (1998), with applications for wave field characterization, e.g. Alfven waves. Stasiewicz and Potemra (1998) analysis is based on the decomposition using orthogonal wavelets. As far as we checked, this paper is cited mainly for the small-scale FACs observations by Freja and not for further application of the orthogonal-wavelet decomposition of the FAC density. Nevertheless, in a future study, we plan to complement the MSMVA analysis with the spectral type of analysis (e.g. (Stasiewicz and Potemra, 1998)) in order to better characterize both the quasi-stationary (spatial FAC structures) and the temporal-structures (e.g. wave phenomena). As of now, we just note that an orthogonal basis function may be intrinsically associated with issues that limit its use to derive the scale and location of FAC elements - as suggested in our study by the results derived with the (almost orthogonal) logarithmic sampling scheme.

I would note that the manuscript does provide a good level of detail in relatively accessible language and notation, which is a credit to the authors.

**2 Major adjustments**

The work on the revision of the manuscript prompted additional consistency checks, which helped us to understand better the results and limitations of the multiscale analysis, as well as to improve the computation and the interpretation of the MSMVA parameters. Among those, perhaps the main improvement is that $\partial_w \lambda_\eta$ does not simply provide the normal scale of the FAC sheet, but the crossing length along spacecraft track. In order to obtain the FAC thickness, one has to project this length onto the FAC normal. The correction of the scale information is done for all MSMVA quantities and for both scale sampling schemes, by assuming that the spacecraft track is essentially in northward direction, i.e., along the $x$ axis of the MFA system. Basically, this correction is needed because the analysis is done in the MFA $(x, y)$ frame, whereas thickness is defined in the FAC sheet $(\xi, \eta)$ frame. In order to explain this behavior we added a paragraph at the end of section (2.3). In this paragraph we explain also that the amplitude of $\partial_w \lambda_\eta$ and the local FAC density have to be corrected as well for the dependence on thickness instead of the parameter $w$.

Following this adjustment, we decided to include also a proper parametrization of the orientation in the synthetic data. The new FAC system from section (3.1) includes now orientations of 0° and 40° for the FD and FU FACs, respectively. The new results show the scale correction in all quantities. The amplitude corrected $j_\parallel$ and $\partial_w \lambda_\eta$ show now the same amplitudes for

the two FACs. The scale and amplitude correction of the multiscale parameters has also impact on the application of MSMVA to Swarm events. While the amplitude of $j_\parallel$ was correctly computed for Swarm events, the scale correction was not applied in the submitted manuscript. Thus, all profiles of MSMVA quantities (Figures 6, 10, and 14) now contain correction of scales (i.e., adjustments in the respective abscissa values) and the various scales and orientations mentioned in the text for all Swarm events (sections (4.2), (4.3), and (4.4)) were changed accordingly. Note that the previous work by Bunescu et al. (2015) did not include tests on inclined FAC structures. Nevertheless, the results of Bunescu et al. (2015) are still fine since the analysis was illustrated with east-west aligned synthetic FACs and also the application to measured data was done on essentially east-west aligned aurora.

Following the referees' comments we looked closer at the scale weightings of the local FAC density to obtain a global FAC density estimate. The conclusion is that the weighting concept is not mature enough to be applied to measured data. One option would have been to still keep the weightings for the synthetic data where we have a good reference FAC density (input FAC density) to compare with global FAC density (output of the weighting). However, such a setup, with weightings only for synthetic data, might still confuse the reader, who could wonder why the wighting is not applied also to measured data. We thus agree with the referees that the weighting is better suited to further work, oriented towards the reconstruction of the total FAC density. A paragraph was added at the end of section (2.4) to explain that we cannot simply integrate over scales to obtain a global FAC density and we also pointed out that proper scale weighting is needed because of the lack of orthogonality of the basis functions. As a consequence of removing the weighting, section (2.5) was removed and the manuscript was cleaned from the references to the weighting of FAC estimates. The associated changes are indicated by the green cuts through text in all sections. The bottom panels in Figures 1 and 2 were removed. Also the two bottom panels in the left and right plots of Figures 5, 9, and 13 were removed.

Following the removal of the global FAC estimates (weighted FAC estimates) we compare the local multiscale FAC density, at the scale given by $\partial_w \lambda_\eta$ (assumed to dominate at a specific location), with single- and dual-spacecraft FAC estimates obtained at the same time/position. We added paragraphs to each section showing Swarm observations (sections (4.2), (4.3), (4.4)), where we compared the FAC estimates and quantified the differences percentage wise. For the synthetic data (sections (3.1), (3.2)) we compared the local multiscale FAC density with the input FAC density. We also updated two paragraphs from the conclusion section to indicate the agreement between the different FAC estimates and also the consistency of the linear and logarithmic scale samplings.

The previous version of the manuscript included two FAC estimates based on the dual-spacecraft methods, FD and LS FAC estimates. A more careful analysis showed that both FD and LS show basically similar FAC density estimates. The differences were caused, essentially, by the time tags assignment. Following discussion with the coauthors we decided to keep only the FD dual-spacecraft estimate in order to reduce the amount of information and panels in Figures 4, 8, and 12. Moreover, at the moment the dual-spacecraft LS estimate is not publicly available and the application of the dual-spacecraft LS technique to Swarm is not published. The associated changes are indicated by the blue cuts through text in all sections.

The revised manuscript includes the mapping of the optical frames to the geographic frame. This change leads to adjustments in the interpretation of the optical information. Thus, the paragraphs describing the optical data were updated accordingly for each Swarm event. For the event on 17 February 2015 the mapped optical data show that actually the southward auroral structure looks curled. For the event on 15 January 2015 we now clearly see a large scale auroral structure inclined by about 20° towards south, and small embedded auroral structures of different inclinations, rather east-west aligned. Also, for the last Swarm event the optical data are now more consistent with the results of the MSMVA analysis.

Section (2.6) was integrated into the discussion section, since we do not actually show comparisons with spectral techniques.

Finally, we found an error in the computation of $\partial_w \lambda_\eta$ for the logarithmic scale sampling scheme. This was caused by using the same code as for the linear scheme, where we have a constant discretization in the scale array, $dw$=const, whereas for the logarithmic scheme this is not the case. The results of the linear and logarithmic schemes show now a better consistency.

Various other corrections of the text were needed to better explain certain features or correct small errors. We hope that the revised manuscript is clearer, follows better the target of this work and avoids confusion on side subjects.

**3  Specific comments**

There is much potential merit in the analysis technique described. However, in my opinion there is a potential underlying flaw that drastically limits the usefulness of this technique; the scales examined are not independent. As described, the minimum variance technique is applied to a collection of increasing scales by simply varying the window length over which the analysis is performed. As such, this does not isolate fluctuations on these scales and there is potential for the scales to 'bleed' into one another. The manuscript does discuss this in a limited fashion, noting that the scales are not orthogonal, however this does not go beyond a discussion.

The model functions implicitly employed to represent the magnetic field measurements are piece-wise linear functions of a certain length $w$, interpreted as the scale of the underlying current structure. The corresponding FAC density profile is a step function of the same width $w$, and centered at the same reference time $tcen$. This approach is compatible with established FAC estimators based on finite differencing. Actual magnetic profiles in the auroral zone are quite similar to these underlying piece-wise linear model functions, at least closer than perfectly smooth functions such as the ones employed for producing the synthetic data in section (3) (which are preferred there because of analytic tractability). Hence, we assume that our FAC scalogram performs actually better on real data than on the synthetic examples. Nonzero correlations among different piece-wise linear model functions lead to the non-orthogonal behavior criticized by both referees. The overall implications, however, depend on the particular subset of model functions associated with the chosen sampling scheme: (a) If for a given scale $w$ all available center times $tcen$ are used, model functions with neighboring $tcen$ are strongly correlated, resulting in a highly redundant and very non-orthogonal representation. This scale sampling scheme we call "linear". (b) If for a given scale $w$ the chosen center times $tcen$ are separated by the scale $w$, the model functions are only weakly correlated, resulting in a representation that is much less redundant and closer to orthogonality. This scale sampling scheme we call "logarithmic". The underlying logic is the same as for the Haar wavelet transform. By comparing the results of linear versus logarithmic scale sampling for synthetic data, one finds that localization of center time/location and scale is more accurate with the linear sampling scheme. In logarithmic sampling, the center location of a current structure is heavily constrained by the scale $w$ that thus effectively represents the uncertainty of the $tcen$ (note also the uncertainty relation in wavelet analysis). Here our emphasis is on constraining FAC scales and center locations using a visualization tool, not on a full reconstruction of the FAC profile, thus we prefer to use a highly redundant set of model functions instead of an orthogonal and thus non-redundant one. Since the synthetic data are smooth profiles, and the scales are the widths of Gaussian profiles, we cannot expect that the piece-wise linear model functions identify the parameters perfectly. We updated the discussion section of the manuscript with this paragraph

As a result, the calculated FACs at each scale are comparable to the total FAC, particularly at the smaller scales and the sum of the FACs across all scales is not the total FAC.

At each scale the FAC density, as well as the MVA analysis, is applied on the detrended magnetic field perturbation obtained by removing the average signal on the respective window. Practically, at a fixed window/scale we compute the MVA and FAC density on the residual perturbation obtained by extracting a running window average. This procedure should partially remove the background / the large scale trend. We agree that one cannot completely and uniquely separate between scales. This is more difficult when the FAC elements have comparable intensities and thicknesses which results in comparable perturbations around the neighboring scale. The resolution of the method in terms of scale identification depends on the characteristics of the superposed FAC structures, e.g. the ratio of the intensities, and thicknesses, relative orientations, and localizations. In the auroral region, there are larger differences between the large/mesoscale FACs and the superposed small-scale FACs, both in terms of intensities and scales. Thus, we expect that our analysis is appropriate to separate such scales.

In order to better illustrate the technique with synthetic data, we adjusted equations (9) and (11) in section (3). The previous equations did not include properly the parametrization of the orientation. One cannot find the inclination when working in the $(\xi,\eta)$ frame of the FAC sheet. We now rotate $B_\eta$ to compute the components $B_x$ and $B_y$ which are subject to MVA. The FAC structure included in section (3.2) emphasizes the use of the technique for visualization of the local FAC density and of the other characteristics of the FACs. This FAC structure consists of a large-scale double FAC system, with fwhm thickness of about 117 km ($\sigma_\perp$=50 km) for each FAC sheet, with superposed small scale FACs of ∼11.7 km ($\sigma_\perp$=5 km) each. The small-scale FACs are organized in two sandwiches of 3 sheets and are centered on the large-scale FACs. The orientation of the large scale FAC system is $\theta_l^{(1)}$=0°/$\theta_l^{(1)}$=40° for FD/FU FACs, similar to the FAC system in section (3.1). For simplicity the

orientation of all small-scale FACs is $\theta_l^{(1)}=0°$. To show more quantitative results we included profiles (vertical cuts) trough the spectrograms. Figure 2 (panel (p)) indicates that the local FAC density at the large and small scale is roughly consistent with the input FAC density. The local FAC density for FD/FU is $4\mu A/m^2$/-4.5$\mu A/m^2$ and indicate a good agreement with the input of $\pm 5\mu A/m^2$. For the small scale FACs centered on FD/FU we have -12$\mu A/m^2$/-16$\mu A/m^2$, which is roughly consistent with expected input FAC density of $\sim$-16$\mu A/m^2$/-12$\mu A/m^2$.

It is not clear to me exactly how to address this. The technique described in (Bunescu et al., 2015) potentially enables different scales to be determined at different times by determining local maxima in $\partial_w\lambda_{max}$, so some iterative process which identifies the relevant scales, filters the data at those scales, then runs the minimum variance analysis on those may be appropriate. Alternatively, filtering at a select number of scales will remove some of the 'bleed' between scales. These additions will not remove the non-orthogonality problem (discrete wavelet analysis or similar would be needed for that), band-pass filtering to attempt to isolate given scales should improve the results of the current calculation and remove the need to apply the weighting functions.

We also think of an iterative algorithm, but for the purpose to reconstruct the observed FAC density. We plan to iteratively identify the most intense planar FAC structures, that can be large or small scale FACs, as indicated by $R_\lambda$ and $\partial_w\lambda_{max}$, and apply MSMVA to the successive residuals obtained by separating the identified FACs. However, the problem might not be uniquely determined, and before further effort to develop the technique and reconstruct the FAC signature, we plan to gain more experience with its use and apply it to more events.

The filtering using zero-phase band-pass filtering comes with other difficulties. We agree that the filtering of the large scale R1/R2 current system would improve the results of the MSMVA analysis at the small- and mesoscale range. But at the same time an unappropriate filtering might introduce additional features. This is likely to be the case even with a good filtering scheme and the reason why we prefer to apply the method on the perturbation obtained only by removing a model magnetic field. Reprocessing/filtering (zero phase) is essentially a projection and distorts the interpretation. It may be preferred on theoretical grounds because of a seemingly more unique interpretation but should depend on the type of filter and then again on the model functions.

I note that in order to attempt to correct for the issue of the total multi-scale FAC, the manuscript describes three ways to weight the data: either taking the mean of the FACs across all scales; or multiplying by either the window width or one over the window width. These are somewhat contradictory to the aims of the paper as they either equally weight all scales or weight the to the larger or smaller scales. However, the principle of this analysis is to determine the most important scales. I believe that by applying the appropriate filtering, the need for these weightings will be removed.

Regarding the pre-processing of the data by band-pass filtering, this would imply an apriori selection of a specific scale range and a bias to the result. Admittedly, the weighting scheme has a similar problem - imperfectly cured for the time being by using a couple of different weights. Initially we thought to change the title of section (2.5) to "Global FAC estimation derived from the multiscale FAC" to partially remove the confusion on the purpose of this global weightings, and to add a paragraph to stress that we do not reconstruct the FAC density but aim for a quantity to qualitatively compare with the other single and dual-spacecraft methods which are not providing deconvoluted information on the FAC density. After more discussions with the coauthor we decided to remove completely the weightings from the paper. As already mentioned above, this change resulted in a few changes of the manuscript, indicated by the green color.

**4 Technical comments**

Figures should have panels labeled. While the panels are described in the captions, none are actually label-led.

All panels are now labeled.

The figures all appear to be fairly low resolution. For multi-panel figures, this makes them hard to examine in detail. Please provide higher resolution figures.

We now provide high resolution eps figures for the linear sampling scheme for both synthetic and Swarm data, whereas for the logarithmic sampling scheme only for the Swarm events. Saving the plots for the logarithmic scheme as eps for the synthetic data turns out to highly affect the discrete character of the respective results. Therefore, in this case we saved the figures as png that does not alter the results.

U1, 2 etc. are not labeled in Figure 4.

We included the U and D labels in the hodogram representation, right panel of Figures 4, 8 and 12. For the other figures, e.g. multipanel figures, these labels would complicate the layout too much. It is difficult to add them in the multipanel figures because of limited space. We explain in the text that the color of the interval is similar to the color of the left vertical line.

Figures 5, 9 and 13 all have a mis-labelled Y-axis in the top left plot (this should be "Magnetic Field (nT)" or similar)

We corrected this error.

The caption for Figure 6 does not described the coloured traces. Furthermore, the dashed lines only appear to be in two panels.

In the submitted manuscript we included the dash line only in the FAC density (panels d of Figures 6, 10, and 14), whereas the dash line in panels (a) of the same Figures was indicating an arbitrary reference level of planarity, $R_\lambda$ =100. We now indicate this reference level by a solid blue line. We added also the other dash traces indicating the results for the logarithmic scanning in all panels of Figures 6, 10, and 14. The color of the traces indicates that the respective profile is taken around the center of the FAC indicated by a solid line of the same color in spectrograms (Figures 5, 9, 13). For instance, we used black and magenta for the U1 and D1 FACs in Figures 5 and 6.

Each hodogram is missing the label for the Y-axis

Previously this axis was removed to reduce the width of Figures 4, 8, and 12 , to fit to one column. We now included the Y axis.

In general, the description of the MSMVA panels in the text should be improved – it is somewhat hard to follow e.g. panel 9e4 etc. I would recommend unique letters for each panel.

We followed the suggestion and changed to unique letters. We did not develop much the discussion about MSMVA results because the basics and some applications (mainly $\partial_w \lambda_{\max}$ quantity) were shown in (Bunescu et al., 2015). By including the inclination in the synthetic FACs (section (3.1)) we now extended the description of the results also for the respective numeric experiment, where we can control the input. For synthetic FAC systems one can control the relative orientation between FAC elements.

P1. Line 11 - the abstract notes that the multiscale FAC is compared with input data and Swarm data, but gives no indication of how good or bad the comparison is.

We now make quantitative comparison between the different FAC density estimates for both synthetic and Swarm data. We compare the local multiscale FAC density with the input synthetic data, as well as with the single- and dual-spacecraft FAC density Swarm products, at the same position/time. As already mentioned above, we improved both the synthetic data and Swarm events sections by adding paragraphs where we make these comparisons and derive the differences between the FAC density estimates. We note that there are also differences between the dual- and single-spacecraft Swarm products even for events which presumably fulfill the assumptions, e.g. the event from 2015-02-17 with planar FACs.

P1. Line 17 – while I agree that solar wind-magnetosphere coupling is a key driver, there is an element of ionospheric feedback into the system which should not be ignored.

Adjusted: "subject to ionospheric feedback" added to the sentence and S-M extended to "S-M-I" in the next sentence.

P1. Line 24 – above the ionosphere, one tends to measure magnetic perturbations due to the in-situ field-aligned currents rather than the ionospheric Pedersen currents

Adjusted. Now the sentence reads: "While above the ionospheres one measures the magnetic perturbation of the field-aligned current (closed in the ionosphere mainly by the Pedersen current), the magnetic perturbation observed on ground is related mainly to the Hall component of the ionospheric current

P2. Line11 – I suggest you reword this – it reads as though the maximum width was around 400-500 m but the average was greater than that. I believe you mean the peak of the distribution was 400-500 m

Adjusted: Trondsen and Cogger (1997) addressed the scale distribution of the black aurora, found to peak around 400-500 m, with an average of 615 m (range between 200 m and 1 km).

P2. Lines 21-32 – please be clear as to whether these scales are in-situ, in which case C3 the height they were measured is important, or mapped to some common altitude

All these scales are indeed mapped to the ionosphere - now made explicit in the text.

P.18 Line 8 – You suggest that your technique is useful for comparing SwA and SwC data, but do not then go on to make this comparison. It would be interesting to see that (or remove this comment).

We removed this comment. We show only the single spacecraft results without comparison.

P.19 Line 14 – is this event a unipolar or multi-polar event from (Wu et al., 2017)

Text was adjusted accordingly - "unipolar" added to the sentence.

The authors may also be interested in a study by (Peria et al., 2013) who examined used MVA to statistically examine auroral zone crossings by FAST.

We included this reference in the introduction section and briefly commented the results.

**References**

Bunescu, C., Marghitu, O., Constantinescu, D., Narita, Y., Vogt, J., and Blăgău, A.: Multiscale field-aligned current analyzer, Journal of Geophysical Research: Space Physics, 120, 9563–9577, https://doi.org/10.1002/2015JA021670, http://dx.doi.org/10.1002/2015JA021670, 2015.

5 Peria, W. J., Carlson, C. W., Ergun, R. E., McFadden, J. P., Bonnell, J., Elphic, R. C., and Strangeway, R. J.: Characteristics of Field-Aligned Currents Near the Auroral Acceleration Region: Fast Observations, pp. 181–189, American Geophysical Union (AGU), https://doi.org/10.1029/GM118p0181, 2013.

Stasiewicz, K. and Potemra, T.: Multiscale current structures observed by Freja, Journal of Geophysical Research: Space Physics, 103, 4315–4325, https://doi.org/10.1029/97JA02396, https://agupubs.onlinelibrary.wiley.com/doi/abs/10.1029/97JA02396, 1998.

10 Trondsen, T. S. and Cogger, L. L.: High-resolution television observations of black aurora, Journal of Geophysical Research: Space Physics, 102, 363–378, https://doi.org/10.1029/96JA03106, http://dx.doi.org/10.1029/96JA03106, 1997.

Wu, J., Knudsen, D. J., Gillies, D. M., Donovan, E. F., and Burchill, J. K.: Swarm Observation of Field-Aligned Currents Associated With Multiple Auroral Arc Systems, Journal of Geophysical Research: Space Physics, pp. n/a–n/a, https://doi.org/10.1002/2017JA024439, 2017JA024439, 2017.

---

## Author Comment (AC2) · 30 Nov 2018

**1   Summary**

The authors describe a technique which aims to disentangle the characteristics of FACs present in the auroral region at multiple scales. The scheme is a development of earlier work by (Bunescu et al., 2015) and uses analysis of pre-processed magnetic field data with sliding windows at multiple scales to conduct analysis on observed disturbances in the time domain, using MVA to calculate characteristics such as current sheet orien- tation and to ultimately identify the relative contributions of various scales to make up a FAC system. The system is tested first on simple simulated FACs and then on real data observed using the Swarm satellite mission.

**2   Major adjustments**

The work on the revision of the manuscript prompted additional consistency checks, which helped us to understand better the results and limitations of the multiscale analysis, as well as to improve the computation and the interpretation of the MSMVA parameters. Among those, perhaps the main improvement is that $\partial_w \lambda_\eta$ does not simply provide the normal scale of the FAC sheet, but the crossing length along spacecraft track. In order to obtain the FAC thickness, one has to project this length onto the FAC normal. The correction of the scale information is done for all MSMVA quantities and for both scale sampling schemes, by assuming that the spacecraft track is essentially in northward direction, i.e., along the $x$ axis of the MFA system. Basically, this correction is needed because the analysis is done in the MFA $(x, y)$ frame, whereas thickness is defined in the FAC sheet $(\xi, \eta)$ frame. In order to explain this behavior we added a paragraph at the end of section (2.3). In this paragraph we explain also that the amplitude of $\partial_w \lambda_\eta$ and the local FAC density have to be corrected as well for the dependence on thickness instead of the parameter $w$.

Following this adjustment, we decided to include also a proper parametrization of the orientation in the synthetic data. The new FAC system from section (3.1) includes now orientations of 0° and 40° for the FD and FU FACs, respectively. The new results show the scale correction in all quantities. The amplitude corrected $j_\parallel$ and $\partial_w \lambda_\eta$ show now the same amplitudes for the two FACs. The scale and amplitude correction of the multiscale parameters has also impact on the application of MSMVA to Swarm events. While the amplitude of $j_\parallel$ was correctly computed for Swarm events, the scale correction was not applied in the submitted manuscript. Thus, all profiles of MSMVA quantities (Figures 6, 10, and 14) now contain correction of scales (i.e., adjustments in the respective abscissa values) and the various scales and orientations mentioned in the text for all Swarm events (sections (4.2), (4.3), and (4.4)) were changed accordingly. Note that the previous work by Bunescu et al. (2015) did not include tests on inclined FAC structures. Nevertheless, the results of Bunescu et al. (2015) are still fine since the analysis was illustrated with east-west aligned synthetic FACs and also the application to measured data was done on essentially east-west aligned aurora.

Following the referees' comments we looked closer at the scale weightings of the local FAC density to obtain a global FAC density estimate. The conclusion is that the weighting concept is not mature enough to be applied to measured data. One option would have been to still keep the weightings for the synthetic data where we have a good reference FAC density (input FAC density) to compare with global FAC density (output of the weighting). However, such a setup, with weightings only for synthetic data, might still confuse the reader, who could wonder why the weighting is not applied also to measured data. We thus agree with the referees that the weighting is better suited to further work, oriented towards the reconstruction of the total FAC density. A paragraph was added at the end of section (2.4) to explain that we cannot simply integrate over scales to obtain a global FAC density and we also pointed out that proper scale weighting is needed because of the lack of orthogonality of the basis functions. As a consequence of removing the weighting, section (2.5) was removed and the manuscript was cleaned from the references to the weighting of FAC estimates. The associated changes are indicated by the green cuts through text in all

sections. The bottom panels in Figures 1 and 2 were removed. Also the two bottom panels in the left and right plots of Figures 5, 9, and 13 were removed.

Following the removal of the global FAC estimates (weighted FAC estimates) we compare the local multiscale FAC density, at the scale given by $\partial_w \lambda_\eta$ (assumed to dominate at a specific location), with single- and dual-spacecraft FAC estimates obtained at the same time/position. We added paragraphs to each section showing Swarm observations (sections (4.2), (4.3), (4.4)), where we compared the FAC estimates and quantified the differences percentage wise. For the synthetic data (sections (3.1), (3.2)) we compared the local multiscale FAC density with the input FAC density. We also updated two paragraphs from the conclusion section to indicate the agreement between the different FAC estimates and also the consistency of the linear and logarithmic scale samplings.

The previous version of the manuscript included two FAC estimates based on the dual-spacecraft methods, FD and LS FAC estimates. A more careful analysis showed that both FD and LS show basically similar FAC density estimates. The differences were caused, essentially, by the time tags assignment. Following discussion with the coauthors we decided to keep only the FD dual-spacecraft estimate in order to reduce the amount of information and panels in Figures 4, 8, and 12. Moreover, at the moment the dual-spacecraft LS estimate is not publicly available and the application of the dual-spacecraft LS technique to Swarm is not published. The associated changes are indicated by the blue cuts through text in all sections.

The revised manuscript includes the mapping of the optical frames to the geographic frame. This change leads to adjustments in the interpretation of the optical information. Thus, the paragraphs describing the optical data were updated accordingly for each Swarm event. For the event on 17 February 2015 the mapped optical data show that actually the southward auroral structure looks curled. For the event on 15 January 2015 we now clearly see a large scale auroral structure inclined by about 20° towards south, and small embedded auroral structures of different inclinations, rather east-west aligned. Also, for the last Swarm event the optical data are now more consistent with the results of the MSMVA analysis.

Section (2.6) was integrated into the discussion section, since we do not actually show comparisons with spectral techniques.

Finally, we found an error in the computation of $\partial_w \lambda_\eta$ for the logarithmic scale sampling scheme. This was caused by using the same code as for the linear scheme, where we have a constant discretization in the scale array, $dw$=const, whereas for the logarithmic scheme this is not the case. The results of the linear and logarithmic schemes show now a better consistency.

Various other corrections of the text were needed to better explain certain features or correct small errors. We hope that the revised manuscript is clearer, follows better the target of this work and avoids confusion on side subjects.

**3   General Comments**

The system performs remarkably well when tested on simulated data and I believe that the development path is a promising one – in particular the time domain nature of the method promises the potential of improved accuracy vs. frequency domain approaches. However when operating in the time domain, it is vitally important to somehow separate signals at various scales e.g. by zero-phase band-pass filtering. At the very least this direction should be explored and the results reported on.

The same concern was raised by Referee #1. We reproduce here the explanation included in the answer to Referee #1. The model functions implicitly employed to represent the magnetic field measurements are piece-wise linear functions of a certain length $w$, interpreted as the scale of the underlying current structure. The corresponding FAC density profile is a step function of the same width $w$, and centered at the same reference time $tcen$. This approach is compatible with established FAC estimators based on finite differencing. Actual magnetic profiles in the auroral zone are quite similar to these underlying piece-wise linear model functions, at least closer than perfectly smooth functions such as the ones employed for producing the synthetic data in section (3) (which are preferred there because of analytic tractability). Hence, we assume that our FAC scalogram performs actually better on real data than on the synthetic examples. Nonzero correlations among different piece-wise linear model functions lead to the non-orthogonal behavior criticized by both referees. The overall implications, however, depend on the particular subset of model functions associated with the chosen sampling scheme: (a) If for a given scale $w$ all available center times $tcen$ are used, model functions with neighboring $tcen$ are strongly correlated, resulting in a highly redundant and very non-orthogonal representation. This scale sampling scheme we call "linear". (b) If for a given scale $w$ the chosen center times $tcen$ are separated by the scale $w$, the model functions are only weakly correlated, resulting in a representation that is much less redundant and closer to orthogonality. This scale sampling scheme we call "logarithmic". The underlying logic is the

same as for the Haar wavelet transform. By comparing the results of linear versus logarithmic scale sampling for synthetic data, one finds that localization of center time/location and scale is more accurate with the linear sampling scheme. In logarithmic sampling, the center location of a current structure is heavily constrained by the scale $w$ that thus effectively represents the uncertainty of the $tcen$ (note also the uncertainty relation in wavelet analysis). Here our emphasis is on constraining FAC scales and center locations using a visualization tool, not on a full reconstruction of the FAC profile, thus we prefer to use a highly redundant set of model functions instead of an orthogonal and thus non-redundant one. Since the synthetic data are smooth profiles, and the scales are the widths of Gaussian profiles, we cannot expect that the piece-wise linear model functions identify the parameters perfectly. We updated the discussion section of the manuscript with this paragraph

As things stand and as evident in Figures 1 and 2, even when running the system on simple test inputs, without appropriately suppressing large scales, small-scale parameters such as variance, dB etc. will be dominated by whatever is happening at those large scales.

As detailed also in answer to Referee #1, we updated section (3) of the manuscript with a proper parametrization of the orientation. Sections (3.1) and (3.2) now describe more complex FAC structures that illustrate both the advantages and limitations of the MSMVA analysis. The superposition of FACs of different scales and orientations is considered. One can see that in this case both the scales and orientations are recovered with rather good accuracy, as described in the text (see also Report #1). The results of the analysis depend highly on the parameters of the superposed structures. Our numerical experiments indicate, not surprisingly, that the more intense FACs are characterized better. A more systematic study is needed to analyze the dependence of the results on the ratio of the relative intensities, scales, and orientations of the superposed structures. However, this is beyond the scope of the present paper.

As commented also in Report #1, we agree with the referee that filtering or isolation of a certain scale would improve the results at the respective scale. On the other hand, this would imply an apriori emphasis on that particular scale, which may introduce unwanted additional features. In particular, preprocessing/filtering (zero phase) is essentially a projection which distorts the interpretation and moreover the results depend on the type of filter and on the model functions.

A zero-level filtering is implicitly included in the MVA analysis. At each scale, the MVA analysis is performed and the FAC density is computed on a detrended perturbation obtained by extracting an average signal. Thus, a background perturbation, influence from large scales, is partially removed. A paragraph was added in section (2.3) to indicate this detrending procedure. As mentioned also in Report #1, an iterative scheme, that progressively removes the more intense FACs, may help overcoming the present limitations and the future improvement of the technique. A paragraph along this line was added at the end of section (3.2).

This is evidenced by the large error in calculated FAC in Figure 1 as the authors themselves admit to on page 11 lines 11-12 of the manuscript. When real data with multiple scales is analyzed (e.g. Figure 5) the FAC contributions at various scales appear to bleed into each other to such a degree that disentangling the contributions from various scales becomes very difficult. The bleeding is actually intrinsic for the current density, as illustrated e.g. by a uniform current sheet: current density remains constant for smaller scales than the sheet width and decreases to zero for larger scales. The current density panel provides just qualitative scale information, complementary to the more quantitative insight provided by the derivative of the maximum eigenvalue. At the end of section (3.1) we added comparison of the local multiscale FAC density with the input FAC density. We explained also the possibility to get the scale information from $j_{\parallel}$, indicated by a slight change of the slope of $j_{\parallel}$ for the linear sampling scheme or a decrease in amplitude for the logarithmic scheme, at the scale of the FAC element. The comparison of the output with the input FAC density shows now a difference of just 10% for the linear scheme (after correcting the technique in several respects, as detailed above under 'Major adjustments'). As discussed in section (3.2), for more complex superposed structures now we get as well consistent results. Section (3.2) was updated accordingly (see also Response to Report #1).

The authors attempt to correct for this as far as FAC density calculations are concerned by using weighting factors. In my opinion this can be potentially dangerous as then there is a risk of pre-supposing assumptions.

The single- and dual-spacecraft techniques do not deconvolve the information about the FAC density. We introduced the weighting as a simple way to qualitatively check the consistency of the MSMVA results with the dual- and single-spacecraft estimates. As explained above (section 2 of this Report) we removed completely the weightings from the manuscript.

Ultimately, we still do not know which scales are relatively the most important – this is what the methodology is designed to find out. By forcing weightings on scales there is a risk of the method presupposing its own conclusions. It is true that with the

weighting factors the system does a good job of reconstructing the observed time series – however the aim is not to reconstruct the time series but to decompose them in a way that reflects the truth.

With the revisions and corrections detailed above, including elimination of weightings, the technique appears to perform pretty well, both on synthetic and observed data. FAC elements are correctly identified and their properties derived by MSMVA are in good agreement with expectations. For the synthetic data, the parameters fit well with the input data. For the Swarm observations, the planarity, scale, and orientation of the various FAC elements are found to agree with the conjugate optical data. Current density of the FAC elements (assumed to dominate at specific locations) is in general consistent with the Swarm FAC products (provided that scale limitations of these products are properly considered).

I would thus recommend a revision of the manuscript where some way of separating the scales, perhaps by selective zero-phase band-pass filtering, is carried out with the results reported on. I believe that simple additions such as these may greatly improve the system's capacity and look forward to seeing the results of the developments.

As detailed in the text (last para of Section 3.2) and in the Response to Report #1 (second last specific comment), including some filtering scheme may also result in unwanted effects. For the time being, we would rather apply the present (corrected and revised) procedure to more events, and leave the (iterative?) filtering for a later stage.

**4   Specific comments:**

Page 2 lines 3-4 "median of the scale distribution around 230 m in the range of fine and small scale auroral arcs (10 m – 1 km)." – please specify where the 10 m to 1 km numbers are from, since they do not appear to be present in the reference (the minimum scale in the reference is 70 m).

The referee is right, 10 m is now corrected to 70 m. A relatively recent review paper on the fine scale aurora with scales below 1 km (Sandahl et al., 2008) is now mentioned as well in the introduction.

Page 2 line 7 "large sampling frequency difference, maximum at about 25 Hz for TV and 0.3 Hz for ASI" – for clarification it would be sensible to add something to the effect of "as any arcs which did not exhibit quasi-stationarity at the exposure timescales would likely have had their optical signals smeared and integrated to appear as larger- scale structures".

We added the sentence: "Note that arcs which are not quasi-stationary at the exposure timescales are likely to be smeared and integrated to larger scale structures in the optical data."

Page 2 lines 12-13 "Overall, the results of all these studies indicate a rather continuous scale spectrum" it may be better to write "Overall, the results of all these studies together indicate a rather continuous scale spectrum" since some of the individual studies alone certainly do not seem to indicate that!

Adjusted.

Page 2 lines 23-24 "(Karlsson and Marklund, 1996) found a median scale of about 4.6 km for the diverging electric fields observed by Freja" – please specify where the referenced paper mentions that number, since I am unfortunately unable to find it

Thank you for checking. The value is taken from Johansson et al. (2007) (Figure 9) where the peak of the distribution is around 4 km. Text adjusted by the sentence:"Johansson et al. (2007) (Figure 9) compare also the scale distribution with former results. We notice the distribution of the diverging electric fields observed by Freja with the peak around 4 km."

Page 4 lines 1-2 "Gillies et al. (2015) pointed out that the single- spacecraft FAC density provides better identification of the boundaries of auroral patches." – Is this "better" as compared to the dual-spacecraft FAC density? Please specify. As I understand this paragraph advocates for the need to analyse multiple scales including small scales which may not be well-captured by the dual-spacecraft method or indeed by the 1 Hz single-spacecraft FAC product. In this context a valuable reference to add would be Miles et al. (2018) who studied an intense discrete auroral arc crossing observed by Swarm and e-POP, the paper vividly demonstrates the limitations of both the dual spacecraft FAC density product and the small-tilt assumption when analyzing small- scale auroral structures.

The text was adjusted accordingly, including the reference to (Miles et al., 2018).

Page 5 lines 26-27 the authors state the Equation (3) was used for many single- spacecraft missions but then proceed to give only 1 reference (to Freja data). Please either amend the wording (e.g. "Equation (3) was used successfully to obtain partial estimates of the FAC density for some single-spacecraft missions") or else add more references to where the single-spacecraft

product was used (e.g. Swarm).

Sentence adjusted by references to FAST and Swarm.

Page 5 line 21 "Thus, the cross-track separation defines the lower limit of the FAC scales" – authors should specify that the lower scale limit is in the cross-track direction. The scale limit in the along-track direction is determined by the inter-spacecraft along- track separation and the degree of quasi-stationarity of those FACs.

We rephrase the text to: "The cross-track separation defines the lower limit of the FAC scales in cross-track direction, whereas the limit in the along-track direction is determined by the along-track separation, provided that the FAC structure is quasi-stationary."

Page 15 lines 5-6 (also Figure 4) – the origin of the time shift is explained fine but it is visually difficult to compare the two results with the time shift in place. I would recommend factoring it out so that the two estimates are overplotted directly and the differences can be more clearly seen.

For the first two events we recomputed the LS FAC density and assigned the time stamp correctly from SwA, similar to the way L2 product (FD estimate) is defined. For the last event (27 September 2014), the default L2 time stamp related to SwA was modified by shifting the FAC density with the computed time lag. The use of a common time stamp for both LS and FD dual-spacecraft FAC density estimates showed a high similarity of these estimates. Thus, we decided to remove the LS dual-spacecraft FAC density from the manuscript for the reasons mentioned above in section (2).

**5 Technical corrections**

Page 8 line 4 please close the 2nd bracket

Corrected

Page 15 line 27 "slightly" Please replace "associated to" to "associated with" (e.g. in lines 22-23 in page 1)

Corrected

**6 Figures:**

Please label the plots in the figures (a1, b1 etc. . . are described in the figure captions but are not actually labeled by the sides of the plots). In Figures 1, 2, 5, 9 and 13 planarity (on the y-axis) appears to be mis-labeled as 'scale' in the plots.

We have indeed mis-labeled panels (a) and (e). According to the suggestion of Referee #1 we relabeled the panels using unique letters. All the spectrogram panels show the respective quantities as a function of time/position (x axis) and scale (y axis). The panel with wrong y label was (e) and is now changed to 'Scale [s]'. Another wrong y label, noticed by Referee #1 in panel (a) of Figures 5, 9, 13, is now corrected.

**References**

Bunescu, C., Marghitu, O., Constantinescu, D., Narita, Y., Vogt, J., and Blăgău, A.: Multiscale field-aligned current analyzer, Journal of Geophysical Research: Space Physics, 120, 9563–9577, https://doi.org/10.1002/2015JA021670, http://dx.doi.org/10.1002/2015JA021670, 2015.

5 Johansson, T., Marklund, G., Karlsson, T., Liléo, S., Lindqvist, P.-A., Nilsson, H., and Buchert, S.: Scale sizes of intense auroral electric fields observed by Cluster, Annales Geophysicae, 25, 2413–2425, https://doi.org/10.5194/angeo-25-2413-2007, http://www.ann-geophys.net/25/2413/2007/, 2007.

Karlsson, T. and Marklund, G. T.: A statistical study of intense low-altitude electric fields observed by Freja, Geophysical Research Letters, 23, 1005–1008, https://doi.org/10.1029/96GL00773, http://dx.doi.org/10.1029/96GL00773, 1996.

10 Miles, D. M., Mann, I. R., Pakhotin, I. P., Burchill, J. K., Howarth, A. D., Knudsen, D. J., Lysak, R. L., Wallis, D. D., Cogger, L. L., and Yau, A. W.: Alfvénic Dynamics and Fine Structuring of Discrete Auroral Arcs: Swarm and e-POP Observations, Geophysical Research Letters, 45, 545–555, https://doi.org/10.1002/2017GL076051, https://agupubs.onlinelibrary.wiley.com/doi/abs/10.1002/2017GL076051, 2018.

Sandahl, I., Sergienko, T., and Brändström, U.: Fine structure of optical aurora, Journal of Atmospheric and Solar-Terrestrial Physics, 70, 2275 – 2292, https://doi.org/https://doi.org/10.1016/j.jastp.2008.08.016, http://www.sciencedirect.com/science/article/pii/S1364682608002162, transport processes in the coupled solar wind-geospace system seen from a high-latitude vantage point, 2008.
15

---

## Author Comment (AC3) · 30 Nov 2018

The comment was uploaded in the form of a supplement: https://www.ann-geophys-discuss.net/angeo-2018-70/angeo-2018-70-AC3-supplement.pdf

---

## Author Comment (AC4) · 30 Nov 2018

[revised manuscript text omitted]
 Δξ(k) we have the current density j||(k) = μ0-1ΔBη(k)/Δξ(k) and the integrated current J||(k) = μ0-1ΔBη(k). The FAC density 20 j||(k) reflects the slope of Bη whereas J||(k) the jump of Bη over the respective scale, wk. Both j||(k) and J||(k) offer complementary useful information. In the following we concentrate on j||(k), similar to linear sampling scheme.

As it is constructed, the multiscale FAC density provides estimates of the local FAC density present at various scales. Both scale sampling schemes relay on a non-orthogonal basis functions because the aim is to precisely infer the scale and location of the FAC as well as the respective current density. As a consequence, one cannot simply integrate over scales to obtain a global FAC density estimate that can be compared with the single- and dual-spacecraft FAC estimates - which provide convoluted information about the FAC scales larger than the discretization interval (single-spacecraft) or the virtual quad scale (dual-spacecraft). As compared to the orthogonal decompositions, e.g. orthogonal wavelet decomposition, where the signal is recovered easily by integration over scales, in our case such an integration would require a proper weighting scheme of the

**30 2.5 FAC time series reconstruction**

multiscale information. This development is considered for a future study.

25

In both scanning procedures, linear and logarithmic, the scale array does not provide an orthogonal basis. As shown in sections 3 and 4, the FAC density is not concentrated around the FAC scale, there is a spread mainly to smaller scales. In order to compensate the dispersion of the FAC density over scales, we need a weighting function when the desire is to (partially) reconstruct a known current density by integration over the scale domain. Such a reference FAC density is available in the case

of synthetic FAC structures (section 3). Based on the performance on synthetic FACs we extend to measured data (section 4), where we can also use reference FACs as provided by the two-spacecraft methods for ideal simple FACs. In the following, we use different weighting functions to compute the scale integrated current  $j_{\parallel}(\xi)$ .

$$j_{\parallel}(\xi) = \sum_{k} j_{\parallel}^{(k)} h^{(k)} / \sum_{k} h^{(k)}$$
(7)

where j(k)|| and h(k) denote the FAC density and the weighting coefficient at the k scale. The average FAC density is obtained
by considering the simplest weighting, h(k)=1. Other weighting functions can emphasis the large-scales, h(k) = w(k) (w the interval length), or the small-scales, h(k) = 1/w(k). We denote average, large-scale, and small-scale weighting by AW, LW, and SW, respectively. The different weighting results are compared with reference FAC density profiles in section 3 and section 4.

**3** Synthetic FAC structures**

In this section we apply the multiscale FAC density technique to synthetic structures consisting of superposed FAC activity. We define complex FAC structures by superposing FACs of different scales (thickness), amplitudes (FAC intensity), and different directions of the current flow (upward and downward). Additionally, one can we consider the orientation of the FAC structures in the plane perpendicular to *B*. The total FAC density in the  $(\xi, \eta)$  frame is given by:

$$j_{\parallel}(\xi) = \sum_{k} s^{(k)} j_{\parallel}^{(k)}(\xi, \underline{\xi_{0}^{(k)}}, \sigma_{\perp}^{(k)}, \underline{\theta^{(k)}})$$

$$\tag{8}$$

where  $j_{\parallel}^{(k)}$  denotes the elementary current associated with a single FAC element;  $s^{(k)}$  is the sign of the FAC element, -/+ for the upward/downward FACs. For the case of uniform FAC density structures  $j_{\parallel}^{(k)} = \text{const}$ ;  $j_{\parallel}^{(k)}$  is parametrized by the perpendicular 15 scale  $\sigma_{\parallel}^{(k)}$ , the location of the FAC center,  $\xi_{0}^{(k)}$ , and the orientation  $\theta^{(k)}$ . below by thickness, position, intensity, and orientation.

In the following, we define  $j_{\parallel}^{(k)}$  elements according to a nonuniform FAC density depending on  $\xi$  by a Gaussian function in the  $(\xi, \eta)$  frame.

$$j_{\parallel}^{(k)}(\xi, J_{0}^{(k)}, \underline{\xi_{0}}, \sigma_{\perp}^{(k)}, \underline{\theta}) = \frac{J_{0}^{(k)}}{\sigma_{\perp}^{(k)}\sqrt{2\pi}} e^{-\left(\frac{\xi\cos(\theta) - \xi_{0}}{\sigma_{\perp}^{(k)}\sqrt{2\pi}}\right)^{2} / \left(2\left(\sigma_{\perp}^{(k)}\right)^{2}\right)}$$
(9)

[revised manuscript text omitted]

---

## Author Comment (AC6) · 30 Nov 2018

The comment was uploaded in the form of a supplement:
https://www.ann-geophys-discuss.net/angeo-2018-70/angeo-2018-70-AC6-supplement.pdf
* * *

---

## Referee Report (RR1)

Reply to authors

The paper details a technique called MSMVA which is used to disentangle parameters at various scales when analyzing the inherently multiscale FAC phenomena observed in the auroral zone by satellites. The technique is tested on both simulated and real data, with results as well as sources of error discussed in the manuscript. After revision by the authors, the work is significantly improved and this referee would recommend the paper for publication after the minor technical corrections detailed below.

Please note that I am working off the manuscript named 'angeo-2018-70-manuscript-version2' with the corrections deliberately shown as crossed out in multiple colors as detailed in the authors' response. The line numbers are accordingly the ones for that particular manuscript, not for the final version with the corrections hidden.

Below, the single quotations marks '' enclose my suggested phrasing, while double quotation marks "" are direct quotes from your paper.

Line 9 'For both synthetic and Swarm data' (comma unnecessary)

Line 10 'namely the linear and the logarithmic scanning' (commas unnecessary)

Line 14 'complex FAC signatures that complements' (comma unnecessary)

Line 1 'The multiscale character is observed also in the measurements'

Line 2 'associated in turn with'

Line 8 'The TV and ASI observations also correspond to'

Line 15 'found to peak around 400-500 m with an average of 615 m' (comma unnecessary)

Line 16 'below 1 km and 1 s' (not bellow)

Line 27-28 'associated with' (not associated to)

Line 2 'also have' (not have also); 'also compare' (not compare also)

Line 26 "the rate of variation is significantly higher" – what is meant in this sentence? Do you mean the uncertainty is higher, or do you mean the rate of the variation of the structure lifetime is much higher? Which figure are you referring to in Gjerloev et al. (2011) to make this sentence? Please clarify

Line 26-27 "the minimum relevant scale size" – the minimum relevant scale size for what? Please clarify. Do you mean the separation between the quasi-static and non-stationary perturbations?

Line 27 "about 20 km" where in the paper is this obtained from? Please clarify. Do you mean 200 km, or 20 sec? 200 km is in the mesoscale range; 20 km is only 2-3 sec in LEO.

Line 2 'arc orientation from the east-west direction' (not form)

Line 32 'both quiet and more dynamic, smaller scale' (the second 'and' sound strange upon reading the text)

Lines 3-4 'FAC scales in the cross-track direction' (missing 'the')

Line 19 'obtained' (not obtain)

Line 20 'over an array' (not over and array)

Line 21 'on small-scale FAC signatures' (not at small-scale FAC signatures)

Line 29 'FAC density in the FAC's own reference system' (not FAC density into the FAC own reference system)

Line 14 'separated into invariant information, which depends' (not separated into invariant, depending)

Line 14 'in the local frame, and' (please add comma)

Line 15 'which depends' (not depending)

Line 20 '(Section 3)', '(Section 4')' not section4

Line 24 'to inclined FAC observations' (not inclined FACs observations)

Line 33 where does the ionospheric mapping factor of 1.1 come from? Is there a reference for this number by any chance?

Line 1 'provided the high resolution needed' (not a high resolution)

Line 12 'the largest scale samples the entire oval' (not sample)

Line 12 'second-largest scale' (not second scale)

Line 20 'reflects the jump'

Line 23 'sampling schemes rely on non-orthogonal basis functions' (not relay on a non-orthogonal basis functions)

Line 19 $\xi_0$ needs to be defined if it is used in equations

22 'associated with' (not associated to)

Lines 12-13 "at the center of the two structures" is it really at the center of the two structures, or is it just the maximum of their summed contributions? Please clarify

Line 19 'the scale is more consistent' (not better consistent)

Line 33 'This behavior' (not this behaviors)

Line 9 'small scale FACs' (not small scale FCAs)

Line 18 'also show' (not show also)

Line 34 'FAC signatures' (not FACs signatures)

Line 1 'engaging in such a development' (not engaging on such a development)

Line 2 'several real events' (not several data events)

Line 27 '~7 km'

Line 2 'the THEMIS mission'

Line 26 'does not capture' (singular; not do not capture)

Line 3 'associated with' (not associated to)

Line 19 'associated with' (not associated to)

Line 28 "about 300 km" – the authors use km but in Figure 5 for both linear and logarithmic methods, the y-axis is in seconds. It is not easy to see where 300 km would be if they are not marked on the axis in the proper units. Please clarify whether the scale is in terms of km or in terms of seconds. This also applies to other figures. Figures 1 and 2 (with simulated data) have the y-axis in km on a linear scale (for the linear method) but the figures thereafter (with real data) have the y-axis in seconds and on a logarithmic scale. What is the reason for changing the setup between the two sets of figures?

Line 4 'agree' (not agrees; plural)

Lines 20-21 'better quantify' (not quantify better)

Line 27 'The event was observed'

Line 13 'associated with'

Line 32 'associated with'

Line 17 'associated with'

Lines 5-6 'separated the observations into two categories' (not in two categories)

Line 11 'The event was observed' (past tense)

Line 30 '(locally planar)' (please enclose "locally planar" in brackets)

Line 2 'is by filtering' (not if by filtering)

Line 9 'well-suited for' (not well suitable for)

Line 16 "sections 4.3 and 4.3" – repetition. Do you mean sections 4.2 and 4.3?

Line 7 'and can provide' (not an can provide)

Figures

Figure 1

It would be good to add a legend to panels a, b, g, and h or remind the reader in the caption which component is which

Would it be possible to make larger labels, especially for the colorbars? Also, would it be possible to make the figures bigger? There is a lot of information in them and it would benefit the reader to have larger figures.

Figure 4

The hodograms (e) seem to have multiple shades of green, orange etc. which do not appear to have a correspondence with the legend colors. Please clarify. This also applies to other hodograms in the manuscript.

Figure 5

As discussed earlier, please clarify whether the scale is in seconds or km, and if it is in seconds, please comment on why it was in km for the synthetic data earlier. This applies to all subsequent spectral plots in the manuscript which use scale as the y-axis.

Figure 6

Caption: 'blue line in (a) indicates a reference level ' (not "blue line in (a) indicate a reference level")

---

## Author Response (AR2)

**1  Summary**

This manuscript applies the minimum variance analysis (MVA) to in-situ magnetic field data during auroral crossings. MVA is applied to a set of sliding windows of across the data with differing window lengths to attempt to examine the orientation, planarity and field-aligned currents at different scales. As noted previously, being able to determine current sheet orientation to correct for this in field-aligned current calculation is useful, however the non-orthogonality of the data remains a major unaddressed issue.

We understand the referee's concern on non-orthogonality and, both in the revised text and in the response to the referee report, we mentioned this limitation and openly addressed the matter. Therefore, we feel that the referee's statement "...the non-orthogonality of the data remains a major unaddressed issue" is not correct. Non-orthogonality was addressed by adjustments to the text and in the response to the referee report, see, e.g., the text paragraph in the Discussion section (page 25, lines from 14 to 34), "The model functions implicitly employed to represent the magnetic field measurements (...) we cannot expect that the piece-wise linear model functions identify the parameters perfectly.".

When we analyze the Swarm events we observe that in the rather ideal case of event from section (4.2) we have quite small deviations with respect to the other heavily used methods, single and dual-spacecraft which are providing the contributions from scales larger than the discretization interval. Considering the simplicity of this event and the small deviations we consider that the method is performing well. Following the previous Reports, we tried to demonstrate that non-orthogonality appears to contribute positively to the method (in particular to the linear sampling scheme, where accuracy is better). Thus, we rephrased the text and aimed for the simplification of the paper and not to add more analysis.

**2  Technical Comments**

The authors have removed the weighting factors that were present in the previous version of this manuscript. However, since they are not using orthogonal basis functions for this analysis means that the interpretation of their analysis remains challenging.

The results based on both synthetic and observed data suggest that our approach is meaningful. While reconstructing the total current density remains, indeed, a challenge, we detailed above, in the previous Response, and in the revised text why we see non-orthogonality as an asset, rather than as an issue.

The manuscript describes the MSMVA procedure as having an embedded scale filtering by removing an average over the window to which the analysis is applied (Page 7, lines 18-23). This procedure is somewhat unclear, however based on later descriptions that indicate that $\Delta B$ is determined from a linear fit across the window (Page 8, lines 4-10), it appears that the average magnetic field within the window is then removed from the same window. This should have no effect on the signal. The effect of large-scale FACs on smaller-scale FACs can only be removed if the sliding average window is larger than the scale over which the currents are calculated.

The referee is right, removing the average magnetic field in the window does not affect the analysis. The computation of the FAC density for a certain window is independent on the average detrending since the slope of the linear fit is the same. We get the scale information from $\partial_\xi \lambda_\eta$. Following the referee's comment we deleted the second paragraph in section (2.3) where we described the average detrending. Also two sentences from the third last paragraph of section (3.2) were deleted and one phrase form section (5) as indicated in the corrected manuscript.

The manuscript describes that this technique provides estimates of the 'local FAC density present at various scales' but that the current cannot be integrated across scales. I would suggest instead that this technique provides an average FAC across scales and some indication of the dominant scale given by peaks in $\partial_w \lambda_\eta$??. In fact, this appears to be how this technique is interpreted in the manuscript. The fact that larger scales tend to be focused on may be as a result of not apply the scale correction to all the spectrograms (page 8, line 20).

We adjusted lines 16-17 at page 9 according to the Referee suggestion: "As it is constructed, the multiscale FAC density provides estimates of the average FAC across scales, as well as an indication of the dominant scales, given by peaks in $\partial_\xi \lambda_\eta$."

We also added two sentences in section (2.3)-at the end of the first and third paragraphs- to better explain that we apply the amplitude corrections for the derivative and FAC density. The representations of the MSMVA parameters as a function of the corrected scale is done for all FACs (Swarm and simulated) only in the vertical cuts (profile) representations. The amplitude correction applies to $\partial_w \lambda_\eta$. For the FAC density the amplitude correction is contained in equation (6) for Swarm events, whereas for the simulated data is also applied by correction with $\cos(\theta)$. We now make a better distinction between the amplitude corrected derivative, labeled by $\partial_\xi \lambda_\eta$, and the amplitude uncorrected derivative, labeled by $\partial_w \lambda_\eta$. Thus, after introducing the previous results (based on $\partial_w \lambda_\eta$) in the first paragraph of section (2.3), we then use only the notation $\partial_\xi \lambda_\eta$ for the rest of the paper. This change was not made visible (marked by red) in the annotated manuscript. We deleted the sentence "Since in this study we are not interested in the value of $\partial_\xi \lambda_\eta$, the amplitude correction is only applied for the synthetic FACs (Section 3) and not to the Swarm events (Section 4)." because we now show also the amplitude correction of the derivative for the Swarm events.

To conclude, the amplitude corrections (i.e. color scale) are done for all spectrograms of the derivatives and of the FAC density. The scale correction is included only in the vertical cuts showing the quantitative information.

While I understand that the application of MVA across a range of scales can potentially provide more information on the orientation of various magnetic field perturbations observed by spacecraft as they pass over the aurora, I struggle to understand the benefit of the analysis as presented. How is one expected to interpret the results if the FACs at different scales are not isolated from one another? For example, comparing the time-series of FACs at two different scales would not appear to be meaningful. How does this analysis differ from computing MVA between local maxima and minima in the magnetic field data?

The main purpose of this paper was the introduction of the FAC scalogram as a visual aid to identify and study multi-scale structures (that in real life may overlap and hence nicely isolated features cannot be always expected). In order to quantitatively use the information one can simply analyze the corrected amplitude and scale cross-sections through the spectrograms, as detailed in the text. The FAC density at the scales associated with signatures in $\partial_\xi \lambda_\eta$ was found to provide consistent information. Last but not least, unlike computing MVA between local maxima and minima in the magnetic field data, which is binded to specific scales (the distances between maxima and minima), our technique is not constrained in this respect.

**3 Specific Comments**

Within Section 4, there are a number of comparisons between single- and dual-spacecraft currents and those calculated from the MSMVA technique. It may help the reader if this information was tabulated to aid comparisons.

For the first event (section (4.2)) and the second event (section (4.3)) we added the respective information into tables. For the last event (section (4.4)) we only discuss the range of variation of the parameters for the selected FACs. Thus, we did not include a similar table for section (4.4).

**Reviewer's comments on ANGEO-2018-70 (revised) by Bunescu et al.**

*Anonymous Referee #2*

Reply to authors

The paper details a technique called MSMVA which is used to disentangle parameters at various scales when analyzing the inherently multiscale FAC phenomena observed in the auroral zone by satellites. The technique is tested on both simulated and real data, with results as well as sources of error discussed in the manuscript. After revision by the authors, the work is significantly improved and this referee would recommend the paper for publication after the minor technical corrections detailed below.

Please note that I am working off the manuscript named 'angeo-2018-70-manuscript-version2' with the corrections deliberately shown as crossed out in multiple colors as detailed in the authors' response. The line numbers are accordingly the ones for that particular manuscript, not for the final version with the corrections hidden.

Below, the single quotations marks '' enclose my suggested phrasing, while double quotation marks "" are direct quotes from your paper.

We thank the referee for suggestions. All suggested adjustments were included in the manuscript. Further clarifications / details in response to the questions / comments of the referee are provided below (blue text).

Line 9 'For both synthetic and Swarm data' (comma unnecessary)
Line 10 'namely the linear and the logarithmic scanning' (commas unnecessary)
Line 14 'complex FAC signatures that complements' (comma unnecessary)

Line 1 'The multiscale character is observed also in the measurements'
Line 2 'associated in turn with'
Line 8 'The TV and ASI observations also correspond to'
Line 15 'found to peak around 400-500 m with an average of 615 m' (comma unnecessary)
Line 16 'below 1 km and 1 s' (not bellow)
Line 27-28 'associated with' (not associated to)

Line 2 'also have' (not have also); 'also compare' (not compare also)
Line 26 "the rate of variation is significantly higher" – what is meant in this sentence? Do you mean the uncertainty is higher, or do you mean the rate of the variation of the structure lifetime is much higher? Which figure are you referring to in Gjerloev et al. (2011) to make this sentence? Please clarify

We extracted this information from Figure (7)-right panel in Gjerloev et al. (2011) showing the location of the boundary between the correlated and the uncorrelated FACs. We meant the rate of variation, but we agree that the sentence might be confusing for the reader. We changed the text to: "The same is true for large-scales, however in this case the lifetime increases faster with the structure scale."

Line 26-27 "the minimum relevant scale size" – the minimum relevant scale size for what? Please clarify. Do you mean the separation between the quasi-static and non-stationary perturbations?

Line 27 "about 20 km" where in the paper is this obtained from? Please clarify. Do you mean 200 km, or 20 sec? 200 km is in the mesoscale range; 20 km is only 2-3 sec in LEO.

We mean 20 km. This is the minimum scale size observed by ST5 and is related to operational conditions, spacecraft velocity of 7 km/s, spin period of ∼3 s. This information was extracted from the last paragraph of section (6) in Gjerloev et al. (2011)

. We slightly adjusted the text to "The ST-5 data constrained the analysis of Gjerloev et al. (2011) to scale sizes above 20 km, which is situated in the mesoscale range (Knudsen et al., 2001)."

Line 2 'arc orientation from the east-west direction' (not form)

Line 32 'both quiet and more dynamic, smaller scale' (the second 'and' sound strange upon reading the text)

Lines 3-4 'FAC scales in the cross-track direction' (missing 'the')

Line 19 'obtained' (not obtain)

Line 20 'over an array' (not over and array)

Line 21 'on small-scale FAC signatures' (not at small-scale FAC signatures)

Line 29 'FAC density in the FAC's own reference system' (not FAC density into the FAC own reference system)

Line 14 'separated into invariant information, which depends' (not separated into invariant, depending)

Line 14 'in the local frame, and' (please add comma)

Line 15 'which depends' (not depending)

Line 20 '(Section 3)', '(Section 4')' not section4

Line 24 'to inclined FAC observations' (not inclined FACs observations)

Line 33 where does the ionospheric mapping factor of 1.1 come from? Is there a reference for this number by any chance?

The ionospheric mapping factor was computed from the data by taking the ratio of the measured magnetic field by Swarm and the ionospheric mapped model magnetic field at ionospheric level (100 km) using field line tracing of the Swarm trajectory. The model magnetic field used is Tsyganenko T04 (internal field given by IGRF) using the solar wind parameters for the respective intervals of the measured data. The coefficient was computed for each of the three events and provided values of 1.087 for the first event (2015-02-17), 1.086 for the second event (2015-01-15), and 1.089 for the third event (2014-09-27). In the text we used 1.1 and 1.09 for some of the ionospheric mapped scale estimates.

Line 1 'provided the high resolution needed' (not a high resolution) Line 12 'the largest scale samples the entire oval' (not sample)

Line 12 'second-largest scale' (not second scale)

Line 20 'reflects the jump'

Line 23 'sampling schemes rely on non-orthogonal basis functions' (not relay on a non-orthogonal basis functions)

Line 19 $\xi_0$ needs to be defined if it is used in equations

The parameter $\xi_0$ is no longer used in Equation (9). Equation (9) refers to the FAC frame ($\xi$,$\eta$). The red underline text in Equation (9) was indicating deleted text. In the revised manuscript we define the center of the FAC elements in the (x,y) frame in Equation (10) by the variable $x_0$.

22 'associated with' (not associated to)

Lines 12-13 "at the center of the two structures" is it really at the center of the two structures, or is it just the maximum of their summed contributions? Please clarify

In Figure (1) panels (a)/(g) show the same information. The individual contribution from the first (downward) FAC (FD structure) is shown by magenta, whereas the the second (upward) FAC (FU structure) is shown by blue line. The total FAC density is shown by the black line. Thus, the plot show the individual FAC densities in the center of each FAC is around 5 $\mu$A/m$^2$ basically equal to the summed contributions (black).

Line 19 'the scale is more consistent' (not better consistent)
Line 33 'This behavior' (not this behaviors)

Line 9 'small scale FACs' (not small scale FCAs)
Line 18 'also show' (not show also)
Line 34 'FAC signatures' (not FACs signatures)

Line 1 'engaging in such a development' (not engaging on such a development) Line 2 'several real events' (not several data events)
Line 27 ' 7 km'

Line 2 'the THEMIS mission'

Line 26 'does not capture' (singular; not do not capture)

Line 3 'associated with' (not associated to)
Line 19 'associated with' (not associated to)
Line 28 "about 300 km" – the authors use km but in Figure 5 for both linear and logarithmic methods, the y-axis is in seconds. It is not easy to see where 300 km would be if they are not marked on the axis in the proper units. Please clarify whether the scale is in terms of km or in terms of seconds. This also applies to other figures. Figures 1 and 2 (with simulated data) have the y-axis in km on a linear scale (for the linear method) but the figures thereafter (with real data) have the y-axis in seconds and on a logarithmic scale. What is the reason for changing the setup between the two sets of figures?

We corrected the text. It was a qualitative discussion/estimate and the units should have been seconds (300 s). We changed the value to 200 s and gave an estimate in km (1381 km) considering the spacecraft velocity of 7.6 km/s and the mapping factor of 1.1. The simulated data are defined in terms of the spatial along-track coordinate, $x$ variable. In this case the crossing is along $x$ at constant $y$ coordinate. The real data consist of time series signals analyzed and represented as a function of the temporal scale domain in sec. The spectrograms are represented as a function of sampled scales (along track scale), whereas the cross-sections (vertical-cuts) in corrected coordinates (perpendicular scale). In order to make the representations of Swarm data as a function of ionospheric mapped scales in km one should simply use a certain constant velocity and a mapping factor (see Bunescu et al. (2015)). For simplicity in this article we decided to keep the representations as a function of their definition or measurement setup, km for the stimulated FACs and sec for the measured data.
Page 21 Line 4 'agree' (not agrees; plural)
Lines 20-21 'better quantify' (not quantify better)
Line 27 'The event was observed'

Page 22 Line 13 'associated with'
Line 32 'associated with'

Page 23 Line 17 'associated with'

Page 24 Lines 5-6 'separated the observations into two categories' (not in two categories) Line 11 'The event was observed' (past tense)

Page 25 Line 30 '(locally planar)' (please enclose "locally planar" in brackets)

Page 28 Line 2 'is by filtering' (not if by filtering)
Line 9 'well-suited for' (not well suitable for)
Line 16 "sections 4.3 and 4.3" – repetition. Do you mean sections 4.2 and 4.3? Page 29
We corrected the text, it is only section (4.3). The text was related to the first submitted manuscript in which the un-mapped optical data were showing periodic structures. In the last manuscript the mapped optical data show quite different, a large scale inclined auroral with small scale structures. Thus, we added also the text "and/or inclined FACs with embedded smaller scale FACs" which corresponds more exactly to the event in section (4.3).
Line 7 'and can provide' (not an can provide)

Figures Figure 1 It would be good to add a legend to panels a, b, g, and h or remind the reader in the caption which component is which.
We included the description of the panels in the caption. Panels a) and g) are similar. Same for b) and h). I added the text: "for FD (magenta), FU (blue) and the summed contribution from both FACs (black)." For panel b) I added the text: "Magnetic field in the $(x, y)$ frame with $B_x$ (blue) and $B_y$ (green)."
Would it be possible to make larger labels, especially for the colorbars? Also, would it be possible to make the figures bigger? There is a lot of information in them and it would benefit the reader to have larger figures.

We updated all figures from the manuscript. To have larger labels on the colorbar axis we increased the font size of all labels. I will provide to Annales bigger plots and they can decide how it would be better to format the article. The current format is just our proposal for a more compact representation with three columns in a single figure for Figures (1) and (2).
Figure 4
The hodograms (e) seem to have multiple shades of green, orange etc. which do not appear to have a correspondence with the legend colors. Please clarify. This also applies to other hodograms in the manuscript.
It is true that there are shades of other colors in the hodograms. The hodogram is scaled and represented first in a rainbow color scale from blue to red to better see the evolution (start/stop of the interval). On this trace we superpose a second layer in which we plot the selected intervals with different colors (associated with the labels) for a better identification of the respective intervals. This was explained shortly in the text:"The hodogram is represented with the time interval running from blue to red". Now we added a short explanation also to the caption of Figure (4):"The hodogram is first represented in a rainbow color scale (blue to red) on which we superpose a layer of identified FAC intervals using discrete colors associated to the labels."
Figure 5 As discussed earlier, please clarify whether the scale is in seconds or km, and if it is in seconds, please comment on why it was in km for the synthetic data earlier. This applies to all subsequent spectral plots in the manuscript which use scale as the y-axis.

The scale representation in spatial coordinate (in km) for synthetic data is the natural representation since the current density is computed by the spatial derivative. For the measured data the natural choice is the temporal coordinate (in seconds) since all quantities are measured as time series. As explained above, to change to ionospheric mapped spatial scale in km we should assume a constant velocity component along the track and a constant mapping factor. Although not the case for Swarm, for a higher altitude spacecraft we might have a variation of the velocity and of the mapping factor over the entire interval and this might relate to a non-uniform spacing in the spatial coordinate.

[revised manuscript text omitted]

---

## Author Response (AR3)

**1   Summary**

5   This manuscript applies the minimum variance analysis (MVA) to in-situ magnetic field data during auroral crossings. MVA is applied to a set of sliding windows of across the data with differing window lengths to attempt to examine the orientation, planarity and field-aligned currents at different scales. The authors have made changes following earlier comments, however these changes reveal a deficiency in this analysis which is not fully resolved.

**2   Technical Comments**

10   The tabulation of the results in Section 4.3 shows that the current estimated from the MSMVA technique is 20-160% greater than that calculated from traditional FAC calculations. The manuscript indicates that this is due to the an alignment of the current system away from east-west. The table shows that the FACs are 20-25 degrees away from the east-west direction. However, such a deviation should mean that the true FAC is  13-20% greater than the single spacecraft approximation (the single spacecraft approximation is corrected by $1/cos^2(\theta)$). As such, the orientation does not account for a deviation of 160%.

15   This discrepancy is not discussed in sufficient detail. While the other case study shows that the technique can give comparable results for simple current sheet configurations, the point of this technique is that it is designed to examine more complex scenarios with greater accuracy.

It is true that the orientation alone cannot account for the differences between the FAC estimates. The L2 single-sc FAC product and the MSMVA analysis provide estimates of the FAC density at different resolutions, are based on a slightly different
20   computation procedure, and address the scale aspect in a different way.

The L2 single-sc FAC density is provided at 1 s resolution and thus enables a characterization of the current at scales larger than or equal to 1 s. This estimate typically takes into account only the magnetic perturbation in the east-west component ($B_y$).

The MSMVA FAC density estimate is computed using the 50 Hz magnetic field perturbation and is given at a resolution of 0.1 s ($\sim$760 m) in time ($x$-axis in the scalograms) and 0.04 s (highest resolution scanning) in the scale ($y$-axis in the
25   scalograms). This estimate is based on the magnetic field perturbation in the FAC's tangential direction ($B_\eta$).

The estimates of the current density included in the previous manuscript for the event on 15 January 2015 correspond to mesoscale FAC signatures (upward current regions labeled by U2, U3, U4). The MSMVA estimates, selected based on $\partial_\xi \lambda_\eta$, were simply compared with the instantaneous values of the L2 single- and dual-sc FAC estimates. In order to properly compare the different FAC estimates, they should be computed at approximately the same scale. Whereas the scalograms of
30   the FAC density provide the possibility to visualize/select the local average current at a certain scale, the L2 single-sc estimate is dependent on the local magnetic field perturbations present also at smaller scales. The MSMVA FAC estimates for U2-U4 structures (Table 2) correspond to the mesoscales delimited by the vertical lines in Figures 8 and 9. The selected values of the L2 single-sc FAC density (Table 2) reflect also the embedded smaller scale perturbations (internal structure of U2-U4 regions).

In order to clarify the differences we estimated the average L2 single-sc FAC density by using a boxcar running average
35   (12 s width) over the L2 single-sc estimate. The 12 s window corresponds to the thickness of the U2-U4 FACs. Thus, we updated Table 2 by including a comparison of MSMVA estimates with this average estimate and shortly commented in the text about it and its relative percentage deviation with respect to MSMVA. We also slightly adjusted some of the MSMVA FAC estimates (see Table 2 and the text in section 4.3). To increase the visibility of smaller scale FACs we also updated Figure 9: the scalograms of $\partial_w \lambda_\eta$ and $j_{\parallel}$ for the linear and log scanning are now represented using a logarithmic color scale, similar to
40   the last Swarm event (2014-09-27).

Figure 1 (this response) shows SwA magnetic field perturbation and the L2 single- and dual-sc FAC estimates. The vertical solid lines delimit the U2-U4 FACs. We notice that most of the mesoscale FACs (including U2 and U4) have an internal structure. For both U2 and U4 regions we have embedded perturbations visible through the slope change of $\Delta B$ inside the

respective intervals (panel-1). Panel-2 shows the L2 single-sc FAC (green), the L2 dual-sc product (black), and the average L2 single-sc FAC density (red) estimated by using the 12 s boxcar average window on the L2 single-sc FAC density. The internal structure provides distinct peaks in the L2 single-sc FAC product which are not visible in the average and the dual-sc current estimates. The comparison of the MSMVA FAC density with the average L2 single-sc current shows a significant decrease in the relative percentage differences (Table 2). A further inclusion of the orientation in the L2 single-sc product would probably lead to an even better agreement with the MSMVA result.

This example suggests that MSMVA can be indeed useful also for complex FAC signatures.

**3 Suggestions for improvements**

It may be useful to the reader if the manuscript included a step-by-step as to how the FAC scalograms are intended to be used and what information can be obtained/not obtained from them at each step. This could appear at the start of the Discussion and Summary section.

Section 6, Conclusions and outlook, was updated as suggested (second paragraph).

[Figure]

**Figure 1.** Upper panel: $\Delta B$ from SwA. Bottom panel: FAC density estimates, L2 single-s/c (green), L2 dual-s/c (black), average L2 single-s/c (red).

---

## Author Response (AR4)

**Topical Editor Decision: Publish subject to technical corrections**

*by Rumi Nakamura*
*Submitted on 29 Apr 2019*

**1 Comments to the Author:**

Dear Dr. Bunescu,

I am happy to write you that your manuscript will be accepted for publication, once you include your reponses (including Figure 1 of response) related to the referee's concerns on applying to the complex cases in your manuscript, which are currently missing except a couple of short sentences. These discussions are important to be included also in the manuscript.

Best regards,
Rumi Nakamura

Dear Editor,

Thank you very much for your evaluation of the manuscript "Multiscale estimation of the field-aligned current density".

The manuscript was updated by including Figure 1 from the previous response. As suggested we updated section (4.3) of the manuscript with the discussion included also in the previous response. The new added text is indicated with blue whereas the red color indicates the text from the previous iteration.

Yours sincerely,
Costel Bunescu